# A Theoretical Perspective: How to Prevent Model Collapse in Self-consuming Training Loops

**Shi Fu**[1]    **Yingjie Wang**[1*]    **Yuzhu Chen**[2]    **Xinmei Tian**[2]    **Dacheng Tao**[1*]

[1]Generative AI Lab, College of Computing and Data Science,
 Nanyang Technological University, Singapore 639798,
[2]University of Science and Technology of China, Hefei, China
`fs311@mail.ustc.edu.cn`, `yingjiewang@upc.edu.cn`,
`cyzkrau@mail.ustc.edu.cn`, `xinmei@ustc.edu.cn`, `dacheng.tao@gmail.com`

## Abstract

High-quality data is essential for training large generative models, yet the vast reservoir of real data available online has become nearly depleted. Consequently, models increasingly generate their own data for further training, forming Self-consuming Training Loops (STLs). However, the empirical results have been strikingly inconsistent: some models degrade or even collapse, while others successfully avoid these failures, leaving a significant gap in theoretical understanding to explain this discrepancy. This paper introduces the intriguing notion of *recursive stability* and presents the first theoretical generalization analysis, revealing how both model architecture and the proportion between real and synthetic data influence the success of STLs. We further extend this analysis to transformers in in-context learning, showing that even a constant-sized proportion of real data ensures convergence, while also providing insights into optimal synthetic data sizing.

## 1 Introduction

The quest of high-quality data is paramount in the training of generative artificial intelligence (AI), such as large language models (LLMs). However, the vast reservoir of publicly available data on the internet has nearly been exhausted (Villalobos et al., 2022), pushing the research community to seek innovative yet plausible solutions. One promising approach is to train the next generation of LLMs using synthetic data generated by earlier generations of the models themselves (Briesch et al., 2023). Additionally, reliance on synthetic data has become almost unavoidable, as many existing datasets are already polluted with synthetic content (Schuhmann et al., 2022), which proves difficult to detect reliably (Sadasivan et al., 2023). This has led to the development of Self-consuming Training Loops (STLs), as illustrated in Figure 1, where generative models are recursively trained on a mix of real and synthetic data generated by the models themselves. In theory, these STLs of data creation and refinement could propel models to new levels of capability, reducing reliance on external datasets.

However, despite their potential, the empirical results of STLs have been highly inconsistent across studies (Shumailov et al., 2024; Alemohammad et al., 2024a; Xing et al., 2025; Dohmatob et al., 2024b). Some studies (Shumailov et al., 2024) have encountered significant setbacks—certain models have shown signs of stagnation, failing to improve or adapt, while others have even regressed, leading to sharp declines in performance. Conversely, other works (Gerstgrasser et al., 2024; Gillman et al., 2024; Alemohammad et al., 2024b; Ferbach et al., 2024) have successfully avoided model collapse by incorporating sufficient real data, augmenting with synthetic data, or introducing guidance during the generation process. However, these observed phenomena lack thorough theoretical explanations.

When and how do STLs generalize effectively, thereby preventing model collapse from a theoretical perspective? Even among "refined" LLMs drawing from similar pools of model-generated data, the results vary significantly (Briesch et al., 2023; Fu et al., 2024a). These inconsistencies highlight the

---

*Corresponding authors

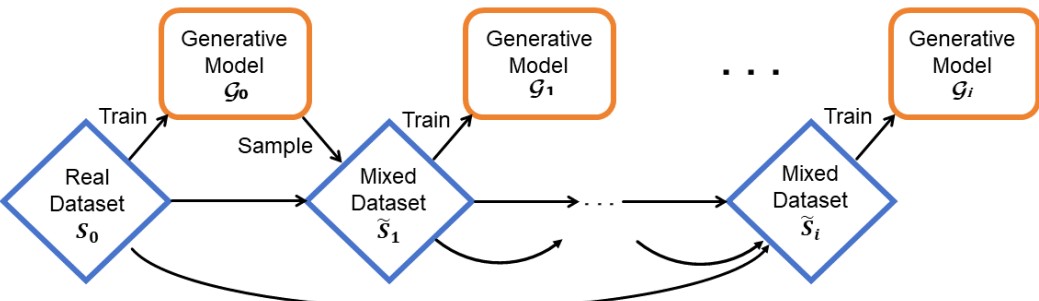

Figure 1: Self-consuming Training Loops: The initial model $\mathcal{G}_0$ is trained on the real dataset $S_0$. For generation $1 \leq j \leq i$, the model $\mathcal{G}_j$ is trained on the mixed dataset $\widetilde{S}_j$.

urgency of establishing theoretical guarantees for STLs by exploring the underlying mechanisms that determine when synthetic data generation either facilitates or impedes model development. Initial theoretical explorations have started to address these gaps. For instance, Shumailov et al. (2024) and Alemohammad et al. (2024a) demonstrated model collapse when models were trained exclusively on synthetic data, using simplified Gaussian models to illustrate this issue. In a more detailed theoretical study, Bertrand et al. (2024) derived upper bounds on parameter deviations for likelihood-based models in STLs, establishing convergence under strict statistical and optimization assumptions. Meanwhile, Fu et al. (2024b) relaxed these assumptions and provided bounds on the divergence between synthetic and real data distributions for a simplified diffusion model.

However, existing theoretical research lacks a unified framework and has yet to thoroughly investigate the generalization error of STLs. Additionally, current studies often overlook the role of model architectures in preventing model collapse. Moreover, the behavior of transformers within STLs remains largely unexamined, leaving significant theoretical gaps in the literature. Notably, there is limited exploration of the theoretical trade-offs introduced by synthetic data augmentation. This paper aims to address these gaps with the following contributions:

1. **Theoretical Generalization Framework:** This paper fills a gap in prior research by being the first to establish generalization error bounds. The key innovation, recursive stability, is introduced to address the core theoretical challenges, specifically the complex recursive structures and the non-i.i.d. nature of the data. Moreover, we demonstrate that the generalization error converges under the following conditions: (1) the generative model satisfies recursive stability, and (2) the proportion of real data is maintained at a non-negligible constant level, thus preventing model collapse.

2. **Application to Transformers in In-Context Learning:** This paper is the first to extend the theoretical framework to transformer models in in-context learning. We prove that transformers in this setting satisfy recursive stability with a constant-level proportion of real data, controlling output differences in STLs under small perturbations to the initial dataset. Consequently, we show that the generalization error is bounded by $\mathcal{O}(n^{-1}\log^2(n) + n^{-1/2}\log(n) + n^{-1/4})$.

3. **Trade-off Analysis of Synthetic Data Augmentation:** We investigate the trade-off in synthetic data augmentation. By employing decomposition techniques, we demonstrate that while synthetic data improves the generalization performance of each generation on mixed datasets, it concurrently exacerbates distribution divergence across successive generations. Our theoretical findings further show that the optimal size of synthetic augmentation increases as the size of real dataset decreases.

## 2 RELATED WORK

This section reviews STLs research and algorithmic stability studies.

**Self-consuming Training Loops**. Recent research has increasingly focused on generative models trained within STLs (Shumailov et al., 2024), with much of the analysis conducted from an empirical perspective (Martínez et al., 2023). For example, Shumailov et al. (2024); Briesch et al. (2023)

observe a decline in diversity in language models when a portion of the model's outputs is recursively used as inputs. Additionally, Wyllie et al. (2024) highlights that recursive training on synthetic data amplifies biases, resulting in significant fairness concerns. To mitigate model collapse, some studies suggest incorporating real data into the training process (Alemohammad et al., 2024a), expanding the size of synthetic datasets (Dohmatob et al., 2024c; Gerstgrasser et al., 2024; Dohmatob et al., 2024a; Feng et al., 2024b), or providing guidance during the generation process (Gillman et al., 2024; Alemohammad et al., 2024b; Feng et al., 2024a).

While empirical research has extensively explored STLs of generative models, theoretical studies on this process remain relatively sparse (Kanabar & Gastpar, 2025; Seddik et al., 2024; Marchi et al., 2024; Gerstgrasser et al., 2024; Zhu et al., 2024; Tao et al., 2024). Notably, Shumailov et al. (2024) and Alemohammad et al. (2024a) offer initial theoretical insights by analyzing a simplified Gaussian model. In a more comprehensive analysis, Bertrand et al. (2024) derive upper bounds on parameter deviations between those obtained within a STL and the optimal values, relying on assumptions about statistical and optimization error bounds. In contrast, Fu et al. (2024b) propose bounds on the divergence between synthetic and real-world data distributions, without such assumptions. However, current research lacks a unified theoretical framework that accounts for the influence of different model architectures and does not provide generalization error bounds for STLs, thus failing to rigorously establish the conditions that guarantee the prevention of model collapse. Furthermore, the behavior of transformers within STLs remains unexplored, leaving substantial theoretical gaps.

**Algorithmic stability**. Algorithmic stability ensures generalization bounds independent of model capacity. A key measure, uniform stability, was introduced by Bousquet & Elisseeff (2002) and has been instrumental in analyzing the generalization behavior of regularization methods. This measure was later extended to SGD (Hardt et al., 2016), including non-convex and non-smooth settings (Charles & Papailiopoulos, 2018; Bassily et al., 2020; Lei, 2023). Recent work shows that uniform stability can also provide near-optimal bounds with high probability (Feldman & Vondrak, 2019; Bousquet et al., 2020; Klochkov & Zhivotovskiy, 2021; Li & Liu, 2022; Wang et al., 2024).

Building on these foundations, recent research has focused on stability in more complex, non-i.i.d. settings. A common approach models data from stationary and mixing sequences (Doukhan, 1994; Yu, 1994), where weakening dependencies allow stability bounds through mixing coefficients (Mohri & Rostamizadeh, 2010; He et al., 2016; Fu et al., 2023). However, estimating these coefficients remains challenging. Additionally, some studies (Zheng et al., 2023) address non-i.i.d. data by leveraging conditional independence properties. Nonetheless, current methodologies struggle with the complexities of STLs, as the non-i.i.d. nature of mixed datasets, where each generation's data is influenced by previous generations, presents unresolved challenges for stability frameworks.

**Remark 1.** Building on previous challenges, our work advances this area by developing a more comprehensive theoretical framework for analyzing generative models within STLs. Specifically, we present the first generalization error bound by addressing the additional complexity arising from the non-i.i.d. nature of mixed datasets. To address this, we propose the key innovation of recursive stability, which quantifies error propagation across generations of synthetic data. Moreover, we are the first to extend this theoretical framework to transformers, explicitly utilizing error decomposition to illustrate the trade-off introduced by augmenting datasets with synthetic data.

## 3 PRELIMINARIES

In this section, we begin by formally describing the training process of generative models in STLs, then introduce algorithmic stability with a focus on uniform stability, and finally define recursive stability to address the challenges specific to STLs.

### 3.1 GENERATIVE MODELS WITHIN SELF-CONSUMING TRAINING LOOPS

Generative models have made significant strides in producing highly realistic data, such as images and text, which are frequently shared online and often indistinguishable from real content. Meanwhile, the supply of real data has nearly been exhausted. Consequently, deep generative models increasingly rely on synthetic data, either unintentionally (Schuhmann et al., 2022) or intentionally (Huang et al., 2022). This reliance creates a recursive cycle where successive generations are trained on mixed datasets of real and synthetic data, a process known as an STL, as shown in Figure 1.

More concretely, we explore a stochastic process that evolves through sequential generations. In an STL, we start with an initial dataset $S_0$, consisting of real data points $z \in \mathcal{Z}$, sampled from the original real distribution $\mathcal{D}_0$. The initial generative model $\mathcal{G}_0$ is trained on this real dataset $S_0$, producing the first generation synthetic dataset $S_1$, whose distribution is denoted as $\mathcal{D}_1$. Next, the real dataset $S_0$ is combined with the synthetic dataset $S_1$ in a certain proportion to form a new mixed dataset $\widetilde{S}_1$, with distribution $\widetilde{\mathcal{D}}_1$. The next generation generative model $\mathcal{G}_1$ is then trained on this mixed dataset $\widetilde{S}_1$. Moving forward, for each subsequent generation $1 \leq j \leq i$, the mixed dataset $\widetilde{S}_j$ is composed of real data and synthetic data from previous generations. The generative model $\mathcal{G}_j$ is trained on $\widetilde{S}_j$, producing the synthetic dataset $S_{j+1}$. This STL proceeds iteratively until the maximum generation, denoted as $i$, is reached.

## 3.2 ALGORITHMIC STABILITY

Algorithmic stability measures the impact of modifying or removing a small number of examples from the training set on the resulting model, a key concept in statistical learning theory (Bousquet & Elisseeff, 2002). Its primary advantage lies in providing generalization bounds independent of model capacity. Among various stability notions (Shalev-Shwartz et al., 2010), we focus on uniform stability, the most widely studied form. Let $S$ and $S'$ be two datasets differing by one point. Then, we formally define uniform stability as follows:

**Definition 1.** (Uniform Stability (Bousquet & Elisseeff, 2002)). Algorithm $\mathcal{A}$ is uniformly $\beta_n$-stable with respect to the loss function $\ell$ if the following holds

$$\forall S, \ S' \in \mathcal{Z}^n, \ \forall \boldsymbol{z} \in \mathcal{Z}, \ \sup_{\boldsymbol{z}} |\ell(\mathcal{A}(S), \boldsymbol{z}) - \ell(\mathcal{A}(S'), \boldsymbol{z})| \leq \beta_n.$$

Traditional notions of stability have predominantly been studied in the context of learning algorithms, such as SGD (Lei & Ying, 2020). More recently, there has been significant progress in extending the concept of stability to generative models (Farnia & Ozdaglar, 2021; Zheng et al., 2023; Li et al., 2023). Building on these advancements, we propose *recursive stability* to specifically address generative models within STLs. This new stability measure is designed to quantify the differences in a generative model's outputs after multiple generations of recursive training when small perturbations are applied to the initial real dataset. The formal definition of recursive stability is presented below.

**Definition 2.** (Recursive Stability) Let $S_0$ represent the original real dataset, and $S_0'$ denote a dataset differing from $S_0$ by a single example. A generative model $\mathcal{G}$ is said to be recursively $\gamma_n^{i,\alpha}$-stable with respect to the distance measure $d$ after the $i$-th generation of STLs, where the ratio of real to synthetic data is set to $\alpha$, if the following condition holds:

$$\forall S_0, S_0' \in \mathbb{Z}^n, \quad d\left(\mathcal{G}^{(i)}(S_0), \mathcal{G}^{(i)}(S_0')\right) \leq \gamma_n^{i,\alpha}.$$

where $\mathcal{G}^{(i)}$ denotes the output of the generative model at the $i$-th generation in the STLs. The distance measure $d$ quantifies the deviation between the outputs generated from inputs $S_0$ and $S_0'$ across STLs. Specifically, $d$ can be defined using Total Variation (TV) distance, Kullback-Leibler (KL) divergence, or various norms (e.g., $\ell_2$ norm), allowing flexibility in assessing the differences in generated outputs.

## 4 GENERAL THEORETICAL RESULTS

In this section, we present a general framework for analyzing generalization error. Moving beyond traditional analyses of parameter changes (Bertrand et al., 2024) and distributional discrepancies (Fu et al., 2024b), we focus on evaluating the utility of synthetic data after recursive training (Hittmeir et al., 2019; Xu et al., 2023). Specifically, we examine the behavior of a uniformly stable learning algorithm $\mathcal{A}$ trained on the mixed dataset $\widetilde{S}_i$ in the $i$-th generation. Our goal is to study the generalization error of the hypothesis $\mathcal{A}(\widetilde{S}_i)$. Formally, we aim to bound $|R_{\mathcal{D}_0}(\mathcal{A}(\widetilde{S}_i)) - \widehat{R}_{\widetilde{S}_i}(\mathcal{A}(\widetilde{S}_i))|$, where $R_{\mathcal{D}_0}(\mathcal{A}(\widetilde{S}_i)) = \mathbb{E}_{\boldsymbol{z} \sim \mathcal{D}_0}[\ell(\mathcal{A}(\widetilde{S}_i), \boldsymbol{z})]$ represents the population risk of $\mathcal{A}(\widetilde{S}_i)$ under the real distribution $\mathcal{D}_0$, and $\widehat{R}_{\widetilde{S}_i}(\mathcal{A}(\widetilde{S}_i)) = \frac{1}{n} \sum_{\boldsymbol{z}_i \in \widetilde{S}_i} \ell(\mathcal{A}(\widetilde{S}_i), \boldsymbol{z}_i)$ denotes the empirical risk on the mixed dataset. To derive this bound, we first decompose the generalization error as follows.

$$\left| R_{\mathcal{D}_0}(\mathcal{A}(\widetilde{S}_i)) - \widehat{R}_{\widetilde{S}_i}(\mathcal{A}(\widetilde{S}_i)) \right| \leq \underbrace{\left| R_{\mathcal{D}_0}(\mathcal{A}(\widetilde{S}_i)) - R_{\widetilde{\mathcal{D}}_i}(\mathcal{A}(\widetilde{S}_i)) \right|}_{\text{Cumulative distribution shift across generations}} + \underbrace{\left| R_{\widetilde{\mathcal{D}}_i}(\mathcal{A}(\widetilde{S}_i)) - \widehat{R}_{\widetilde{S}_i}(\mathcal{A}(\widetilde{S}_i)) \right|}_{\text{Generalization error on mixed distributions}}.$$

The first term captures the accumulation of error and distribution divergence over multiple generations within the STLs. This heavily depends on the capacity of the generative model to preserve distributional fidelity across generations, requiring recursive techniques to manage error propagation. The second term reflects the generalization performance of the learning algorithm on the non-i.i.d. mixed dataset, where synthetic data points are influenced by the initial real dataset. Drawing on Zheng et al. (2023), we observe that while $S_0$ satisfies the i.i.d. assumption, the synthetic datasets $S_i$ follow a conditional i.i.d. assumption given $S_0$. Leveraging this, along with moment bounds and concentration inequalities, we address the challenge of bounding the second term and managing dependencies within the STLs. We now present the following result.

**Theorem 1** (General Generalization Bound). *Assume that $\mathcal{A}$ is a $\beta_n$-uniformly stable learning algorithm and the loss function $\ell$ is bounded by $M$. Let $n$ represent the sample size of the mixed dataset $\widetilde{S}_j$, defined as $\widetilde{S}_j = \alpha S_0 + (1-\alpha)S_j$ for $1 \leq j \leq i$, where $0 < \alpha \leq 1$ denotes the proportion of real data. Assume further that the generative model $\mathcal{G}$ is recursively $\gamma_n^i$-stable, and the TV distance for each generation $TV(\widetilde{\mathcal{D}}_j, \mathcal{D}_{j+1})$ is of the same order, denoted by $d_{\mathrm{TV}}(n)$. Then, for any $\delta \in (0, 1)$, with probability at least $1 - \delta$, the following holds:*

$$\left| R_{\mathcal{D}_0}(\mathcal{A}(\widetilde{S}_i)) - \widehat{R}_{\widetilde{S}_i}(\mathcal{A}(\widetilde{S}_i)) \right| \lesssim \gamma_n^i \alpha M \log(n\alpha) \log(1/\delta) + n^{-1/2} M \sqrt{\log 1/\delta}$$

$$+ \beta_n \left( \log n \log(1/\delta) + \alpha \sqrt{(1-\alpha)n \log(1/\delta)} \right) + d_{\mathrm{TV}}(n) M \left( 1 - (1-\alpha)^i \right) \alpha^{-1}, \quad (1)$$

*where $\gamma_n^i = \sup_j TV(\mathcal{D}_i^{n(1-\alpha)}(S_0'), \mathcal{D}_i^{n(1-\alpha)}(S_0))$, with $S_0$ and $S_0'$ representing two real datasets of size $n$, differing by only a single data point.*

**Remark 2. Recursive Stability in STLs**. In Theorem 1, the recursive stability parameter is quantified using the TV distance to measure the divergence between the distributions of the $n(1 - \alpha)$ synthetic data points generated by the model $\mathcal{G}_i$ at the $i$-th generation. Notably, the concept of recursive stability, introduced in Definition 2, is adaptable to various metrics, making it applicable across different types of generative models. In Theorem 2, the recursive stability parameter for transformers is instead defined using the $\ell_2$ norm between tokens, allowing this concept to be generalized to a broader range of model architectures.

Moreover, Theorem 1 demonstrates that generative models with higher recursive stability exhibit better performance after undergoing the STL. Specifically, the results indicate that the convergence rate of recursive stability parameter is at least faster than $\mathcal{O}(1/\log n)$, which is a relatively mild condition. Furthermore, Theorem 2 shows that, under mild assumptions, the recursive stability parameter for transformers in in-context learning settings achieves a convergence rate of $\gamma_n^i = \mathcal{O}(1/n)$ when measured by the $\ell_2$ norm between tokens.

**Remark 3. Effect of Real Data Proportion on Error Control**. Previous experimental results (Shumailov et al., 2024; Alemohammad et al., 2024a) have demonstrated that incorporating real data can mitigate model collapse and help control errors. This remark focuses on exploring the effect of the real data proportion $\alpha$ on the generalization error within the STLs. As shown in Theorem 1, the real data proportion $\alpha$ plays a significant role in the cumulative distribution shift across generations, specifically in the term $2M \left( 1 - (1-\alpha)^i \right) \alpha^{-1} d_{\mathrm{TV}}(n)$.

When $\alpha \to 0$, we observe that $\frac{(1-(1-\alpha)^i)}{\alpha} \to i$, leading to a linear accumulation of errors due to the Distribution Shift, making it increasingly challenging to control the overall error. This observation aligns with the theoretical results reported in Shumailov et al. (2024); Dohmatob et al. (2024a); Fu et al. (2024b). However, it is important to note that the conditions on $\alpha$ for controlling this term are not strict. In fact, as long as $\alpha$ remains at a non-negligible constant level, the expression $\left( 1 - (1-\alpha)^i \right) \alpha^{-1}$ remains bounded, effectively controlling the error. This aligns with theoretical intuition: when $\alpha$ is too small, the mixed dataset contains insufficient real data, resulting in a more severe distribution shift.

Moreover, the proportion of real data $\alpha$ also impacts the generalization error on mixed distributions, primarily through its effect on the recursive stability parameter $\gamma_n^i$. As $\alpha$ increases, the generative model becomes more recursively stable. We will further explore the influence of $\alpha$ on the recursive stability parameter $\gamma_n^i$ for specific generative models, such as transformers, in Theorem 3, particularly in Remark 8.

**Remark 4. Convergence Rate of Uniform Stability Parameter**. With respect to the uniform stability parameter $\beta_n$, we observe from the third term on the right-hand side of inequality 1 that the

convergence rate of $\beta_n$ must be at least $\mathcal{O}(1/\sqrt{n})$ to adequately control the error. This is a relatively mild requirement.

For example, in the case of widely used algorithms such as SGD, it has been shown that the uniform stability parameter $\beta_n$ converges at a rate of $\mathcal{O}(\log(n)/n)$ under the assumptions of Lipschitz continuity and smoothness of the loss function (Zhang et al., 2022). Additionally, for regularization-based algorithms, such as kernel regularization schemes and the Minimum Relative Entropy (MRE) algorithm, it has been demonstrated that $\beta_n$ can achieve a convergence rate of $\mathcal{O}(1/n)$ under certain conditions (Bousquet & Elisseeff, 2002).

**Remark 5. Convergence of the Distribution Shift Term** $d_{\mathrm{TV}}(n)$. Regarding the convergence of the term $2M\left(1-(1-\alpha)^i\right)\alpha^{-1}d_{\mathrm{TV}}(n)$, as discussed in Remark 3, when $\alpha$ remains a non-negligible constant, attention turns to the distribution shift term $d_{\mathrm{TV}}(n)$. This term critically depends on the generative model's capacity and quantifies the divergence between the learned distribution and the input distribution in each generation.

Theoretical studies have provided various convergence rates for $d_{\mathrm{TV}}(n)$ across different generative models. For instance, in diffusion models, $d_{\mathrm{TV}}(n)$ has been shown to converge at a rate of $\mathcal{O}\left(1/n^{1/4}\right)$ (Fu et al., 2024b). Similarly, for GANs, the convergence rate is also $\mathcal{O}\left(1/n^{1/4}\right)$ (Liang, 2021). More generally, by applying Pinsker's inequality to relate KL divergence and TV distance, the convergence rates for other models, such as Bias potential models and Normalizing flows, have been explored in previous works (Yang, 2022). Additionally, we will further examine the behavior of transformer models in Theorem 3, demonstrating the flexibility of our theoretical framework across a wide range of generative models.

**Remark 6. Comparision with Previous Works**. In the realm of theoretical research on the STL, where models are recursively trained on the synthetic data they generate, the foundational work was introduced by Shumailov et al. (2024) and Alemohammad et al. (2024a). They provided the initial theoretical definitions and analyzed the behavior of a simplistic multivariate Gaussian toy model in such loops. However, their analyses were limited to basic theoretical insights and lacked in-depth exploration of more complex generative models.

Recent advancements in this field have primarily come from Bertrand et al. (2024) and Fu et al. (2024b). Bertrand et al. (2024) established an upper bound on the deviation of likelihood-based model output parameters from the optimal ones, denoted as $\|\theta_i - \theta^*\|$. This was achieved by making direct assumptions on the upper bounds of both statistical and optimization errors in generative models, as outlined in their Assumption 3. In contrast, Fu et al. (2024b) derived bounds on the TV distance, addressing the distribution divergence between the synthetic data distributions produced by future models and the original real data distribution, with a specific focus on diffusion models. Our work makes significant theoretical advancements over both Bertrand et al. (2024) and Fu et al. (2024b) in several key aspects:

1. **Innovative Concept of Recursive Stability.** A central technical contribution of our work is the extension of the traditional notion of algorithmic stability. We define recursive stability, a crucial factor for controlling error propagation across generations. This novel concept tackles the theoretical challenges posed by non-i.i.d. data and recursive structures within STLs, while also incorporating the influence of model architectures into the generalization error. Moreover, recursive stability serves as a new measure for assessing the stability of generative models within STLs. In Theorem 2, we further establish an upper bound on the recursive stability parameter for transformers under mild conditions, underscoring the broad applicability and robustness of our framework.

2. **Establishing the First Generalization Error Bound for STLs.** While Bertrand et al. (2024) primarily focused on parameter deviations in generative models and Fu et al. (2024b) concentrated on distribution divergence, our work emphasizes the utility of the generated data produced by STLs. Specifically, by utilizing recursive stability, we present the first generalization error bound that quantifies the gap between the population risk on the initial real data distribution $\mathcal{D}_0$ and the empirical risk of the hypothesis $\mathcal{A}(\widetilde{S}_i)$, generated by applying learning algorithms to the synthetic data produced after multiple generations of STLs. This introduces a new layer of complexity compared to prior work, as it necessitates handling not only the distribution shifts within STLs but also the challenges arising from the non-i.i.d. nature of the mixed datasets, where each generation's data is influenced by all preceding generations.

3. **A More General Framework Accounting for Model Structure.** Our proposed theoretical framework is more comprehensive than previous studies. Bertrand et al. (2024) restricted their analysis to simplified likelihood-based generative models, while Fu et al. (2024b) focused specifically on diffusion models. Importantly, neither of their theoretical results accounted for the impact of different model architectures. In contrast, as discussed in Remark 5, our framework explicitly incorporates the effects of varying model structures, thereby extending its applicability to a broader range of generative models. Notably, we are the first to extend the theory of SLTs to transformer models, further broadening the scope of our approach across diverse generative model architectures.

4. **Comprehensive Collapse Prevention Through Recursive Stability**. In addition to the existing theoretical work, which primarily analyzes conditions to avoid model collapse based on the proportion of real data (e.g., Bertrand et al. (2024); Fu et al. (2024b)), our work extends these analyses by considering the impact of model architecture. Specifically, Theorem 1 demonstrates that under a recursive stability condition and a non-negligible constant level of real data, model collapse can be avoided across a variety of model architectures. This analysis offers broader conditions for preventing collapse by incorporating recursive stability, deepening the understanding of how model architecture affects training robustness.

**Remark 7. Proof Sketch of Theorem 1**. We first decompose the generalization error of STLs into two distinct terms: (1) the cumulative distribution shift across generations, and (2) the generalization error on the mixed dataset.

**Cumulative Distribution Shift:** This term measures the shift between the real dataset $\mathcal{D}_0$ and the mixed distribution $\mathcal{D}_i$ after the $i$-th generation. Using the TV distance to quantify the shift introduced by the generative model, we bound the difference as:

$$\left| R_{\mathcal{D}_0}(\mathcal{A}(\widetilde{S}_i)) - R_{\widetilde{\mathcal{D}}_i}(\mathcal{A}(\widetilde{S}_i)) \right| \leq (1-\alpha) \left| R_{\mathcal{D}_0}(\mathcal{A}(\widetilde{S}_i)) - R_{\widetilde{\mathcal{D}}_{i-1}}(\mathcal{A}(\widetilde{S}_i)) \right| + 2(1-\alpha)MTV(\widetilde{\mathcal{D}}_{i-1}, \mathcal{D}_i).$$

By leveraging the recursive structure of the generative process, this cumulative distribution shift can be bounded across all generations as:

$$|R_{\mathcal{D}_0}(A(S_i)) - R_{\mathcal{D}_i}(A(S_i))| \leq 2M \left(1 - (1 - \alpha)^i\right) \alpha^{-1} d_{\text{TV}}(n).$$

**Generalization Error on the Mixed Dataset:** The second term quantifies the generalization error when training on the mixed dataset $\widetilde{S}_i$, which consists of both real and synthetic data. Our goal is to establish a moment bound on the generalization error, which can be decomposed as follows:

$$\|\alpha R_{\mathcal{D}_0}(\mathcal{A}(\widetilde{S}_i)) - \frac{1}{n} \sum_{\boldsymbol{z}_i \in S_{0,\alpha}} \ell(\mathcal{A}(\widetilde{S}_i), \boldsymbol{z}_i)\|_p + \|(1-\alpha) R_{\mathcal{D}_i}(\mathcal{A}(\widetilde{S}_i)) - \frac{1}{n} \sum_{\boldsymbol{z}_i \in S_{i,1-\alpha}} \ell(\mathcal{A}(\widetilde{S}_i), \boldsymbol{z}_i)\|_p.$$

In this context, $S_{0,\alpha}$ represents a proportion $\alpha$ of the $n$ data points in $S_0$, leading to a total of $n \times \alpha$ data points. Similarly, $S_{i,1-\alpha}$ denotes a subset of the synthetic dataset $S_i$, where $S_{i,1-\alpha} \subseteq S_i$ and its size is $(1 - \alpha) \times |S_i|$. For each term, we leverage the uniform stability $\beta_n$ of the learning algorithm $\mathcal{A}$ and the recursive stability $\gamma_n^i$ of the generative model to address the non-i.i.d. nature of the mixed dataset. The mixed dataset exhibits conditional independence (Zheng et al., 2023), with synthetic data conditioned on the initial real dataset $S_0$, allowing the application of recursive techniques to derive the moment bound. Subsequently, Lemma 8 and Lemma 9 are utilized to derive the high-probability bound for the final result.

## 5 THEORETICAL ANALYSIS OF TRANSFORMERS IN IN-CONTEXT LEARNING

In this section, we first present the transformer in in-context learning (ICL) and its settings within SLTs in Section 5.1. In Section 5.2, we prove that it satisfies recursive stability, followed by the derivation of the generalization error bound for transformers in ICL in Section 5.3. Finally, in Section 5.4, we explore the scenario of synthetic data augmentation and investigate the associated trade-offs.

### 5.1 SETTINGS OF TRANSFORMER IN IN-CONTEXT LEARNING

**In-Context Learning Setting**. ICL involves a transformer model processing a sequence of input-output examples to perform inference without parameter updates. Unlike traditional supervised learning, where a model is trained on a fixed dataset and then makes predictions, ICL allows the model

to adapt on-the-fly to new queries based on the provided examples. We denote a prompt, containing $n$ in-context examples followed by the $(n + 1)$-th query input, as $S_0 = (\boldsymbol{z}_1, \boldsymbol{z}_2, \ldots, \boldsymbol{z}_n, \boldsymbol{x}_{n+1})$, where $(\boldsymbol{z}_i)_{i=1}^n = (\boldsymbol{x}_i, \boldsymbol{y}_i)_{i=1}^n \in \mathcal{Z} = \mathcal{X} \times \mathcal{Y}$ represents i.i.d. in-context samples, and $\boldsymbol{x}_{n+1} \in \mathcal{X}$ is the query input whose label we want to predict. The transformer model, denoted as $\mathrm{TF}(\cdot)$, takes the prompt $S_0$ as input and outputs the predicted label $\hat{\boldsymbol{y}}_{n+1}$ for the query $\boldsymbol{x}_{n+1}$: $\hat{\boldsymbol{y}}_{n+1} = \mathrm{TF}(S_0)$.

**Recursive Data Generation in STLs with ICL**. We extend the traditional ICL setting to an STL, where the transformer recursively generates new data using its own ICL predictions. Starting with an initial real dataset $S_0$, this serves as the initial real in-context examples for the transformer. The process begins by sampling the first generation queries $\{\boldsymbol{x}_{1,j}\}_{j=1}^n$ i.i.d. from the input distribution $\mathcal{X}$. Each query $\boldsymbol{x}_{1,j}$ is incorporated into the in-context examples from $S_0$ as a new query $\boldsymbol{x}_{0,n+1}$, and the transformer predicts the corresponding label $\hat{\boldsymbol{y}}_{1,j}$. This produces a synthetic dataset $S_1$, consisting of inputs $\{\boldsymbol{x}_{1,j}\}_{j=1}^n$ and their predicted labels $\{\hat{\boldsymbol{y}}_{1,j}\}_{j=1}^n$. A mixed dataset $\widetilde{S}_j$ is then formed and used as the in-context examples for the next generation. This process continues, with each generation producing a new synthetic dataset $S_{j+1}$ based on the updated mixed dataset $\widetilde{S}_j$.

## 5.2 Recursive Stability of In-Context Learning with Transformers

In this section, we demonstrate that transformers exhibit recursive stability within the ICL framework. Following the ICL setting from Li et al. (2023), we show that the model effectively controls error propagation from perturbations in the initial real dataset, ensuring stability across the STLs.

**Theorem 2.** *Let $S_0, S_0'$ be two initial real datasets that only differ at the inputs $\boldsymbol{z}_j = (\boldsymbol{x}_j, \boldsymbol{y}_j)$ and $\boldsymbol{z}_j' = (\boldsymbol{x}_j', \boldsymbol{y}_j')$ where $1 \le j \le n$. Assume the inputs and labels lie within the unit Euclidean ball in $\mathbb{R}^d$. Represent the prompts $S_0$ and $S_0'$ as matrices $\boldsymbol{Z}_0, \boldsymbol{Z}_0' \in \mathbb{R}^{(2n+1) \times d}$. Let $\mathrm{TF}(\cdot)$ be an $L$-layer transformer. Given $\boldsymbol{Z}_0$ as the initial input, the $k$-th layer applies MLPs and self-attention, producing the output:*

$$\boldsymbol{Z}_k = \mathrm{Parallel\_MLPs}(\mathrm{ATTN}\left(\boldsymbol{Z}_{k-1}\right)) \text{ where } \mathrm{ATTN}(\boldsymbol{Z}) := \mathrm{softmax}\left(\boldsymbol{Z}\boldsymbol{W}\boldsymbol{Z}^\top\right)\boldsymbol{Z}\boldsymbol{V}.$$

*Assume $\mathrm{TF}$ is normalized as $\|\boldsymbol{V}\| \le 1, \|\boldsymbol{W}\| \le B_W$ and MLPs obey $\mathrm{MLP}(\boldsymbol{z}) = \mathrm{ReLU}(\boldsymbol{M}\boldsymbol{z})$ with $\|\boldsymbol{M}\| \le 1$. Let $\mathrm{TF}$ output the last token of the final layer $\boldsymbol{Z}_L$ that corresponds to the query $\boldsymbol{x}_{j,n+1}$. Let $n$ represent the sample size of the mixed dataset $\widetilde{S}_j$, where $\widetilde{S}_j = \alpha S_0 + (1 - \alpha)S_j$ for $1 \le j \le i$. Then, we obtain:*

$$\left\| \mathrm{TF}(\widetilde{S}_i) - \mathrm{TF}(\widetilde{S}_i') \right\|_{\ell_2} \lesssim (1 - \alpha)^i \frac{\widetilde{B}_W^{(i+1)L}}{2n + 1},$$

*where $\widetilde{B}_W = (1 + 2B_W)\, e^{2B_w}$ and $\widetilde{S}_i'$ denotes the mixed dataset at the $i$-th generation in the STL when the initial real dataset is $S_0'$. Additionally, if the measure $d$ for the recursive stability parameter in Definition 2 is taken as the $\ell_2$ norm, then the recursive stability $\gamma_n^i \lesssim (1 - \alpha)^i \frac{\widetilde{B}_W^{(i+1)L}}{2n+1}$.*

**Remark 8. Controlling Exponential Growth with Real Data Proportion**. In this remark, we further investigate the influence of the proportion of real data $\alpha$ on the recursive stability of transformers. As outlined in Theorem 2, the upper bound of the recursive stability parameter includes a term that grows exponentially with the number of generations $i$ in the STL, specifically $\widetilde{B}_W^{(i+1)L}$. However, we show that even a constant proportion of real data, $\alpha$, is sufficient to control this growth.

Specifically, setting $\alpha = \Omega(1 - \widetilde{B}_W^{-((i+1)L)/i})$, we establish that the recursive stability parameter in Theorem 2 satisfies $\gamma_n^i \lesssim \frac{1}{2n+1}$. Additionally, as the number of generations $i$ in the STL approaches infinity, the proportion $\alpha$ asymptotically converges to $1 - \widetilde{B}_W^{-L}$. Notably, the depth $L$ is typically small in practical settings. For example, studies on LLM performance in STLs, such as Briesch et al. (2023), often employ models with $L = 6$. Furthermore, techniques like layer normalization effectively constrain the norm of weights $B_W$, ensuring numerical stability during training. Thus, with a constant real data proportion $\alpha$ independent of the STL generation number $i$, the exponential growth term $\widetilde{B}_W^{(i+1)L}$ can be effectively controlled, ensuring that $\gamma_n^i = \mathcal{O}(1/n)$.

## 5.3 Generalization Bound for Transformers in In-Context Learning

In this section, we investigate the behavior of transformers under the ICL framework in STLs. We select SGD as the learning algorithm $\mathcal{A}$ and consider a binary task with $\mathcal{Y} = \{0, 1\}$. Applying

our general theoretical framework from Theorem 1, we derive the generalization error bound by addressing the terms $\beta_n$ and $d_{\text{TV}}(n)$ using recent results on SGD (Zhang et al., 2022) and ICL (Zhang et al., 2023). The recursive stability parameter $\gamma_n^i$ is obtained from Theorem 2. We assume that the loss function $\ell(\cdot; z)$ is $\kappa$-smooth and $\rho$-Lipschitz, which are standard assumptions in related works (Hardt et al., 2016; Lei & Ying, 2020), with formal definitions provided in Appendix A.1. Examples include logistic and Huber losses. We now present the generalization error bound:

**Theorem 3.** *Consider an L-layer transformer under the setting described in Theorem 2. Let $n$ represent the sample size of the mixed dataset $\widetilde{S}_j$, where $\widetilde{S}_j = \alpha S_0 + (1 - \alpha)S_j$ for $1 \leq j \leq i$. Suppose that the loss function $\ell(\cdot; z)$ is $\kappa$-smooth, $\rho$-Lipschitz and bounded by $M > 0$ for every $z$. Let $\mathcal{A}(\widetilde{S}_i)$ denote the output after running SGD for $T \gtrsim n$ iterations with a step size $\eta_t = \mathcal{O}(\frac{1}{\kappa t})$ on the mixed dataset $\widetilde{S}_i$. Then, for any $\delta \in (0, 1)$, with probability at least $1 - \delta$, the following holds:*

$$\left| R_{\mathcal{D}_0}(\mathcal{A}(\widetilde{S}_i)) - \widehat{R}_{\widetilde{S}_i}(\mathcal{A}(\widetilde{S}_i)) \right| \lesssim n^{-1/2} \log(n) M \rho^2 \alpha \sqrt{1 - \alpha} \log \frac{1}{\delta}$$

$$+ n^{-1} \log^2(n) \rho^2 ((1 - \alpha) \widetilde{B}_W^L)^i \alpha \log(\frac{1}{\delta}) + n^{-1/4} \alpha^{-1} M \left( 1 - (1 - \alpha)^i \right) \log(\frac{1}{\delta}). \quad (2)$$

**Remark 9.** In this remark, we provide a detailed explanation of the theoretical results of Theorem 3. As discussed earlier in Remark 8, $\alpha$ is set to $1 - \widetilde{B}_W^{-L}$. To enhance clarity and focus on the primary results, we omit constant terms and the $\log(1/\delta)$ factor. Consequently, the bound in Theorem 3 can be expressed as follows:

$$\left| R_{\mathcal{D}_0}(\mathcal{A}(\widetilde{S}_i)) - \widehat{R}_{\widetilde{S}_i}(\mathcal{A}(\widetilde{S}_i)) \right| \lesssim n^{-1/2} \log(n) + n^{-1} \log^2(n) + n^{-1/4}.$$

In this bound, the terms $n^{-1/2} \log(n) + n^{-1} \log^2(n)$ correspond to the generalization error on the mixed dataset, while the term $n^{-1/4}$ represents the cumulative distribution shift across generations, which is primarily governed by the learnability of the generative model.

It is evident from this result that the generative model's capacity plays a crucial role in the performance within the STLs. The ability of the generative model to maintain distributional fidelity over multiple generations directly impacts the generalization error and determines how well the model can control the propagation of errors across generations.

## 5.4 Synthetic Data Augmentation

The previous theorem addresses the scenario where the training dataset is unintentionally contaminated by synthetic data, leading to STLs. In contrast, many researchers intentionally incorporate synthetic data to augment the real dataset, also creating STLs. Next, we explore this synthetic data augmentation scenario, where each generation's synthetic data is added to the mixed dataset, i.e., $\widetilde{S}_i = \sum_{j=0}^{i} S_j$.

**Theorem 4.** *Consider an L-layer transformer under the setting described in Theorem 2. Let $n$ and $\lambda n$ represent the sample size of the real dataset $S_0$ and the synthetic dataset $S_j$, respectively, where $1 \leq j \leq i$. The mixed dataset $\widetilde{S}_i$ is denoted as $\sum_{j=0}^{i} S_j$. Suppose that the loss function $\ell(\cdot; z)$ is $\kappa$-smooth, $\rho$-Lipschitz and bounded by $M > 0$ for every $z$. Let $\mathcal{A}(\widetilde{S}_i)$ denote the output after running SGD for $T \gtrsim n$ iterations with a step size $\eta_t = \mathcal{O}(\frac{1}{\kappa t})$ on the mixed dataset $S_i$. Then, for any $\delta \in (0, 1)$, with probability at least $1 - \delta$, the following holds:*

$$\left| R_{\mathcal{D}_0}(\mathcal{A}(\widetilde{S}_i)) - \widehat{R}_{\widetilde{S}_i}(\mathcal{A}(\widetilde{S}_i)) \right| \lesssim n^{-\frac{1}{4}} \log((1 + i\lambda)n) M \log \frac{1}{\delta}$$

$$+ n^{-1} \frac{\rho^2}{(1 + i\lambda)^2} \log((1 + i\lambda)n) i! \widetilde{B}_W^{(i+1)L} \log \frac{1}{\delta} + n^{-\frac{1}{2}} \frac{Mi}{1 + i\lambda} \sqrt{\log \frac{1}{\delta}}.$$

**Remark 10. Analyzing the Trade-off in Synthetic Data Augmentation for STLs**. In this remark, we examine the trade-off between generalization and distribution shifts from increased synthetic data, providing insights into optimal synthetic data size. At each generation, $\lambda n$ synthetic data points are added to the mixed dataset. We analyze how the coefficient $\lambda$, representing the scale of synthetic data augmentation, affects the generalization error in STLs. From the bound in Theorem 4, we observe that the **Cumulative Distribution Shift Across Generations** term is expressed as:

$$n^{-\frac{1}{4}} \log((1 + i\lambda)n) M \log(1/\delta).$$

As the coefficient $\lambda$ increases, the cumulative distribution shift correspondingly grows, thereby amplifying the associated error. This behavior aligns with intuition, as an increase in $\lambda$ reduces the proportion of real data within the mixed dataset at each generation. Consequently, this reduction in real data leads to a greater divergence between the mixed distribution and the true underlying distribution, exacerbating the deviation and compounding the error across successive generations. In contrast, for the **Generalization Error on Mixed Distributions** term:

$$n^{-1}\frac{\rho^2}{(1+i\lambda)^2}\log((1+i\lambda)n)i!\widetilde{B}_W^{(i+1)L}\log\frac{1}{\delta} + n^{-\frac{1}{2}}\frac{Mi}{1+i\lambda}\sqrt{\log\frac{1}{\delta}}.$$

We observe that as $\lambda$ increases, the corresponding error decreases. This outcome is consistent with theoretical intuition, as augmenting the dataset with synthetic data effectively enlarges the mixed dataset. A larger dataset provides a more comprehensive representation of the mixed distribution, which in turn reduces the generalization error associated with this distribution. By incorporating more synthetic data, the mixed dataset better approximates the underlying mixed distribution, leading to improved generalization performance.

From the above discussion, we can conclude that the inclusion of synthetic data introduces a trade-off: on one hand, it increases the error from the cumulative distribution shift, while on the other, it reduces the generalization error on the mixed distribution. This trade-off has been explored theoretically in Fu et al. (2024b), though they primarily provided theoretical intuition. In contrast, our work explicitly decomposes the error into two terms, offering a deeper understanding of this trade-off and its implications for model performance in STLs. As for the optimal augmentation coefficient $\lambda^*$, it must satisfy the following condition:

$$\lambda^* = \inf_\lambda \Big\{ n^{-\frac{1}{4}}\log((1+i\lambda)n)M\log(1/\delta)$$
$$\lesssim n^{-1}\frac{\rho^2}{(1+i\lambda)^2}\log((1+i\lambda)n)i!\widetilde{B}_W^{(i+1)L}\log\frac{1}{\delta} + n^{-\frac{1}{2}}\frac{Mi}{1+i\lambda}\sqrt{\log\frac{1}{\delta}} \Big\}.$$

Unfortunately, obtaining a closed-form solution for $\lambda^*$ from this equation proves to be analytically intractable. However, we can derive the relationship between $\lambda^*$, the size of the real dataset $n$ from the above equation. Specifically, by omitting irrelevant constants and the $\log(1/\delta)$ term, we obtain that $\lambda^*$ should satisfy the following expression:

$$\frac{i!\widetilde{B}_W^{(i+1)L}}{n^{3/4}(1+i\lambda^*)^2} + \frac{i}{n^{1/4}(1+i\lambda^*)\log((1+i\lambda^*)n)} = \mathcal{O}(1).$$

We observe an important trend: the value of $\lambda^*$ increases as the size of the real dataset $n$ decreases. This aligns with theoretical intuition, as a smaller real dataset struggles to adequately represent the underlying distribution, leading to higher generalization error. Consequently, more synthetic data is required to control the generalization error of each generation on the mixed distribution. Conversely, when the real dataset is sufficiently large, the need for synthetic data augmentation diminishes.

## 6 CONCLUSION

As real-world data becomes increasingly scarce and existing datasets are progressively contaminated with synthetic content, STLs have emerged as a necessary strategy. STLs enable generative models to recursively train on a mix of real and synthetic data. However, empirical outcomes have varied significantly, revealing the need for a theoretical foundation to guide their successful application.

In this work, we introduced recursive stability as a key technical innovation and established the first generalization error bounds for STLs, which consider the impact of different model architectures. Our analysis demonstrated that preventing model collapse requires two critical conditions: maintaining a non-negligible proportion of real data and ensuring that models satisfy recursive stability. Furthermore, we were the first to extend this framework to transformers in in-context learning, showing that they also satisfy recursive stability and establish their generalization error bounds. Finally, we explored the trade-off introduced by synthetic data augmentation, balancing generalization improvement with potential distributional shifts. These contributions provide new insights into enhancing the stability and performance of generative models in STLs.

ACKNOWLEDGEMENT

This project is supported by the National Research Foundation, Singapore, under its NRF Professorship Award No. NRF-P2024-001.

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

# A   APPENDIX

## A.1   AUXILIARY DEFINITIONS

Below, we present some essential definitions.

**Definition 3.** (Lipschitz and Smoothness). Let constants $\kappa, \rho > 0$. Consider the function $\ell : \mathcal{W} \times \mathcal{Z} \to \mathbb{R}$. We define the following properties:

- **Lipschitz Continuity**: The loss $\ell$ is said to be $\rho$-Lipschitz continuous if $\|\ell(\boldsymbol{w}_1, \boldsymbol{z}) - \ell(\boldsymbol{w}_2, \boldsymbol{z})\| \leq \rho\|\boldsymbol{w}_1 - \boldsymbol{w}_2\|$ for any $\boldsymbol{w}_1, \boldsymbol{w}_2, \boldsymbol{z}$.

- **Smoothness**: The loss $\ell$ is said to be $\kappa$-Smooth if $\|\nabla_{\boldsymbol{w}}\ell(\boldsymbol{w}_1, \boldsymbol{z}) - \nabla_{\boldsymbol{w}}\ell(\boldsymbol{w}_2, \boldsymbol{z})\| \leq \kappa\|\boldsymbol{w}_1 - \boldsymbol{w}_2\|$ for any $\boldsymbol{w}_1, \boldsymbol{w}_2, \boldsymbol{z}$.

## A.2   EXPANSION TO GAUSSIAN MIXTURE MODELS

We adopt the setup from prior works Zheng et al. (2023) and consider a binary classification task where $Y = \{-1, 1\}$. Given a vector $\mu \in \mathbb{R}^d$ with $\|\mu\|_2 = 1$ and noise variance $\sigma^2 > 0$, the data distribution is specified as follows: $y \sim \text{uniform}\{-1, 1\}$ and $x \mid y \sim \mathcal{N}(y\mu, \sigma^2 I_d)$. We define the conditional generative model using parameters $\mu_y$ and $\sigma_k^2$, where $y \in \{-1, 1\}$ and $k \in [d]$. For $n$ data points, let $n_y$ represent the number of samples in class $y$. The parameters of the Gaussian mixture model are then learned as:

$$\hat{\mu}_y = \frac{\sum_{y_i=y} x_i}{n_y}, \quad \hat{\sigma}_k^2 = \sum_y \frac{n_y}{n} \frac{\sum_{y_i=y}(x_{ik} - \hat{\mu}_{yk})^2}{n_y - 1}.$$

Then we can generate new samples from the distribution: $y \sim \text{uniform}\{-1, 1\}$ and $x \mid y \sim \mathcal{N}(\hat{\mu}_y, \Sigma)$, where $\Sigma = \text{diag}(\sigma_1^2, \ldots, \sigma_d^2)$. Additionally, the learning algorithm functions as a linear classifier, parameterized by $\theta \in \mathbb{R}^d$, with predictions given by: $\hat{y} = \text{sign}(\theta^\top \mathbf{x})$. The loss function is defined as:

$$\ell(\theta, (x, y)) = \frac{1}{2\sigma^2}(x - y\theta)^\top(x - y\theta).$$

Thus, the output is $\hat{\theta} = \frac{1}{m}\sum_{i=1}^m y_i x_i$.

In this setting, we demonstrate recursive stability for the Gaussian mixture model as follows:

**Theorem 5.** *Let $S_0, S_0'$ denote two initial real datasets differing by a single example. Let $n$ represent the sample size of the mixed dataset $\tilde{S}_j$, where $\tilde{S}_j = \alpha S_0 + (1 - \alpha)S_j$ for $1 \leq j \leq i$. Choose $m = \mathcal{O}(\sqrt{n})$. Consider the previously described sampling and learning steps, where real data samples are drawn from the Gaussian Mixture Model distribution $\mathcal{D}$, and the synthetic data for the $i$-th generation is generated from the learned Gaussian Mixture distribution of the $i$-th generation. Then with probability at least $1 - \delta$, we have:*

$$\gamma_n^i \lesssim n^{-1/2}\alpha^{-1}(1 - (1 - \alpha)^i)\log(nd/\delta), \tag{3}$$

*where the measure for the recursive stability parameter is taken as the KL divergence.*

As $\alpha$ approaches 0, indicating that no real data is incorporated during each generation of training, we observe

$$\gamma_n^i \lesssim in^{-1/2}\log\frac{nd}{\delta},$$

which suggests a linear accumulation of errors. This finding aligns closely with the theoretical insights presented in Shumailov et al. (2024); Alemohammad et al. (2024a), where a Gaussian model

trained without real data demonstrated a linear divergence in variance. Thus, this underscores the validity of our theoretical results, confirming that the derived bound is meaningful and not vacuous.

Moreover, by leveraging the generalization error bound established in Theorem 1, we derive the following:

**Theorem 6.** *Consider the Gaussian Mixture Model in the setting outlined above. Let $n$ represent the sample size of the mixed dataset $\tilde{S}_j$, where $\tilde{S}_j = \alpha S_0 + (1 - \alpha) S_j$ for $1 \le j \le i$. Suppose the loss function is defined as $\ell(\theta, (\mathbf{x}, y)) = \frac{1}{2\sigma^2}(\mathbf{x} - y\theta)^\top(\mathbf{x} - y\theta)$. Let $\mathcal{A}(\tilde{S}_i)$ denote the output of applying the linear classifier described above to the mixed dataset $\tilde{S}_i$. Then, for any $\delta \in (0, 1)$, with probability at least $1 - \delta$, the following holds:*

$$\left| R_{\mathcal{D}_0}(\mathcal{A}(\widetilde{S}_i)) - \widehat{R}_{\widetilde{S}_i}(\mathcal{A}(\widetilde{S}_i)) \right| \lesssim n^{-1/2}(d + \log(n/\delta)) \log n \log(1/\delta)$$

$$+ n^{-1/4}(1 - (1 - \alpha)^i)\alpha^{-1}(d + \log(n/\delta))\sqrt{d \log(nd/\delta)}. \tag{4}$$

We observe that when $\alpha$ is set to a constant (e.g., $\alpha = 0.1$), the generalization error can be effectively controlled, preventing model collapse. This result aligns with the experimental findings in Alemohammad et al. (2024a) for Gaussian models.

## A.3 ADDITIONAL COMPARISON WITH RELATED WORK ON THEOREM 1

Dohmatob et al. (2024a) examined a linear regression setting, focusing solely on statistical approximation error without addressing the functional approximation error described in Shumailov et al. (2024). They did not consider incorporating real data to prevent collapse and demonstrated a linear dependency of degradation on the generation number in the case of fully synthetic data. Similarly, Alemohammad et al. (2024a) and Shumailov et al. (2024) provided theoretical insights using simple Gaussian models without incorporating real data, proving that the variance diverges linearly with the generation number. Seddik et al. (2024) explored a linear softmax classifier and, while also neglecting functional approximation error, demonstrated that adding real data can mitigate model collapse. Marchi et al. (2024) used asymptotic analysis to study parameter variance, assuming an infinite number of training generations and considering scenarios where the generative model is controlled via a "temperature" parameter. They proved that parameter variance is bounded under these conditions.

In contrast, our work addresses a much more complex and realistic scenario by introducing the novel concept of **recursive stability** and providing the **first** generalization analysis for STLs. Our analysis accounts for **statistical approximation error, functional approximation error, and optimization error** during the training of generative models. Unlike the settings explored in prior theoretical works, such as linear regression (Dohmatob et al., 2024a; Gerstgrasser et al., 2024), Gaussian models (Alemohammad et al., 2024a; Shumailov et al., 2024), or asymptotic assumptions (Marchi et al., 2024), our framework accommodates more complex generative model architectures, such as transformers. Specifically, we reveal how both **model architecture** and **the ratio** of real to synthetic data influence the success of STLs. For example, in Theorem 3, we demonstrate how our general generalization bound applies to transformer-based generative models, providing a theoretical framework that aligns with practical and more sophisticated use cases.

Additionally, while Marchi et al. (2024) assumed an **infinite number** of training generations for their asymptotic analysis, we consider **finite generations**, which is more practical since most experimental setups limit generations to fewer than 10 (as noted in Shumailov et al. (2024)). Moreover, our results confirm that when $\alpha = 0$ (i.e., no real data is used), the last term in our bound, representing the Cumulative Distribution Shift ($d_{\mathrm{TV}}(n)M(1 - (1 - \alpha)^i)\alpha^{-1}$), grows linearly. This finding aligns with the theoretical results of Dohmatob et al. (2024a); Alemohammad et al. (2024a); Shumailov et al. (2024); Fu et al. (2024b). Furthermore, we show that introducing even a constant proportion of real data significantly mitigates model collapse, aligning with experimental findings by Alemohammad et al. (2024a) and Bertrand et al. (2024).

## A.4 ADDITIONAL COMPARISON WITH RELATED WORK ON THEOREM 4

Gerstgrasser et al. (2024) also explored the use of accumulating data to prevent model collapse. They considered a simple linear regression setting without accounting for the dynamic process of

training generative models, focusing solely on statistical approximation error. They demonstrated that under the assumption of fixed synthetic data quality matching the original real data, statistical approximation error can be controlled.

By contrast, our work addresses a much more complex and realistic scenario, incorporating the dynamic behavior of transformer-based generative models, learning algorithms, and both statistical and functional approximation errors. Additionally, we allow for dynamic regulation of synthetic data size via a $\lambda$ coefficient, enabling us to identify the optimal synthetic dataset size for avoiding model collapse in these more challenging settings.

## A.5 AUXILIARY LEMMAS

In this section, we begin by introducing a set of auxiliary theorems that will be utilized in the subsequent proofs.

**Lemma 7** (McDiarmid's Inequality). *Consider independent random variables $Z_1, \cdots, Z_n \in \mathcal{Z}$ and a mapping $\phi : \mathcal{Z}^n \to \mathbb{R}$. If, for all $i \in \{1, \cdots, n\}$, and for all $z_1, \cdots, z_n, z_i' \in \mathcal{Z}$, the function $\phi$ satisfies*

$$|\phi(z_1, \cdots, z_{i-1}, z_i, z_{i+1}, \cdots, z_n) - \phi(z_1, \cdots, z_{i-1}, z_i', z_{i+1}, \cdots, z_n)| \leq c,$$

*then,*

$$P(|\phi(Z_1, \cdots, Z_n) - \mathbb{E}\phi(Z_1, \ldots, Z_n) \geq t|) \leq 2\exp\left(\frac{-2t^2}{nc^2}\right).$$

*Furthermore, for any $p \geq 2$,*

$$\|\phi(Z_1, \ldots, Z_n) - \mathbb{E}[\phi(Z_1, \ldots, Z_n)]\|_p \leq 2\sqrt{np}c.$$

**Lemma 8.** *((Bousquet et al., 2020)). Let $\boldsymbol{z} = (Z_1, \ldots, Z_n)$ be a vector of independent random variables each taking values in $\mathcal{Z}$, and let $g_1, \ldots, g_n$ be some functions $g_i : \mathcal{Z}^n \to \mathbb{R}$ such that the following holds for any $i \in [n]$ :*

- $|\mathbb{E}[g_i(\boldsymbol{z}) \mid Z_i]| \leq M$,

- $\mathbb{E}[g_i(\boldsymbol{z}) \mid \boldsymbol{z}^{\setminus i}] = 0$,

- *$g_i$ has a bounded difference $\beta$ with respect to all variables except the $i$-th variable, that is, for all $j \neq i, \boldsymbol{z} = (Z_1, \ldots, Z_n)$ and $\boldsymbol{z}^j = (Z_1, \ldots, Z_j', \ldots, Z_n) \in \mathbb{R}^n$, we have $|g_i(\boldsymbol{z}) - g_i(\boldsymbol{z}^j)| \leq \beta$.*

*Then, for any $p \geq 2$,*

$$\left\|\sum_{i=1}^n g_i(\boldsymbol{z})\right\|_p \leq 12\sqrt{2}pn\beta\log n + 4M\sqrt{pn}.$$

**Lemma 9.** *If $\|Y\|_p \leq \sqrt{p}a + pb$ for any $p \geq 1$, then for any $\delta \in (0, 1)$, with probability at least $1 - \delta$,*

$$|Y| \leq e\left(a\sqrt{\log\left(\frac{e}{\delta}\right)} + b\log\left(\frac{e}{\delta}\right)\right).$$

In addition, we introduce the definition of the Total Variation (TV) distance as follows:

**Definition 4** (Total Variation Distance). Given two probability distributions $p$ and $q$ over a multidimensional space $\mathbb{R}^d$, the Total Variation Distance between $p$ and $q$ is:

$$TV(p, q) = \frac{1}{2}\int_{\mathbb{R}^d} |p(\boldsymbol{z}) - q(\boldsymbol{z})|\, d\boldsymbol{z}.$$

A.6 PROOF OF THEOREM 1

In this Section, we prove Theorem 1 by first decomposing the generalization error into two components: the *Cumulative Distribution Shift Across Generations* and the *Generalization Error on Mixed Distributions*. We then proceed to bound the *Cumulative Distribution Shift Across Generations* by leveraging the properties of the generative model and recursive techniques. For the *Generalization Error on Mixed Distributions*, we follow the framework of Zheng et al. (2023), leveraging the fact that within the mixed dataset $\widetilde{S}_i$, the set $S_i$ satisfies the conditional i.i.d. assumption when $S_0$ is fixed. Combined with moment bounds, this allows us to effectively bound the generalization error.

The main proof is as follows:

*Proof of Theorem 1.* We begin by decomposing the generalization error as follows:

$$\left| R_{\mathcal{D}_0}(\mathcal{A}(\widetilde{S}_i)) - \widehat{R}_{\widetilde{S}_i}(\mathcal{A}(\widetilde{S}_i)) \right| \leq \underbrace{\left| R_{\mathcal{D}_0}(\mathcal{A}(\widetilde{S}_i)) - R_{\widetilde{\mathcal{D}}_i}(\mathcal{A}(\widetilde{S}_i)) \right|}_{\text{Cumulative distribution shift across generations}} + \underbrace{\left| R_{\widetilde{\mathcal{D}}_i}(\mathcal{A}(\widetilde{S}_i)) - \widehat{R}_{\widetilde{S}_i}(\mathcal{A}(\widetilde{S}_i)) \right|}_{\text{Generalization error on mixed distributions}}.$$

**Upper Bounding Cumulative Distribution Shift Term**

For the term $\left| R_{\mathcal{D}_0}(\mathcal{A}(\widetilde{S}_i)) - R_{\widetilde{\mathcal{D}}_i}(\mathcal{A}(\widetilde{S}_i)) \right|$, we first note that $\widetilde{\mathcal{D}}_i = \alpha \mathcal{D}_0 + (1-\alpha)\mathcal{D}_i$. Therefore, we obtain:

$$\left| R_{\mathcal{D}_0}(\mathcal{A}(\widetilde{S}_i)) - R_{\widetilde{\mathcal{D}}_i}(\mathcal{A}(\widetilde{S}_i)) \right|$$
$$= \left| R_{\mathcal{D}_0}(\mathcal{A}(\widetilde{S}_i)) - \alpha R_{\mathcal{D}_0}(\mathcal{A}(\widetilde{S}_i) - (1-\alpha)R_{\mathcal{D}_i}(\mathcal{A}(\widetilde{S}_i)) \right|$$
$$= (1-\alpha) \left| R_{\mathcal{D}_0}(\mathcal{A}(\widetilde{S}_i) - R_{\mathcal{D}_i}(\mathcal{A}(\widetilde{S}_i)) \right|. \tag{5}$$

Furthermore, we can further decompose it as follows:

$$\left| R_{\mathcal{D}_0}(\mathcal{A}(\widetilde{S}_i) - R_{\mathcal{D}_i}(\mathcal{A}(\widetilde{S}_i)) \right| \leq \left| R_{\mathcal{D}_0}(\mathcal{A}(\widetilde{S}_i)) - R_{\widetilde{\mathcal{D}}_{i-1}}(\mathcal{A}(\widetilde{S}_i)) \right| + \left| R_{\widetilde{\mathcal{D}}_{i-1}}(\mathcal{A}(\widetilde{S}_i)) - R_{\mathcal{D}_i}(\mathcal{A}(\widetilde{S}_i)) \right|. \tag{6}$$

By substituting inequality 6 into inequality 5, we obtain:

$$\left| R_{\mathcal{D}_0}(\mathcal{A}(\widetilde{S}_i)) - R_{\widetilde{\mathcal{D}}_i}(\mathcal{A}(\widetilde{S}_i)) \right|$$
$$\leq (1-\alpha) \left| R_{\mathcal{D}_0}(\mathcal{A}(\widetilde{S}_i)) - R_{\widetilde{\mathcal{D}}_{i-1}}(\mathcal{A}(\widetilde{S}_i)) \right| + (1-\alpha) \left| R_{\widetilde{\mathcal{D}}_{i-1}}(\mathcal{A}(\widetilde{S}_i)) - R_{\mathcal{D}_i}(\mathcal{A}(\widetilde{S}_i)) \right|. \tag{7}$$

Then, for the term $|R_{\widetilde{\mathcal{D}}_{i-1}}(\mathcal{A}(\widetilde{S}_i)) - R_{\mathcal{D}_i}(\mathcal{A}(\widetilde{S}_i))|$, we have:

$$\left| R_{\widetilde{\mathcal{D}}_{i-1}}(\mathcal{A}(\widetilde{S}_i)) - R_{\mathcal{D}_i}(\mathcal{A}(\widetilde{S}_i)) \right| = \left| \int_{\boldsymbol{z}} \ell(\mathcal{A}(\widetilde{S}_i), \boldsymbol{z}) \left( \mathbb{P}_{\widetilde{\mathcal{D}}_{i-1}}(\boldsymbol{z}) - \mathbb{P}_{\mathcal{D}_i}(\boldsymbol{z}) \right) d\boldsymbol{z} \right|$$
$$\leq \int_{\boldsymbol{z}} \left| \ell(\mathcal{A}(\widetilde{S}), \boldsymbol{z}) \left( \mathbb{P}_{\widetilde{\mathcal{D}}_{i-1}}(\boldsymbol{z}) - \mathbb{P}_{\mathcal{D}_i}(\boldsymbol{z}) \right) \right| d\boldsymbol{z}$$
$$\leq M \int_{\boldsymbol{z}} \left| \mathbb{P}_{\widetilde{\mathcal{D}}_{i-1}}(\boldsymbol{z}) - \mathbb{P}_{\mathcal{D}_i}(\boldsymbol{z}) \right| d\boldsymbol{z}$$
$$= 2M TV \left( \widetilde{\mathcal{D}}_{i-1}, \mathcal{D}_i \right). \tag{8}$$

Incorporating inequality 8 into inequality 7, we arrive at:

$$|R_{\mathcal{D}_0}(\mathcal{A}(\widetilde{S}_i)) - R_{\widetilde{\mathcal{D}}_i}(\mathcal{A}(\widetilde{S}_i))|$$
$$\leq (1-\alpha)|R_{\mathcal{D}_0}(\mathcal{A}(\widetilde{S}_i)) - R_{\widetilde{\mathcal{D}}_{i-1}}(\mathcal{A}(\widetilde{S}_i))| + 2(1-\alpha)M TV \left( \widetilde{\mathcal{D}}_{i-1}, \mathcal{D}_i \right). \tag{9}$$

Next, we apply recursive techniques to address the problem further. First, we obtain

$$|R_{\mathcal{D}_0}(\mathcal{A}(\widetilde{S}_i)) - R_{\widetilde{\mathcal{D}}_{i-1}}(\mathcal{A}(\widetilde{S}_i))|$$
$$\leq (1-\alpha)|R_{\mathcal{D}_0}(\mathcal{A}(\widetilde{S}_i)) - R_{\widetilde{\mathcal{D}}_{i-2}}(\mathcal{A}(\widetilde{S}_i))| + 2(1-\alpha)M TV \left( \widetilde{\mathcal{D}}_{i-2}, \mathcal{D}_{i-1} \right). \tag{10}$$

Plugging inequality 10 into inequality 9into the inequality, we obtain that:

$$|R_{\mathcal{D}_0}(\mathcal{A}(\widetilde{S}_i)) - R_{\widetilde{\mathcal{D}}_i}(\mathcal{A}(\widetilde{S}_i))|$$

$$\leq (1-\alpha)^2 |R_{\mathcal{D}_0}(\mathcal{A}(\widetilde{S}_i)) - R_{\widetilde{\mathcal{D}}_{i-2}}(\mathcal{A}(\widetilde{S}_i))| + 2(1-\alpha)^2 MTV\left(\widetilde{\mathcal{D}}_{i-2}, \mathcal{D}_{i-1}\right) + 2(1-\alpha)MTV\left(\widetilde{\mathcal{D}}_{i-1}, \mathcal{D}_i\right).$$

By recursion, we obtain:

$$|R_{\mathcal{D}_0}(\mathcal{A}(\widetilde{S}_i)) - R_{\widetilde{\mathcal{D}}_i}(\mathcal{A}(\widetilde{S}_i))|$$

$$\leq (1-\alpha)^{i-1}|R_{\mathcal{D}_0}(\mathcal{A}(\widetilde{S}_i)) - R_{\widetilde{\mathcal{D}}_1}(\mathcal{A}(\widetilde{S}_i))| + 2(1-\alpha)^{i-1}MTV\left(\widetilde{\mathcal{D}}_1, \mathcal{D}_2\right) + ... + 2(1-\alpha)MTV\left(\widetilde{\mathcal{D}}_{i-1}, \mathcal{D}_i\right)$$

$$\leq 2(1-\alpha)^i MTV\left(\mathcal{D}_0, \mathcal{D}_1\right) + 2(1-\alpha)^{i-1}MTV\left(\widetilde{\mathcal{D}}_1, \mathcal{D}_2\right) + ... + 2(1-\alpha)MTV\left(\widetilde{\mathcal{D}}_{i-1}, \mathcal{D}_i\right).$$

Let $n_0$ represent the sample size of the real dataset $S_0$, and let $n_i$ denote the sample size of the mixed dataset $\widetilde{S}_i$ in the $i$-th generation. Thus, $TV\left(\widetilde{\mathcal{D}}_j, \mathcal{D}_{j+1}\right)$ can be written as a function of $n_j$. Assuming that the sample size for each generation's dataset is identical, i.e., $n_0 = n_1 = \cdots = n_i = n$, and that the TV distance for each generation is of the same order, denoted by $d_{\text{TV}}(n)$, we can derive the following result:

$$|R_{\mathcal{D}_0}(\mathcal{A}(\widetilde{S}_i)) - R_{\widetilde{\mathcal{D}}_i}(\mathcal{A}(\widetilde{S}_i))| \leq 2M d_{\text{TV}}(n)\left[(1-\alpha)^i + (1-\alpha)^{i-1} + ... + (1-\alpha)\right]$$

$$= 2M\left(1 - (1-\alpha)^i\right)\alpha^{-1}d_{\text{TV}}(n). \tag{11}$$

Then we obtain:

$$|R_{\mathcal{D}_0}(\mathcal{A}(\widetilde{S}_i)) - \widehat{R}_{\widetilde{S}_i}(\mathcal{A}(\widetilde{S}_i))| \leq |R_{\mathcal{D}_0}(\mathcal{A}(\widetilde{S}_i)) - R_{\widetilde{\mathcal{D}}_i}(\mathcal{A}(\widetilde{S}_i))| + |R_{\widetilde{\mathcal{D}}_i}(\mathcal{A}(\widetilde{S}_i)) - \widehat{R}_{\widetilde{S}_i}(\mathcal{A}(\widetilde{S}_i))|$$

$$\leq 2M\left(1 - (1-\alpha)^i\right)\alpha^{-1}d_{\text{TV}}(n) + |R_{\widetilde{\mathcal{D}}_i}(\mathcal{A}(\widetilde{S}_i)) - \widehat{R}_{\widetilde{S}_i}(\mathcal{A}(\widetilde{S}_i))|. \tag{12}$$

**Upper Bounding Generalization Error on Mixed Distributions Term**

Next, we turn our attention to the term $|R_{\widetilde{\mathcal{D}}_i}(\mathcal{A}(\widetilde{S}_i)) - \widehat{R}_{\widetilde{S}_i}(\mathcal{A}(\widetilde{S}_i))|$. Our primary objective is to establish a moment bound for this expression.

$$\left\|R_{\widetilde{\mathcal{D}}_i}(\mathcal{A}(\widetilde{S}_i)) - \widehat{R}_{\widetilde{S}_i}(\mathcal{A}(\widetilde{S}_i))\right\|_p$$

$$= \left\|\alpha R_{\mathcal{D}_0}(\mathcal{A}(\widetilde{S}_i)) + (1-\alpha)R_{\mathcal{D}_i}(\mathcal{A}(\widetilde{S}_i)) - \frac{1}{n}\sum_{\boldsymbol{z}_i \in S_{0,\alpha}} \ell(\mathcal{A}(\widetilde{S}_i), \boldsymbol{z}_i) - \frac{1}{n}\sum_{\boldsymbol{z}_i \in S_{i,1-\alpha}} \ell(\mathcal{A}(\widetilde{S}_i), \boldsymbol{z}_i)\right\|_p$$

$$\leq \underbrace{\left\|\alpha R_{\mathcal{D}_0}(\mathcal{A}(\widetilde{S}_i)) - \frac{1}{n}\sum_{\boldsymbol{z}_i \in S_{0,\alpha}} \ell(\mathcal{A}(\widetilde{S}_i), \boldsymbol{z}_i)\right\|_p}_{\text{Term 1}} + \underbrace{\left\|(1-\alpha)R_{\mathcal{D}_i}(\mathcal{A}(\widetilde{S}_i)) - \frac{1}{n}\sum_{\boldsymbol{z}_i \in S_{i,1-\alpha}} \ell(\mathcal{A}(\widetilde{S}_i), \boldsymbol{z}_i)\right\|_p}_{\text{Term 2}}.$$

$$\tag{13}$$

The newly sampled dataset, denoted as $S_{0,\alpha}$, is a subset of the original dataset $S_0$, where $S_{0,\alpha} \subseteq S_0$ and its size is $\alpha \times |S_0|$. Specifically, $S_{0,\alpha}$ contains a proportion $\alpha$ of the $n$ data points in $S_0$, resulting in a total of $n \times \alpha$ data points. Similarly, $S_{i,1-\alpha}$ is a subset of the synthetic dataset $S_i$, where $S_{i,1-\alpha} \subseteq S_i$, and its size is $(1-\alpha) \times |S_i|$. Specifically, $S_{i,1-\alpha}$ contains a proportion $1-\alpha$ of the $n$ data points in $S_i$, resulting in $n \times (1-\alpha)$ data points.

We observe that for any function $f(S)$, if there exists a bound $\|f\|_p(S_j) \leq C$ for some subset $S_j \subseteq S$, then we have the following:

$$\|f\|_p = (\mathbb{E}\mathbb{E}\left[|f|^p \mid S_j\right])^{1/p} \leq (\mathbb{E}\left[C^p\right])^{1/p} \leq C.$$

Fix $S_0$, then data in $S_i$ are independent. We use this property and Lemma 8 to bound the Term 2. We introduce functions $f_j(S_{i,1-\alpha})$ which play the same role as $g_j$'s in Lemma 8 as

$$f_j(S_{i,1-\alpha}) = \mathbb{E}_{\boldsymbol{z}'_{i,j} \sim \mathcal{D}_i}\left[\mathbb{E}_{\boldsymbol{z} \sim \mathcal{D}_i}\ell(\mathcal{A}(S_{0,\alpha} \cup S^j_{i,1-\alpha}), \boldsymbol{z}) - \ell(\mathcal{A}(S_{0,\alpha} \cup S^j_{i,1-\alpha}), \boldsymbol{z}_{i,j})\right],$$

where $z_{i,j}$ is the $j$-th data in $S_{i,1-\alpha}$, and $S_{i,1-\alpha}^j$ obtained by replacing $z_{i,j}$ by $z'_{i,j}$. Next, we prove that $f_j$ satisfies the three conditions outlined in Lemma 8. First, we demonstrate condition $|f_j| \leq M$.

$$
\begin{aligned}
|f_j| &= \left| \mathbb{E}_{z'_{i,j} \sim \mathcal{D}_i} \left[ \mathbb{E}_{z \sim \mathcal{D}_i} \ell(\mathcal{A}(S_{0,\alpha} \cup S_{i,1-\alpha}^j), z) - \ell(\mathcal{A}(S_{0,\alpha} \cup S_{i,1-\alpha}^j), z_{i,j}) \right] \right| \\
&\leq \mathbb{E}_{z'_{i,j} \sim \mathcal{D}_i} \mathbb{E}_{z \sim \mathcal{D}_i} \left| \ell(\mathcal{A}(S_{0,\alpha} \cup S_{i,1-\alpha}^j), z) - \ell(\mathcal{A}(S_{0,\alpha} \cup S_{i,1-\alpha}^j), z_{i,j}) \right|. \\
&\leq M
\end{aligned}
$$

We then continue by proving conditions $\mathbb{E}[f_j | S_{i,1-\alpha}^{\backslash j}] = 0$:

$$
\begin{aligned}
&\mathbb{E}\left[ f_j \mid S_{i,1-\alpha}^{\backslash j} \right] \\
&= \mathbb{E}_{z_{i,j} \sim \mathcal{D}_i} \left[ \mathbb{E}_{z'_{i,j} \sim \mathcal{D}_i} \left[ \mathbb{E}_{z \sim \mathcal{D}_i} \ell(\mathcal{A}(S_{0,\alpha} \cup S_{i,1-\alpha}^j), z) - \ell(\mathcal{A}(S_{0,\alpha} \cup S_{i,1-\alpha}^j), z_{i,j}) \right] \mid S_{i,1-\alpha}^{\backslash j} \right] \\
&= \mathbb{E}_{z'_{i,j} \sim \mathcal{D}_i} \left[ \left[ \mathbb{E}_{z \sim \mathcal{D}_i} \ell\left( \mathcal{A}\left( S_{0,\alpha} \cup S_{i,1-\alpha}^j \right), z \right) - \mathbb{E}_{z_{i,j} \sim \mathcal{D}_i} \ell\left( \mathcal{A}\left( S_{0,\alpha} \cup S_{i,1-\alpha}^j \right), z_{i,j} \right) \right] \mid S_{i,1-\alpha}^{\backslash j} \right] \\
&= \mathbb{E}_{z'_{i,j} \sim \mathcal{D}_i} \left[ 0 \mid S_{i,1-\alpha}^{\backslash j} \right] = 0.
\end{aligned}
$$

Finally, we prove that $f_j$ has a bounded difference $2\beta_n$ with respect to all variables except the $j$-th variable. Let $t \neq j$, then we obtain:

$$
\begin{aligned}
&|f_j\left(S_{i,1-\alpha}\right) - f_j\left(S_{i,1-\alpha}^t\right)| \\
&= |\mathbb{E}_{z'_{i,j} \sim \mathcal{D}_i} \left[ \mathbb{E}_{z \sim \mathcal{D}_i} \ell(\mathcal{A}(S_{0,\alpha} \cup S_{i,1-\alpha}^j), z) - \ell(\mathcal{A}(S_{0,\alpha} \cup S_{i,1-\alpha}^j), z_{i,j}) \right] \\
&\quad - \mathbb{E}_{z'_{i,j} \sim \mathcal{D}_i} \left[ \mathbb{E}_{z \sim \mathcal{D}_i} \ell(\mathcal{A}(S_{0,\alpha} \cup (S_{i,1-\alpha}^t)^j), z) - \ell(\mathcal{A}(S_{0,\alpha} \cup (S_{i,1-\alpha}^t)^j), z_{i,j}) \right] | \\
&\leq \left| \mathbb{E}_{z'_{i,j} \sim \mathcal{D}_i} \mathbb{E}_{z \sim \mathcal{D}_i} \left[ \ell(\mathcal{A}(S_{0,\alpha} \cup S_{i,1-\alpha}^j), z) - \ell(\mathcal{A}(S_{0,\alpha} \cup (S_{i,1-\alpha}^t)^j), z) \right] \right| \\
&\quad + \left| \mathbb{E}_{z'_{i,j} \sim \mathcal{D}_i} \left[ \ell(\mathcal{A}(S_{0,\alpha} \cup S_{i,1-\alpha}^j), z_{i,j}) - \ell(\mathcal{A}(S_{0,\alpha} \cup (S_{i,1-\alpha}^t)^j), z_{i,j}) \right] \right| \\
&\leq \beta_n + \beta_n = 2\beta_n.
\end{aligned}
$$

Therefore, for any fixed $S_0$, by Lemma 8, for any $p \geq 2$, we have

$$
\left\| \sum_{j=1}^{n(1-\alpha)} f_j\left(S_{i,1-\alpha}\right) \right\|_p \lesssim pn(1-\alpha)\beta_n \log(n(1-\alpha)) + M\sqrt{pn(1-\alpha)}. \tag{14}
$$

We note that the difference between Term 2 and $\sum_{j=1}^{n(1-\alpha)} f_j$ is minimal. Consequently, for any fixed $S_0$, we can bound Term 2 using inequality 14 as follows.

$$
\left\| (1-\alpha)R_{\mathcal{D}_i}(\mathcal{A}(\widetilde{S}_i)) - \frac{1}{n}\sum_{\boldsymbol{z}_i \in S_{i,1-\alpha}} \ell(\mathcal{A}(\widetilde{S}_i), \boldsymbol{z}_i) \right\|_p
$$

$$
= \left\| (1-\alpha)R_{\mathcal{D}_i}(\mathcal{A}(S_{0,\alpha} \cup S_{i,1-\alpha})) - \frac{1}{n}\sum_{j=1}^{n(1-\alpha)} \ell(\mathcal{A}(S_{0,\alpha} \cup S_{i,1-\alpha}), \boldsymbol{z}_{i,j}) \right\|_p \qquad \text{Fix } S_{0,\alpha}
$$

$$
= \left\| \frac{1}{n}\sum_{j=1}^{n(1-\alpha)} \left( \mathbb{E}_{\boldsymbol{z}\sim\mathcal{D}_i} \ell\left(\mathcal{A}\left(S_{0,\alpha} \cup S_{i,1-\alpha}\right), \boldsymbol{z}\right) - \ell\left(\mathcal{A}\left(S_{0,\alpha} \cup S_{i,1-\alpha}\right), \boldsymbol{z}_{i,j}\right) \right) \right\|_p
$$

$$
\leq \frac{1}{n} \left\| \sum_{j=1}^{n(1-\alpha)} \left( \mathbb{E}_{\boldsymbol{z}'_{i,j}\sim\mathcal{D}_i} \left[ \mathbb{E}_{\boldsymbol{z}\sim\mathcal{D}_i} \ell(\mathcal{A}(S_{0,\alpha} \cup S^j_{i,1-\alpha}), \boldsymbol{z}) - \ell(\mathcal{A}(S_{0,\alpha} \cup S^j_{i,1-\alpha}), \boldsymbol{z}_{i,j}) \right] \right) \right\|_p + (1-\alpha)\left\| 2\beta_n \right\|_p
$$

$$
= \frac{1}{n} \left\| \sum_{j=1}^{n(1-\alpha)} f_j\left(S_{i,1-\alpha}\right) \right\|_p + (1-\alpha)\left\| 2\beta_n \right\|_p
$$

$$
\lesssim p(1-\alpha)\beta_n \log(n(1-\alpha)) + M\sqrt{\frac{p(1-\alpha)}{n}} + 2(1-\alpha)\beta_n
$$

$$
\lesssim p(1-\alpha)\beta_n \log(n(1-\alpha)) + M\sqrt{\frac{p(1-\alpha)}{n}}.
$$

Next, for Term 2, we derive the following result:

$$
\left\| (1-\alpha)R_{\mathcal{D}_i}(\mathcal{A}(\widetilde{S}_i)) - \frac{1}{n}\sum_{\boldsymbol{z}_i \in S_{i,1-\alpha}} \ell(\mathcal{A}(\widetilde{S}_i), \boldsymbol{z}_i) \right\|_p \lesssim p(1-\alpha)\beta_n \log(n(1-\alpha)) + M\sqrt{\frac{p(1-\alpha)}{n}}.
$$
(15)

Now, we use a similar idea to bound Term 1 $\left\| \alpha R_{\mathcal{D}_0}(\mathcal{A}(\widetilde{S}_i)) - \frac{1}{n}\sum_{\boldsymbol{z}_i \in S_{0,\alpha}} \ell(\mathcal{A}(\widetilde{S}_i), \boldsymbol{z}_i) \right\|_p$. We decompose it as the following.

$$
\left\| \alpha R_{\mathcal{D}_0}(\mathcal{A}(\widetilde{S}_i)) - \frac{1}{n}\sum_{\boldsymbol{z}_i \in S_{0,\alpha}} \ell(\mathcal{A}(\widetilde{S}_i), \boldsymbol{z}_i) \right\|_p
$$

$$
\leq \underbrace{\left\| (\alpha R_{\mathcal{D}_0}(\mathcal{A}(\widetilde{S}_i)) - \frac{1}{n}\sum_{\boldsymbol{z}_i \in S_{0,\alpha}} \ell(\mathcal{A}(\widetilde{S}_i), \boldsymbol{z}_i)) - \mathbb{E}_{S_{i,1-\alpha}\sim\mathcal{D}_i^{n(1-\alpha)}}(\alpha R_{\mathcal{D}_0}(\mathcal{A}(\widetilde{S}_i)) - \frac{1}{n}\sum_{\boldsymbol{z}_i \in S_{0,\alpha}} \ell(\mathcal{A}(\widetilde{S}_i), \boldsymbol{z}_i)) \right\|_p}_{\text{Term 3}}
$$

$$
+ \underbrace{\left\| \mathbb{E}_{S_{i,1-\alpha}\sim\mathcal{D}_i^{n(1-\alpha)}}(\alpha R_{\mathcal{D}_0}(\mathcal{A}(\widetilde{S}_i)) - \frac{1}{n}\sum_{\boldsymbol{z}_i \in S_{0,\alpha}} \ell(\mathcal{A}(\widetilde{S}_i), \boldsymbol{z}_i)) \right\|}_{\text{Term 4}}.
$$
(16)

We proceed by bounding each term. Specifically, Term 3 can be bounded using McDiarmid's inequality, as outlined in Lemma 7, and Term 4 can be bounded by applying Lemma 8.

To bound Term 3, we begin by fixing $S_{0,\alpha}$ and utilizing the conditional independence property of $S_i$ once again. In order to apply Lemma 8, we must show that $\alpha R_{\mathcal{D}_0}(\mathcal{A}(\widetilde{S}_i)) - \frac{1}{n}\sum_{\boldsymbol{z}_i \in S_{0,\alpha}} \ell(\mathcal{A}(\widetilde{S}_i), \boldsymbol{z}_i)$ exhibits a bounded difference with respect to $S_{i,1-\alpha}$ when $S_{0,\alpha}$ is fixed. This expression can be

formulated as follows.

$$\left| \alpha R_{\mathcal{D}_0}(\mathcal{A}(S_{0,\alpha} \cup S_{i,1-\alpha})) - \frac{1}{n} \sum_{\boldsymbol{z}_i \in S_{0,\alpha}} \ell(\mathcal{A}(S_{0,\alpha} \cup S_{i,1-\alpha}), \boldsymbol{z}_i) \right.$$

$$\left. - \alpha R_{\mathcal{D}_0}(\mathcal{A}(S_{0,\alpha} \cup S_{i,1-\alpha}^j)) + \frac{1}{n} \sum_{\boldsymbol{z}_i \in S_{0,\alpha}} \ell(\mathcal{A}(S_{0,\alpha} \cup S_{i,1-\alpha}^j), \boldsymbol{z}_i) \right|$$

$$\leq \alpha \left| R_{\mathcal{D}_0}(\mathcal{A}(S_{0,\alpha} \cup S_{i,1-\alpha})) - R_{\mathcal{D}_0}(\mathcal{A}(S_{0,\alpha} \cup S_{i,1-\alpha}^j)) \right|$$

$$+ \frac{1}{n} \left| \sum_{\boldsymbol{z}_i \in S_{0,\alpha}} \ell(\mathcal{A}(S_{0,\alpha} \cup S_{i,1-\alpha}), \boldsymbol{z}_i) - \sum_{\boldsymbol{z}_i \in S_{0,\alpha}} \ell\left(\mathcal{A}\left(S_{0,\alpha} \cup S_{i,1-\alpha}^j\right), \boldsymbol{z}_i\right) \right|$$

$$\leq \alpha \beta_n + \alpha \beta_n = 2\alpha \beta_n.$$

Thus, by Mcdiarmid Inequality, we have

$$\left\| (\alpha R_{\mathcal{D}_0}(\mathcal{A}(\widetilde{S}_i)) - \frac{1}{n} \sum_{\boldsymbol{z}_i \in S_{0,\alpha}} \ell(\mathcal{A}(\widetilde{S}_i), \boldsymbol{z}_i)) - \mathbb{E}_{S_{i,1-\alpha} \sim \mathcal{D}_i^{n(1-\alpha)}} (\alpha R_{\mathcal{D}_0}(\mathcal{A}(\widetilde{S}_i)) - \frac{1}{n} \sum_{\boldsymbol{z}_i \in S_{0,\alpha}} \ell(\mathcal{A}(\widetilde{S}_i), \boldsymbol{z}_i)) \right\|_p$$

$$\leq 4\sqrt{n(1-\alpha)p} \alpha \beta_n \lesssim \sqrt{n(1-\alpha)p} \alpha \beta_n. \tag{17}$$

We now introduce a set of functions and apply Lemma 8 once more to bound Term 4. Specifically, we define $h_j(S)$, which serves a similar role to the $g_i$'s in Lemma 8, as follows:

$$h_j(S_{0,\alpha})$$
$$= \mathbb{E}_{\boldsymbol{z}_{0,j}' \sim \mathcal{D}_0} \mathbb{E}_{S_{i,1-\alpha} \sim \mathcal{D}_i^{n(1-\alpha)}(S_{0,\alpha}^j)} \left[ \mathbb{E}_{\boldsymbol{z} \sim \mathcal{D}_0} \ell\left(\mathcal{A}\left(S_{0,\alpha}^j \cup S_{i,1-\alpha}\right), \boldsymbol{z}\right) - \ell\left(\mathcal{A}\left(S_{0,\alpha}^j \cup S_{i,1-\alpha}\right), \boldsymbol{z}_{0,j}\right) \right] \tag{18}$$

where $\boldsymbol{z}_{0,j}$ denote the $j$-th data point in $S_{0,\alpha}$, and $S_{0,\alpha}^j$ represent the dataset obtained by replacing $\boldsymbol{z}_{0,j}$ with $\boldsymbol{z}_{0,j}'$. Additionally, it is important to note that $S_{i,1-\alpha} \sim \mathcal{D}_i^{n(1-\alpha)}\left(S_{0,\alpha}^j\right)$ indicates that $S_{i,1-\alpha}$ is the synthetic dataset generated after the self-consuming loop, following $i$-generations, and obtained by modifying a single data point from the initial real dataset $S_0$. This complex scenario can be addressed using the recursive stability we have defined for the self-consuming loop in Definition 2. Moreover, similar to the process above, we observe that $|h_j| \leq M$ and $\mathbb{E}\left[h_j \mid S_{0,\alpha}^{\setminus j}\right] = 0$. More intricately, we will now prove that $h_j$ exhibits a bounded difference. This will be demonstrated as follows.

$$|h_j(S_{0,\alpha}) - h_j\left(S_{0,\alpha}^t\right)|$$
$$= | \mathbb{E}_{\boldsymbol{z}_{0,j}' \sim \mathcal{D}_0} \mathbb{E}_{S_{i,1-\alpha} \sim \mathcal{D}_i^{n(1-\alpha)}(S_{0,\alpha}^j)} \left[ \mathbb{E}_{\boldsymbol{z} \sim \mathcal{D}_0} \ell\left(\mathcal{A}\left(S_{0,\alpha}^j \cup S_{i,1-\alpha}\right), \boldsymbol{z}\right) - \ell\left(\mathcal{A}\left(S_{0,\alpha}^j \cup S_{i,1-\alpha}\right), \boldsymbol{z}_{0,j}\right) \right]$$
$$- \mathbb{E}_{\boldsymbol{z}_{0,j}' \sim \mathcal{D}_0} \mathbb{E}_{S_{i,1-\alpha} \sim \mathcal{D}_i^{n(1-\alpha)}((S_{0,\alpha}^t)^j)} \left[ \mathbb{E}_{\boldsymbol{z} \sim \mathcal{D}_0} \ell\left(\mathcal{A}\left((S_{0,\alpha}^t)^j \cup S_{i,1-\alpha}\right), \boldsymbol{z}\right) - \ell\left(\mathcal{A}\left((S_{0,\alpha}^t)^j \cup S_{i,1-\alpha}\right), \boldsymbol{z}_{0,j}\right) \right]$$
$$\leq | \mathbb{E}_{\boldsymbol{z}_{0,j}' \sim \mathcal{D}_0} \mathbb{E}_{S_{i,1-\alpha} \sim \mathcal{D}_i^{n(1-\alpha)}(S_{0,\alpha}^j)} \left[ \mathbb{E}_{\boldsymbol{z} \sim \mathcal{D}_0} \ell\left(\mathcal{A}\left(S_{0,\alpha}^j \cup S_{i,1-\alpha}\right), \boldsymbol{z}\right) - \ell\left(\mathcal{A}\left(S_{0,\alpha}^j \cup S_{i,1-\alpha}\right), \boldsymbol{z}_{0,j}\right) \right]$$
$$- \mathbb{E}_{\boldsymbol{z}_{0,j}' \sim \mathcal{D}_0} \mathbb{E}_{S_{i,1-\alpha} \sim \mathcal{D}_i^{n(1-\alpha)}(S_{0,\alpha}^j)} \left[ \mathbb{E}_{\boldsymbol{z} \sim \mathcal{D}_0} \ell\left(\mathcal{A}\left((S_{0,\alpha}^t)^j \cup S_{i,1-\alpha}\right), \boldsymbol{z}\right) - \ell\left(\mathcal{A}\left((S_{0,\alpha}^t)^j \cup S_{i,1-\alpha}\right), \boldsymbol{z}_{0,j}\right) \right] |$$
$$\tag{19}$$
$$+ | \mathbb{E}_{\boldsymbol{z}_{0,j}' \sim \mathcal{D}_0} \mathbb{E}_{S_{i,1-\alpha} \sim \mathcal{D}_i^{n(1-\alpha)}(S_{0,\alpha}^j)} \left[ \mathbb{E}_{\boldsymbol{z} \sim \mathcal{D}_0} \ell\left(\mathcal{A}\left((S_{0,\alpha}^t)^j \cup S_{i,1-\alpha}\right), \boldsymbol{z}\right) - \ell\left(\mathcal{A}\left((S_{0,\alpha}^t)^j \cup S_{i,1-\alpha}\right), \boldsymbol{z}_{0,j}\right) \right]$$
$$- \mathbb{E}_{\boldsymbol{z}_{0,j}' \sim \mathcal{D}_0} \mathbb{E}_{S_{i,1-\alpha} \sim \mathcal{D}_i^{n(1-\alpha)}((S_{0,\alpha}^t)^j)} \left[ \mathbb{E}_{\boldsymbol{z} \sim \mathcal{D}_0} \ell\left(\mathcal{A}\left((S_{0,\alpha}^t)^j \cup S_{i,1-\alpha}\right), \boldsymbol{z}\right) - \ell\left(\mathcal{A}\left((S_{0,\alpha}^t)^j \cup S_{i,1-\alpha}\right), \boldsymbol{z}_{0,j}\right) \right] |.$$
$$\tag{20}$$

We can bound equation 19 by applying the concept of uniform stability, resulting in an upper bound of $2\beta_n$. Regarding equation 20, for ease of notation, let us represent $\ell\left(\mathcal{A}\left((S_{0,\alpha}^t)^j \cup S_{i,1-\alpha}\right), z\right) - \ell\left(\mathcal{A}\left((S_{0,\alpha}^t)^j \cup S_{i,1-\alpha}\right), z_{0,j}\right)$ as $Q$. Consequently, we obtain the following:

$$
\begin{aligned}
&\left| \mathbb{E}_{z'_{0,j}\sim\mathcal{D}_0} \mathbb{E}_{S_{i,1-\alpha}\sim\mathcal{D}_i^{n(1-\alpha)}(S_{0,\alpha}^j)} \left[ \mathbb{E}_{z\sim\mathcal{D}_0} \ell\left(\mathcal{A}\left((S_{0,\alpha}^t)^j \cup S_{i,1-\alpha}\right), z\right) - \ell\left(\mathcal{A}\left((S_{0,\alpha}^t)^j \cup S_{i,1-\alpha}\right), z_{0,j}\right) \right] \right. \\
&\left. \quad - \mathbb{E}_{z'_{0,j}\sim\mathcal{D}_0} \mathbb{E}_{S_{i,1-\alpha}\sim\mathcal{D}_i^{n(1-\alpha)}((S_{0,\alpha}^t)^j)} \left[ \mathbb{E}_{z\sim\mathcal{D}_0} \ell\left(\mathcal{A}\left((S_{0,\alpha}^t)^j \cup S_{i,1-\alpha}\right), z\right) - \ell\left(\mathcal{A}\left((S_{0,\alpha}^t)^j \cup S_{i,1-\alpha}\right), z_{0,j}\right) \right] \right| \\
&= \left| \mathbb{E}_{z'_{0,j}\sim\mathcal{D}_0} \mathbb{E}_{z\sim\mathcal{D}_0} \left[ \mathbb{E}_{S_{i,1-\alpha}\sim\mathcal{D}_i^{n(1-\alpha)}(S_{0,\alpha}^j)} Q - \mathbb{E}_{S_{i,1-\alpha}\sim\mathcal{D}_i^{n(1-\alpha)}((S_{0,\alpha}^t)^j)} Q \right] \right| \\
&= \mathbb{E}_{z'_{0,j}\sim\mathcal{D}_0} \mathbb{E}_{z\sim\mathcal{D}_0} \left| \int_{S_{i,1-\alpha}} \left( \mathbb{P}\left(S_{i,1-\alpha} \mid S_{0,\alpha}^j\right) - \mathbb{P}\left(S_{i,1-\alpha} \mid (S_{0,\alpha}^t)^j\right) \right) Q \, dS_{i,1-\alpha} \right| \\
&\leq \mathbb{E}_{z'_{0,j}\sim\mathcal{D}_0} \mathbb{E}_{z\sim\mathcal{D}_0} \left[ \int_{S_{i,1-\alpha}} \left| \left( \mathbb{P}\left(S_{i,1-\alpha} \mid S_{0,\alpha}^j\right) - \mathbb{P}\left(S_{i,1-\alpha} \mid (S_{0,\alpha}^t)^j\right) \right) Q \right| dS_{i,1-\alpha} \right] \\
&\leq M \mathbb{E}_{z'_{0,j}\sim\mathcal{D}_0} \mathbb{E}_{z\sim\mathcal{D}_0} \left[ \int_{S_{i,1-\alpha}} \left| \left( \mathbb{P}\left(S_{i,1-\alpha} \mid S_{0,\alpha}^j\right) - \mathbb{P}\left(S_{i,1-\alpha} \mid (S_{0,\alpha}^t)^j\right) \right) \right| dS_{i,1-\alpha} \right] \\
&\leq 2M \sup_j TV\left( \mathcal{D}_i^{n(1-\alpha)}(S_0^j), \mathcal{D}_i^{n(1-\alpha)}(S_0) \right) \\
&= 2M\gamma_n^i.
\end{aligned}
\tag{21}
$$

Thus, $h_j$ exhibits a bounded difference of $2\beta_n + 2M\gamma_n^i$ with respect to all variables except the $j$-th variable. By applying Lemma 8, we obtain the following:

$$
\begin{aligned}
\left\| \sum_{j=1}^{n\alpha} h_j(S_{0,\alpha}) \right\|_p &\leq 12\sqrt{2} pn\alpha \left(2\beta_n + 2M\gamma_n^i\right) \log(n\alpha) + 4M\sqrt{pn\alpha} \\
&\lesssim pn\alpha \left(\beta_n + M\gamma_n^i\right) \log(n\alpha) + M\sqrt{pn\alpha}.
\end{aligned}
$$

We observe that the difference between Term 4 and $\left\| \sum_{j=1}^{n\alpha} h_j(S_{0,\alpha}) \right\|_p$ is negligible. Thus, we can bound Term 4 as follows:

$$
\begin{aligned}
&\left\| \mathbb{E}_{S_{i,1-\alpha}\sim\mathcal{D}_i^{n(1-\alpha)}} \left[ \alpha R_{\mathcal{D}_0}(\mathcal{A}(\widetilde{S}_i)) - \frac{1}{n} \sum_{z_i \in S_{0,\alpha}} \ell\left(\mathcal{A}\left(\widetilde{S}_i\right), z_i\right) \right] \right\|_p \\
&= \left\| \frac{1}{n} \sum_{z_i \in S_{0,\alpha}} \mathbb{E}_{S_{i,1-\alpha}\sim\mathcal{D}_i^{n(1-\alpha)}} \left[ R_{\mathcal{D}_0}(\mathcal{A}(\widetilde{S}_i)) - \ell\left(\mathcal{A}(\widetilde{S}_i), z_i\right) \right] \right\|_p \\
&\leq \left\| \frac{1}{n} \sum_{j=1}^{n\alpha} \left( \mathbb{E}_{z'_{0,j}\sim\mathcal{D}_0} \mathbb{E}_{S_{i,1-\alpha}\sim\mathcal{D}_i^{n(1-\alpha)}(S_{0,\alpha}^j)} \left[ \mathbb{E}_{z\sim\mathcal{D}_0} \ell\left(\mathcal{A}\left(S_{0,\alpha}^j \cup S_{i,1-\alpha}\right), z\right) - \ell\left(\mathcal{A}\left(S_{0,\alpha}^j \cup S_{i,1-\alpha}\right), z_{0,j}\right) \right] \right) \right\|_p \\
&\quad + \left\| 2\alpha\beta_n + 2\alpha M\gamma_n^i \right\|_p \\
&= \left\| \frac{1}{n} \sum_{j=1}^{n\alpha} h_j(S_{0,\alpha}) \right\|_p + \left\| 2\alpha\beta_n + 2\alpha M\gamma_n^i \right\|_p \\
&\lesssim p\alpha \left(\beta_n + M\gamma_n^i\right) \log(n\alpha) + M\sqrt{p\alpha n^{-1}} + \alpha\beta_n + \alpha M\gamma_n^i \\
&\lesssim p\alpha \left(\beta_n + M\gamma_n^i\right) \log(n\alpha) + M\sqrt{p\alpha n^{-1}}.
\end{aligned}
$$

By substituting the above inequality and inequality 17 into the decomposition 16, we obtain:

$$
\left\| \alpha R_{\mathcal{D}_0} \left( \mathcal{A} \left( \widetilde{S}_i \right) \right) - \frac{1}{n} \sum_{\boldsymbol{z}_i \in S_{0,\alpha}} \ell \left( \mathcal{A} \left( \widetilde{S}_i \right), \boldsymbol{z}_i \right) \right\|_p
$$
$$
\lesssim \sqrt{n(1-\alpha)p}\alpha\beta_n + p\alpha \left( \beta_n + M\gamma_n^i \right) \log(n\alpha) + M\sqrt{p\alpha n^{-1}}
$$
$$
\lesssim \sqrt{p}(\sqrt{(1-\alpha)n}\alpha\beta_n + M\sqrt{\alpha n^{-1}}) + p\alpha \left( \beta_n + M\gamma_n^i \right) \log(n\alpha). \tag{22}
$$

Plug inequalities 22 and 15 into the inequality 57, then we obtain:

$$
\| R_{\widetilde{\mathcal{D}}_i}(\mathcal{A}(\widetilde{S}_i)) - \widehat{R}_{\widetilde{S}_i}(\mathcal{A}(\widetilde{S}_i)) \|_p
$$
$$
\lesssim p(1-\alpha)\beta_n \log(n(1-\alpha)) + M\sqrt{p(1-\alpha)n^{-1}} + \sqrt{p}(\sqrt{(1-\alpha)n}\alpha\beta_n + M\sqrt{\alpha n^{-1}})
$$
$$
+ p\alpha \left( \beta_n + M\gamma_n^i \right) \log(n\alpha)
$$
$$
= \sqrt{p} \left( \sqrt{(1-\alpha)n}\alpha\beta_n + Mn^{-1/2}(\sqrt{1-\alpha} + \sqrt{\alpha}) \right)
$$
$$
+ p \left( (1-\alpha)\beta_n \log(n(1-\alpha)) + \alpha \left( \beta_n + M\gamma_n^i \right) \log(n\alpha) \right). \tag{23}
$$

By applying Lemma 8, we can derive a bound on the generalization error with respect to the mixed distribution. $\left| R_{\widetilde{\mathcal{D}}_i} \left( \mathcal{A} \left( \widetilde{S}_i \right) \right) - \widehat{R}_{\widetilde{S}_i} \left( \mathcal{A} \left( \widetilde{S}_i \right) \right) \right|$ as follows.

$$
\left| R_{\widetilde{\mathcal{D}}_i} \left( \mathcal{A} \left( \widetilde{S}_i \right) \right) - \widehat{R}_{\widetilde{S}_i} \left( \mathcal{A} \left( \widetilde{S}_i \right) \right) \right|
$$
$$
\lesssim \left( \sqrt{(1-\alpha)n}\alpha\beta_n + Mn^{-1/2}(\sqrt{1-\alpha} + \sqrt{\alpha}) \right) \sqrt{\log \left( \frac{1}{\delta} \right)}
$$
$$
+ \left( (1-\alpha)\beta_n \log(n(1-\alpha)) + \alpha \left( \beta_n + M\gamma_n^i \right) \log(n\alpha) \right) \log \left( \frac{1}{\delta} \right).
$$

Finally, we conclude that:

$$
\left| R_{\mathcal{D}_0} \left( A \left( \widetilde{S}_i \right) \right) - \widehat{R}_{\widetilde{S}_i} \left( A \left( \widetilde{S}_i \right) \right) \right|
$$
$$
\leq \left| R_{\mathcal{D}_0} \left( A \left( \widetilde{S}_i \right) \right) - R_{\widetilde{\mathcal{D}}_i} \left( A \left( \widetilde{S}_i \right) \right) \right| + \left| R_{\widetilde{\mathcal{D}}_i} \left( A \left( \widetilde{S}_i \right) \right) - \widehat{R}_{\widetilde{S}_i} \left( A \left( \widetilde{S}_i \right) \right) \right|
$$
$$
\leq 2M \left( 1 - (1-\alpha)^i \right) \alpha^{-1} d_{\mathrm{TV}}(n)
$$
$$
+ \left( (1-\alpha)\beta_n \log(n(1-\alpha)) + \alpha \left( \beta_n + M\gamma_n^i \right) \log(n\alpha) \right) \log \left( \frac{1}{\delta} \right)
$$
$$
+ \left( \sqrt{(1-\alpha)n}\alpha\beta_n + Mn^{-1/2}(\sqrt{1-\alpha} + \sqrt{\alpha}) \right) \sqrt{\log \left( \frac{1}{\delta} \right)}. \tag{24}
$$

$\square$

## A.7 PROOF OF THEOREM 2

In this section, we prove that transformers in in-context learning exhibit recursive stability. Specifically, we utilize the framework and lemmas from Li et al. (2023), combined with recursive techniques, to establish the proof.

**Lemma 10** (Li et al. (2023)). *Let $z, \varepsilon \in \mathbb{R}^n$ be vectors obeying $\|z\|_{\ell_\infty}, \|z + \varepsilon\|_{\ell_\infty} \leq c$. Then, there exists a constant $C = C(c)$, such that*

$$
\| \mathrm{softmax}(z) \|_{\ell_\infty} \leq e^{2c}/n \quad \text{and} \quad \| \mathrm{softmax}(z) - \mathrm{softmax}(z + \varepsilon) \|_{\ell_1} \leq e^{2c}\|\varepsilon\|_{\ell_1}/n.
$$

*Proof of Theorem 2.* . Let $\boldsymbol{Z} = [\boldsymbol{z}_1, \ldots, \boldsymbol{z}_n]^\top$ and $\boldsymbol{E} = [\varepsilon_1, \ldots, \varepsilon_n]^\top$ be the input and perturbation matrices respectively. Given that the tokens $\boldsymbol{z}_i$ lie in the unit ball, and assuming $\boldsymbol{z}_i + \varepsilon_i$ also

lies in the unit ball, we can proceed with the following. For a matrix, let $\| \cdot \|_{2,p}$ denote the $\ell_p$-norm of the vector formed by the $\ell_2$-norms of its rows. Therefore, we obtain $\|\bar{Z}\|_{2,\infty} \leq 1$ and $\|\bar{Z}\|_{2,\infty} = \|Z + E\|_{2,\infty} \leq 1$. Let the attention outputs be defined as $A = \mathrm{softmax}\left(ZWZ^\top\right)ZV$ and $\bar{A} = \mathrm{softmax}\left(\bar{Z}W\bar{Z}^\top\right)\bar{Z}V$. Define the perturbation as $\bar{E} = \bar{A} - A := [\bar{\varepsilon}_1, \ldots \bar{\varepsilon}_n]^\top$.

Let us examine the attention output difference $\bar{E} = \bar{A} - A$, which can be further decomposed as follows:

$$
\begin{aligned}
\bar{E} &= \mathrm{softmax}\left(\bar{Z}W\bar{Z}^\top\right)\bar{Z}V - \mathrm{softmax}\left(ZWZ^\top\right)ZV \\
&= \underbrace{\left[\mathrm{softmax}\left(\bar{Z}W\bar{Z}^\top\right) - \mathrm{softmax}\left(ZWZ^\top\right)\right]ZV}_{\bar{E}_1} + \underbrace{\mathrm{softmax}\left(\bar{Z}W\bar{Z}^\top\right)EV}_{\bar{E}_2}.
\end{aligned}
\tag{25}
$$

We first observe that $V$ preserves the norm, meaning that $ZV$ satisfies $\|ZV\|_{2,\infty} \leq \|Z\|_{2,\infty} \leq 1$ and $\|EV\|_{2,1} \leq \|E\|_{2,1}$. Moreover, for any pair of tokens, it holds that $\left|z_i^\top W z_j\right| \leq B_W$. Applying Lemma 10, we can therefore derive the following:

$$
\left\|\bar{E}_2\right\|_{2,1} = \left\|\mathrm{softmax}\left(\bar{Z}W\bar{Z}^\top\right)EV\right\|_{2,1} \leq e^{2B_W}\|E\|_{2,1}.
\tag{26}
$$

Subsequently, for $\bar{E}_1$, we establish the following expression

$$
\begin{aligned}
\left\|\bar{E}_1\right\|_{2,1} &= \left\|[\mathrm{softmax}\left(\bar{Z}W\bar{Z}^\top\right) - \mathrm{softmax}\left(ZWZ^\top\right)]ZV\right\|_{2,1} \\
&\leq \left\|\mathrm{softmax}\left(\bar{Z}W\bar{Z}^\top\right) - \mathrm{softmax}\left(ZWZ^\top\right)\right\|_{\ell_1}\|ZV\|_{2,\infty} \\
&\leq \left\|\mathrm{softmax}\left(\bar{Z}W\bar{Z}^\top\right) - \mathrm{softmax}\left(ZWZ^\top\right)\right\|_{\ell_1}.
\end{aligned}
$$

To advance the analysis, we introduce the $\delta$-scaled perturbation $E' = \delta E = \bar{Z}' - Z$, where $\delta$ is constrained within $0 \leq \delta \leq 1$. Our approach involves first bounding the derivative as $\delta \to 0$, and then integrating this bound along the path of $E$, effectively covering the interval from $\delta = 0$ to $\delta = 1$. Notably, as $\delta \to 0$, the quadratic terms proportional to $\delta^2 E$ diminish, simplifying the analysis at this limit.

$$
\begin{aligned}
&\left\|\mathrm{softmax}\left(\bar{Z}'W\bar{Z}'^\top\right) - \mathrm{softmax}\left(ZWZ^\top\right)\right\|_{\ell_1} \\
&\leq \left\|\mathrm{softmax}(\bar{Z}'WZ^\top) - \mathrm{softmax}(ZWZ^\top)\right\|_{\ell_1} + \left\|\mathrm{softmax}(ZW\bar{Z}'^\top) - \mathrm{softmax}(ZWZ^\top)\right\|_{\ell_1}.
\end{aligned}
$$

To bound the latter, we focus on each row separately. Consider a row from $Z$ and its perturbed version $Z + E'$, represented by the pair $(z, z + \varepsilon')$. It follows that for any cross product, we have the guarantees $\left|(z + \varepsilon')^\top W z_i\right| \leq B_W$ and $\left|z^\top W z_i\right| \leq B_W$. Additionally, the bounds $\left\|\varepsilon'^\top WZ\right\|_{\ell_1} \leq B_W n \|\varepsilon'\|_{\ell_2}$ and $\left\|z^\top WE'^\top\right\|_{\ell_1} \leq B_W \|E'\|_{2,1}$ hold. Applying the perturbation bounds provided by Lemma 10, we obtain the desired result

$$
\left\|\mathrm{softmax}\left((z + \varepsilon')^\top WZ^\top\right) - \mathrm{softmax}\left(z^\top WZ^\top\right)\right\|_{\ell_1} \leq B_W e^{2B_W}\|\varepsilon'\|_{\ell_2}
$$

$$
\left\|\mathrm{softmax}\left(z^\top W\left(Z + E'\right)^\top\right) - \mathrm{softmax}\left(z^\top WZ^\top\right)\right\|_{\ell_1} \leq B_W e^{2B_W}\|E'\|_{2,1}/n.
$$

Summing across all $n$ rows, we obtain the following:

$$
\lim_{\delta \to 0}\left\|\mathrm{softmax}\left((Z + \delta E)W\bar{Z}^\top\right) - \mathrm{softmax}\left(ZWZ^\top\right)\right\|_{\ell_1}/\delta \leq 2B_W e^{2B_W}\|E\|_{2,1}.
$$

By integrating the derivative over the interval $\delta = 0$ to $\delta = 1$, we obtain the final expression,

$$
\left\|\mathrm{softmax}\left(\bar{Z}W\bar{Z}^\top\right) - \mathrm{softmax}\left(ZWZ^\top\right)\right\|_{\ell_1} \leq 2B_W e^{2B_W}\|E\|_{2,1}.
\tag{27}
$$

By substituting inequality 27 and inequality 26 into the decomposition 25, we derive the following result:

$$
\|\bar{A} - A\|_{2,1} = \|\bar{E}\|_{2,1} \leq (2B_W + 1)e^{2B_W}\|E\|_{2,1}.
\tag{28}
$$

To continue, we aim to control the output for a specific index $j$ where the input perturbation remains small, specifically $\|\varepsilon_j\|_{\ell_2} \leq \frac{\|E\|_{2,1}}{n}$. To address this, we will apply the same argument, focusing

on the $j$-th token. For the $j$-th token (omitting subscripts for clarity), let the inputs be denoted as $z, \bar{z}, \varepsilon = \bar{z} - z$, and the corresponding outputs as $a, \bar{a}, \bar{\varepsilon} = \bar{a} - a$. Similar to the previous decomposition, we can derive the following:

$$\bar{\varepsilon} = \underbrace{V^\top Z^\top \left[ \text{softmax} \left( \bar{Z} W^\top \bar{Z} \right) - \text{softmax} \left( Z W^\top Z \right) \right]}_{\bar{\varepsilon}_1} + \underbrace{V^\top E^\top \text{softmax} \left( \bar{Z} W^\top \bar{Z} \right)}_{\bar{\varepsilon}_2}. \quad (29)$$

By leveraging the fact that $\left| z_i^\top W z_j \right| \le B_W$ for all $i, j$, and applying Lemma 10, we can establish a bound similar to that in equation 26. Specifically, we can constrain the terms involved as follows:

$$\left\| \bar{\varepsilon}_2 \right\|_{\ell_2} \le \left\| E^\top \text{softmax} \left( \bar{Z} W^\top \bar{z} \right) \right\|_{\ell_2} \le \frac{e^{2B_W}}{n} \| E \|_{2,1}. \quad (30)$$

Similarly, for $\bar{\varepsilon}_1$, we can derive the following:

$$\begin{aligned}
\left\| \bar{\varepsilon}_1 \right\|_{\ell_2} &\le \left\| Z^\top \left[ \text{softmax} \left( \bar{Z} W^\top \bar{z} \right) - \text{softmax} \left( Z W^\top z \right) \right] \right\|_{\ell_2} \\
&\le \| Z \|_{2,\infty} \left\| \text{softmax} \left( \bar{Z} W^\top \bar{z} \right) - \text{softmax} \left( Z W^\top z \right) \right\|_{\ell_1} \\
&\le \left\| \text{softmax} \left( \bar{Z} W^\top \bar{z} \right) - \text{softmax} \left( Z W^\top z \right) \right\|_{\ell_1}.
\end{aligned}$$

Now, considering the perturbation $E' = \delta E$, and letting $\delta \to 0$, we apply the triangle inequality to obtain the following result:

$$\begin{aligned}
\lim_{\delta \to 0} \delta^{-1} &\left\| \text{softmax} \left( (Z + \delta E) W^\top (z + \delta \varepsilon) \right) - \text{softmax} \left( Z W^\top z \right) \right\|_{\ell_1} \\
&\le \lim_{\delta \to 0} \delta^{-1} \left\| \text{softmax} \left( (Z + \delta E) W^\top z \right) - \text{softmax} \left( Z W^\top z \right) \right\|_{\ell_1} \\
&\quad + \delta^{-1} \left\| \text{softmax} \left( Z W^\top (z + \delta \varepsilon) \right) - \text{softmax} \left( Z W^\top z \right) \right\|_{\ell_1} \\
&\le B_W e^{2B_W} \| E \|_{2,1} / n + B_W e^{2B_W} \| \varepsilon \|_{\ell_2} \\
&\le 2 B_W e^{2B_W} \| E \|_{2,1} / n. \quad (31)
\end{aligned}$$

In a similar manner to the previous steps, we can derive the following:

$$\left\| \bar{\varepsilon} \right\|_{\ell_2} \le \frac{1}{n} (2 B_W + 1) e^{2B_W} \| E \|_{2,1}. \quad (32)$$

Next, we examine the effect of the MLP layer on the model's behavior. Let $(M_i)_{i=1}^n \in \mathbb{R}^{d \times d}$ represent the weights of the parallel MLPs that follow the self-attention mechanism. Given that $\| M_i \| \le 1$, we denote the MLP outputs corresponding to the self-attention results $A$ and $\bar{A}$ as $U$ and $\bar{U}$, respectively. From this, we can derive the following relationship.

Let $\phi$ denote the ReLU function, which is a 1-Lipschitz continuous activation function with $\phi(0) = 0$. First, observe that each row of $U$ is given by $u_i = \phi(M_i a_i)$, where $M_i \in \mathbb{R}^{d \times d}$ represents the weights of the MLPs. Given the properties of the ReLU function, we can derive the following bound:

$$\| u_i \|_{\ell_2} \le \| \phi(M_i a_i) \|_{\ell_2} \le \| M_i a_i \|_{\ell_2} \le \| a_i \|_{\ell_2} \le 1.$$

Next, we consider the difference between the perturbed and original outputs. We can express the difference as $\| u_i - \bar{u}_i \|_{\ell_2} \le \| \phi(M_i a_i) - \phi(M_i \bar{a}_i) \|_{\ell_2}$, which, due to the 1-Lipschitz property of $\phi$, is further bounded by $\| M_i (a_i - \bar{a}_i) \|_{\ell_2} \le \| a_i - \bar{a}_i \|_{\ell_2}$. Finally, we obtain:

$$\| u_i - \bar{u}_i \|_{\ell_2} \le \| a_i - \bar{a}_i \|_{\ell_2}. \quad (33)$$

Thus, we conclude that the perturbations in the rows of $U$ are controlled by the corresponding perturbations in $A$. Consequently, we establish the bound

$$\| U - \bar{U} \|_{2,1} \le \| A - \bar{A} \|_{2,1}.$$

Thus, from inequality 28, we derive the following result:

$$\| U - \bar{U} \|_{2,1} \le (2 B_W + 1) e^{2B_W} \| E \|_{2,1}. \quad (34)$$

Furthermore, for any $i \in [n]$ such that $\| \varepsilon_i \|_{\ell_2} \le \frac{\| E \|_{2,1}}{n}$, it holds that

$$\| u_i - \bar{u}_i \|_{\ell_2} \le \frac{1}{n} (2 B_W + 1) e^{2B_W} \| E \|_{2,1},$$

where $\boldsymbol{u}_i$ represents the $i$-th row of $\boldsymbol{U}$. With this, we have addressed the stability of the single-layer transformer. Moving forward, we will extend our analysis and focus on the stability of $L$-layer transformer. First, we can derive the following:

$$\left\|\boldsymbol{Z}_{(k)} - \bar{\boldsymbol{Z}}_{(k)}\right\|_{2,1} \leq (1 + 2B_W)e^{2B_W}\left\|\boldsymbol{Z}_{(k-1)} - \bar{\boldsymbol{Z}}_{(k-1)}\right\|_{2,1},$$

where $1 \leq k \leq L$ represents the number of layers in the transformer. Then, for $L$-layer transformer, we have the following:

$$\left\|\boldsymbol{Z}_{(L)} - \bar{\boldsymbol{Z}}_{(L)}\right\|_{2,1} \leq ((1 + 2B_W)e^{2B_W})^L\left\|\boldsymbol{Z}_{(0)} - \bar{\boldsymbol{Z}}_{(0)}\right\|_{2,1}$$

What remains is to perform induction on the difference between the last tokens $\boldsymbol{z}_n^{(i)} - \boldsymbol{z}_n^{\prime(i)}$. We claim that, for all layers,

$$\left\|\boldsymbol{z}_n^{(i)} - \boldsymbol{z}_n^{\prime(i)}\right\|_{\ell_2} \leq \frac{1}{n}((1 + 2B_W)e^{2B_W})^i\left\|\boldsymbol{Z}_{(0)} - \bar{\boldsymbol{Z}}_{(0)}\right\|_{2,1}.$$

This claim holds at $i = 0$ because the change in the last token is at most $\left\|\boldsymbol{Z}_{(0)} - \bar{\boldsymbol{Z}}_{(0)}\right\|_{2,1}/n$. By induction, the claim holds for all layers, and we conclude the proof by setting $i = L$, covering the entire depth of the $L$-layer transformer. Finally, we obtain:

$$\left\|\boldsymbol{z}_n^{(L)} - \boldsymbol{z}_n^{\prime(L)}\right\|_{\ell_2} \leq \frac{1}{n}((1 + 2B_W)e^{2B_W})^L\left\|\boldsymbol{Z}_{(0)} - \bar{\boldsymbol{Z}}_{(0)}\right\|_{2,1}. \tag{35}$$

Next, we further analyze the self-consuming process. Let $S_0 = [\boldsymbol{z}_{0,1}, ..., \boldsymbol{z}_{0,j}, ..., \boldsymbol{z}_{0,n}]^\top$ and $S_0' = [\boldsymbol{z}_{0,1}, ..., \boldsymbol{z}_{0,j}', ..., \boldsymbol{z}_{0,n}]^\top$ represent two initial real datasets that differ only in their inputs, specifically $\boldsymbol{z}_{0,j} = (\boldsymbol{x}_{0,j}, \boldsymbol{y}_{0,j})$ and $\boldsymbol{z}_{0,j}' = (\boldsymbol{x}_{0,j}', \boldsymbol{y}_{0,j}')$, where $j \leq n$. Since $\|S_0 - S_0'\|_{2,1} \leq 2$, then, we have the following:

$$\|\mathrm{TF}(S_0) - \mathrm{TF}(S_0')\|_{\ell_2} \leq \frac{1}{2n+1}\left((1 + 2B_W)e^{2B_W}\right)^L \|S_0 - S_0'\|_{2,1} \tag{36}$$

$$\leq \frac{2}{2n+1}\left((1 + 2B_W)e^{2B_W}\right)^L.$$

Then, $S_0$ and $S_0'$ are used as in-context examples, and i.i.d. queries $\{\boldsymbol{x}_{1,j}\}_{j=1}^n$ are sampled from $\mathcal{X}$. These queries, along with the in-context examples $S_0$ and $S_0'$, are processed through the transformer model to predict their respective labels. As a result, the first generation of synthetic datasets, $S_1 = [\boldsymbol{z}_{1,1}, ..., \boldsymbol{z}_{1,j}, ..., \boldsymbol{z}_{1,n}]^\top$ and $S_1' = [\boldsymbol{z}_{1,1}', ..., \boldsymbol{z}_{1,j}', ..., \boldsymbol{z}_{1,n}']^\top$, is produced. Then we obtain:

$$\|S_1 - S_1'\|_{2,1} \leq \frac{2n}{2n+1}\left((1 + 2B_W)e^{2B_W}\right)^L. \tag{37}$$

Given the mixed dataset $\widetilde{S}_j$, where $\widetilde{S}_j = \alpha S_0 + (1-\alpha)S_j$ for $1 \leq j \leq i$, we can proceed with further analysis based on the specified combination of the original dataset $S_0$ and the synthetic dataset $S_j$.

$$\|\widetilde{S}_1 - \widetilde{S}_1'\|_{2,1} \leq \alpha\|S_0 - S_0'\|_{2,1} + (1-\alpha)\|S_1 - S_1'\|_{2,1}$$

$$\leq 2\alpha + (1-\alpha)\frac{2n}{2n+1}\left((1 + 2B_W)e^{2B_W}\right)^L. \tag{38}$$

By reintroducing the mixed datasets $\widetilde{S}_1$ and $\widetilde{S}_1'$ as in-context examples into the transformer model, and considering the query set $\{\boldsymbol{x}_{2,j}\}_{j=1}^n$ as i.i.d. samples from the distribution $\mathcal{X}$, we can derive the transformer's output according to Equation 36:

$$\left\|\mathrm{TF}\left(\widetilde{S}_1\right) - \mathrm{TF}\left(\widetilde{S}_1'\right)\right\|_{\ell_2}$$

$$\leq \frac{1}{2n+1}\left((1 + 2B_W)e^{2B_W}\right)^L\|\widetilde{S}_1 - \widetilde{S}_1'\|_{2,1}$$

$$\leq \frac{1}{2n+1}\left((1 + 2B_W)e^{2B_W}\right)^L\left(2\alpha + (1-\alpha)\frac{2n}{2n+1}\left((1 + 2B_W)e^{2B_W}\right)^L\right)$$

$$\leq (1-\alpha)\frac{2n}{(2n+1)^2}\left((1 + 2B_W)e^{2B_W}\right)^{2L} + \alpha\frac{2}{2n+1}\left((1 + 2B_W)e^{2B_W}\right)^L. \tag{39}$$

From the above expression, we can further derive that

$$\|S_2 - S_2'\|_{2,1} \le (1-\alpha)\frac{2n^2}{(2n+1)^2}\left((1+2B_W)e^{2B_W}\right)^{2L} + \alpha\frac{2n}{2n+1}\left((1+2B_W)e^{2B_W}\right)^L.$$

Thus,

$$
\begin{aligned}
&\|\widetilde{S}_2 - \widetilde{S}_2'\|_{2,1} \\
&\le \alpha\|S_0 - S_0'\|_{2,1} + (1-\alpha)\|S_2 - S_2'\|_{2,1} \\
&\le 2\alpha + (1-\alpha)^2\frac{2n^2}{(2n+1)^2}\left((1+2B_W)e^{2B_W}\right)^{2L} + \alpha(1-\alpha)\frac{2n}{2n+1}\left((1+2B_W)e^{2B_W}\right)^L.
\end{aligned}
$$

Similarly, for the 2-th generation, following analogous steps, we can derive that

$$
\begin{aligned}
&\left\|\mathrm{TF}\left(\widetilde{S}_2\right) - \mathrm{TF}\left(\widetilde{S}_2'\right)\right\|_{\ell_2} \\
&\le \frac{1}{2n+1}\left((1+2B_W)e^{2B_W}\right)^L \|\widetilde{S}_2 - \widetilde{S}_2'\|_{2,1} \\
&\le (1-\alpha)^2\frac{2n^2}{(2n+1)^3}\left((1+2B_W)e^{2B_W}\right)^{3L} + \alpha(1-\alpha)\frac{2n}{(2n+1)^2}\left((1+2B_W)e^{2B_W}\right)^{2L} \\
&\quad + \alpha\frac{2}{2n+1}\left((1+2B_W)e^{2B_W}\right)^L.
\end{aligned}
\tag{40}
$$

Building on the above expression, we can further deduce that

$$
\begin{aligned}
&\|S_3 - S_3'\|_{2,1} \\
&\le (1-\alpha)^2\frac{2n^3}{(2n+1)^3}\left((1+2B_W)e^{2B_W}\right)^{3L} + \alpha(1-\alpha)\frac{2n^2}{(2n+1)^2}\left((1+2B_W)e^{2B_W}\right)^{2L} \\
&\quad + \alpha\frac{2n}{2n+1}\left((1+2B_W)e^{2B_W}\right)^L.
\end{aligned}
\tag{41}
$$

The discrepancy between the mixed datasets is as follows:

$$
\begin{aligned}
&\|\widetilde{S}_3 - \widetilde{S}_3'\|_{2,1} \\
&\le \alpha\|S_0 - S_0'\|_{2,1} + (1-\alpha)\|S_3 - S_3'\|_{2,1} \\
&\le (1-\alpha)^3\frac{2n^3}{(2n+1)^3}\left((1+2B_W)e^{2B_W}\right)^{3L} + \alpha(1-\alpha)^2\frac{2n^2}{(2n+1)^2}\left((1+2B_W)e^{2B_W}\right)^{2L} \\
&\quad + \alpha(1-\alpha)\frac{2n}{2n+1}\left((1+2B_W)e^{2B_W}\right)^L + 2\alpha.
\end{aligned}
\tag{42}
$$

Utilizing recursive techniques, we can obtain the following:

$$
\begin{aligned}
&\|\widetilde{S}_i - \widetilde{S}_i'\|_{2,1} \\
&\le (1-\alpha)^i\frac{2n^i}{(2n+1)^i}\left((1+2B_W)e^{2B_W}\right)^{iL} + \alpha(1-\alpha)^{i-1}\frac{2n^{i-1}}{(2n+1)^{i-1}}\left((1+2B_W)e^{2B_W}\right)^{(i-1)L} \\
&\quad + ... + \alpha(1-\alpha)^2\frac{2n^2}{(2n+1)^2}\left((1+2B_W)e^{2B_W}\right)^{2L} + \alpha(1-\alpha)\frac{2n}{2n+1}\left((1+2B_W)e^{2B_W}\right)^L \\
&\quad + 2\alpha \\
&\le 2(1-\alpha)^i\frac{n^i}{(2n+1)^i}\left((1+2B_W)e^{2B_W}\right)^{iL} \\
&\quad + 2\alpha\left[1 - (1-\alpha)\frac{n}{2n+1}\left((1+2B_W)e^{2B_W}\right)^L\right]^{-1}\left[1 - (1-\alpha)^i\frac{n^i}{(2n+1)^i}\left((1+2B_W)e^{2B_W}\right)^{iL}\right].
\end{aligned}
\tag{43}
$$

Ultimately, the discrepancy between the transformer outputs after $i$ generations of the self-consuming loop for $S_0$ and $S_0'$ can be obtained as follows:

$$
\left\| \mathrm{TF}\left( \widetilde{S}_i \right) - \mathrm{TF}\left( \widetilde{S}_i' \right) \right\|_{\ell_2}
$$

$$
\leq \frac{1}{2n+1} \left( (1+2B_W)e^{2Bw} \right)^L \|\widetilde{S}_i - \widetilde{S}_i'\|_{2,1}
$$

$$
\leq (1-\alpha)^i \frac{2n^i}{(2n+1)^{i+1}} \left( (1+2B_W) e^{2Bw} \right)^{(i+1)L} + \alpha(1-\alpha)^{i-1} \frac{2n^{i-1}}{(2n+1)^i} \left( (1+2B_W) e^{2Bw} \right)^{iL}
$$

$$
+ ... + \alpha(1-\alpha)^2 \frac{2n^2}{(2n+1)^3} \left( (1+2B_W) e^{2Bw} \right)^{3L} + \alpha(1-\alpha) \frac{2n}{(2n+1)^2} \left( (1+2B_W) e^{2Bw} \right)^{2L}
$$

$$
+ 2\alpha \frac{1}{2n+1} \left( (1+2B_W)e^{2Bw} \right)^L
$$

$$
\leq 2(1-\alpha)^i \frac{n^i}{(2n+1)^{i+1}} \left( (1+2B_W) e^{2Bw} \right)^{(i+1)L} + 2\alpha \left[ \frac{1}{2n+1} \left( (1+2B_W)e^{2Bw} \right)^L \right]
$$

$$
\times \left[ 1 - (1-\alpha)\frac{n}{2n+1} \left( (1+2B_W) e^{2Bw} \right)^L \right]^{-1} \left[ 1 - (1-\alpha)^i \frac{n^i}{(2n+1)^i} \left( (1+2B_W) e^{2Bw} \right)^{iL} \right].
$$

Subsequently, given that $\widetilde{B}_W = (1+2B_W)e^{2Bw}$, we define the measure $d$ as the $\ell_2$-norm to quantify the output discrepancy of the generative transformer model after $i$ iterations of the self-consuming loop, starting from the initial real datasets $S_0$ and $S_0'$. In this context, the recursive stability parameter $\gamma_n^i$, as described in Definition 2, can be bounded by the following expression, providing a formal measure of the model's stability across iterations:

$$
\left\| \mathrm{TF}(\widetilde{S}_i) - \mathrm{TF}(\widetilde{S}_i') \right\|_{\ell_2} \lesssim (1-\alpha)^i \frac{\widetilde{B}_W^{(i+1)L}}{2n+1}.
$$

The proof is complete.

$\square$

### A.8 PROOF OF THEOREM 3

In this section, building on the general theoretical framework established in Theorem 1, we provide the proof of Theorem 3 by analyzing the terms $\beta_n$ and $d_{\mathrm{TV}}(n)$, leveraging recent advancements in SGD (Zhang et al., 2022) and ICL (Zhang et al., 2023). The recursive stability parameter $\gamma_n^i$ is derived from Theorem 2.

**Lemma 11.** *(Uniform stability of SGD in the non-convex case (Zhang et al., 2022)). Assume $f$ is $\kappa$-smooth and $\rho$-Lipschitz. Running $T \gtrsim n$ iterations of SGD with step size $\eta_t = \frac{1}{\beta t}$. Choose the stability of SGD satisfies*

$$
\beta_n \lesssim \frac{16\rho^2 \log n}{n}.
$$

**Lemma 12.** *(Zhang et al., 2023) Let $\mathbb{P}_\theta$ represent the probability distribution induced by the transformer with parameter $\theta$. Additionally, the model $\mathbb{P}_{\hat{\theta}}$ is pretrained by the algorithm:*

$$
\hat{\theta} = \underset{\theta \in \Theta}{\mathrm{argmin}} - \frac{1}{n} \sum_{t=1}^{n-1} \log \mathbb{P}_\theta \left( \boldsymbol{x}_{t+1}^n \mid S_t^n \right),
$$

*where $S_t^n = (\boldsymbol{x}_1, \boldsymbol{y}_1, \ldots \boldsymbol{x}_t, \boldsymbol{y}_t)$. Furthermore, we consider the realizable setting, where ground truth probability distribution $\mathbb{P}(\cdot \mid S)$ and $\mathbb{P}_{\theta^*}(\cdot \mid S)$ are consistent for some $\theta^* \in \Theta$. Then, with probability at least $1 - \delta$, the following inequality holds:*

$$
\mathrm{TV}\left( \mathbb{P}(\cdot \mid S), \mathbb{P}_{\hat{\theta}}(\cdot \mid S) \right) \lesssim \frac{1}{n^{1/2}} \log(1+n) + \frac{1}{n^{1/4}} \log(1/\delta), \tag{44}
$$

*where $\lesssim$ denotes that we omit constants that are independent of $n$ and $\delta$.*

*Proof of Theorem 3.* First, we note that in the setting where the transformer generates data through in-context learning, the generalization error of the self-consuming loop is given by:

$$\left| R_{\mathcal{D}_0}(\mathcal{A}(\widetilde{S}_i)) - \widehat{R}_{\widetilde{S}_i}(\mathcal{A}(\widetilde{S}_i)) \right| = \left| \mathbb{E}_{\boldsymbol{z} \sim \mathbb{P}(\cdot | S_0)} \ell(\mathcal{A}(\widetilde{S}_i), \boldsymbol{z}) - \frac{1}{n} \sum_{\boldsymbol{z}_i \in \widetilde{S}_i} \ell(\mathcal{A}(\widetilde{S}_i), \boldsymbol{z}_i) \right|. \quad (45)$$

Now, we are ready to prove Theorem 3. The main idea is to bound the uniform stability parameter $\beta_n$, the recursive stability parameter $\gamma_n^i$, and the learnability of the generative model through the total variation distance $d_{\text{TV}}(n)$ as stated in Theorem 1. First, as for the bound for the total variation distance $d_{\text{TV}}(n)$ in Theorem 1. For Equation 8 in the proof of Theorem 1, we can rewrite it in the setting of in-context learning as follows:

$$
\begin{aligned}
\left| R_{\widetilde{\mathcal{D}}_{i-1}}(\mathcal{A}(\widetilde{S}_i)) - R_{\mathcal{D}_i}(\mathcal{A}(\widetilde{S}_i)) \right| &= \left| \mathbb{E}_{\boldsymbol{z} \sim \mathbb{P}(\cdot | \widetilde{S}_{i-1})} \ell(\mathcal{A}(\widetilde{S}_i), \boldsymbol{z}) - \mathbb{E}_{\boldsymbol{z} \sim \mathbb{P}(\cdot | S_i)} \ell(\mathcal{A}(\widetilde{S}_i), \boldsymbol{z}) \right| \\
&= \left| \mathbb{E}_{\boldsymbol{z} \sim \mathbb{P}(\cdot | \widetilde{S}_{i-1})} \ell(\mathcal{A}(\widetilde{S}_i), \boldsymbol{z}) - \mathbb{E}_{\boldsymbol{z} \sim \mathbb{P}_{\hat{\theta}}(\cdot | \widetilde{S}_{i-1})} \ell(\mathcal{A}(\widetilde{S}_i), \boldsymbol{z}) \right| \\
&= \left| \int_{\boldsymbol{z}} \ell(\mathcal{A}(\widetilde{S}_i), \boldsymbol{z}) \left( \mathbb{P}\left( \boldsymbol{z} \mid \widetilde{S}_{i-1} \right) - \mathbb{P}_{\hat{\theta}} \left( \boldsymbol{z} \mid \widetilde{S}_{i-1} \right) \right) d\boldsymbol{z} \right| \\
&\leq \int_{\boldsymbol{z}} \left| \ell(\mathcal{A}(\widetilde{S}), \boldsymbol{z}) \left( \mathbb{P}\left( \boldsymbol{z} \mid \widetilde{S}_{i-1} \right) - \mathbb{P}_{\hat{\theta}} \left( \boldsymbol{z} \mid \widetilde{S}_{i-1} \right) \right) \right| d\boldsymbol{z} \\
&\leq M \int_{\boldsymbol{z}} \left| \mathbb{P}\left( \boldsymbol{z} \mid \widetilde{S}_{i-1} \right) - \mathbb{P}_{\hat{\theta}} \left( \boldsymbol{z} \mid \widetilde{S}_{i-1} \right) \right| d\boldsymbol{z} \\
&= 2MTV \left( \mathbb{P}(\cdot \mid \widetilde{S}_{i-1}), \mathbb{P}_{\hat{\theta}}(\cdot \mid \widetilde{S}_{i-1}) \right). \quad (46)
\end{aligned}
$$

Where, the second equality holds because, in the $(i-1)$-th generation of the self-consuming loop, the mixed data distribution from the $(i-1)$-th generation is reintroduced as the ground truth distribution to train the transformer. As a result, the transformer outputs the synthetic data distribution for the $i$-th generation. Thus, $TV \left( \mathbb{P}(\cdot \mid \widetilde{S}_j), \mathbb{P}_{\hat{\theta}}(\cdot \mid \widetilde{S}_j) \right)$ corresponds to $d_{\text{TV}}(n)$ in Theorem 1. Finally, the bound for the total variation distance $d_{\text{TV}}(n)$ follows from Lemma 12.

$$d_{\text{TV}}(n) \lesssim \frac{1}{n^{1/2}} \log(1+n) + \frac{1}{n^{1/4}} \log(1/\delta). \quad (47)$$

Similarly, for the recursive stability parameter in the self-consuming loop, we rederive Equation 21 from the proof of Theorem 1 under the in-context learning setting:

$$
\begin{aligned}
& \left| \mathbb{E}_{\boldsymbol{z}'_{0,j} \sim \mathcal{D}_0} \mathbb{E}_{S_{i,1-\alpha} \sim \mathcal{D}_i^{n(1-\alpha)}(S_{0,\alpha}^j)} \left[ \mathbb{E}_{\boldsymbol{z} \sim \mathcal{D}_0} \ell \left( \mathcal{A} \left( (S_{0,\alpha}^t)^j \cup S_{i,1-\alpha} \right), \boldsymbol{z} \right) - \ell \left( \mathcal{A} \left( (S_{0,\alpha}^t)^j \cup S_{i,1-\alpha} \right), \boldsymbol{z}_{0,j} \right) \right] \right. \\
& \left. - \mathbb{E}_{\boldsymbol{z}'_{0,j} \sim \mathcal{D}_0} \mathbb{E}_{S_{i,1-\alpha} \sim \mathcal{D}_i^{n(1-\alpha)}((S_{0,\alpha}^t)^j)} \left[ \mathbb{E}_{\boldsymbol{z} \sim \mathcal{D}_0} \ell \left( \mathcal{A} \left( (S_{0,\alpha}^t)^j \cup S_{i,1-\alpha} \right), \boldsymbol{z} \right) - \ell \left( \mathcal{A} \left( (S_{0,\alpha}^t)^j \cup S_{i,1-\alpha} \right), \boldsymbol{z}_{0,j} \right) \right] \right| \\
&= \left| \mathbb{E}_{\boldsymbol{z}'_{0,j} \sim \mathcal{D}_0} \mathbb{E}_{\boldsymbol{z} \sim \mathcal{D}_0} \left[ \mathbb{E}_{S_{i,1-\alpha} \sim \mathcal{D}_i^{n(1-\alpha)}(S_{0,\alpha}^j)} \ell \left( \mathcal{A} \left( (S_{0,\alpha}^t)^j \cup S_{i,1-\alpha} \right), \boldsymbol{z} \right) \right. \right. \\
& \left. \left. \quad - \mathbb{E}_{S_{i,1-\alpha} \sim \mathcal{D}_i^{n(1-\alpha)}((S_{0,\alpha}^t)^j)} \ell \left( \mathcal{A} \left( (S_{0,\alpha}^t)^j \cup S_{i,1-\alpha} \right), \boldsymbol{z} \right) \right] \right| \\
& \quad + \left| \mathbb{E}_{\boldsymbol{z}'_{0,j} \sim \mathcal{D}_0} \mathbb{E}_{\boldsymbol{z} \sim \mathcal{D}_0} \left[ \mathbb{E}_{S_{i,1-\alpha} \sim \mathcal{D}_i^{n(1-\alpha)}(S_{0,\alpha}^j)} \ell \left( \mathcal{A} \left( (S_{0,\alpha}^t)^j \cup S_{i,1-\alpha} \right), \boldsymbol{z}_{0,j} \right) \right. \right. \\
& \left. \left. \quad - \mathbb{E}_{S_{i,1-\alpha} \sim \mathcal{D}_i^{n(1-\alpha)}((S_{0,\alpha}^t)^j)} \ell \left( \mathcal{A} \left( (S_{0,\alpha}^t)^j \cup S_{i,1-\alpha} \right), \boldsymbol{z}_{0,j} \right) \right] \right| \\
&\leq 2n(1-\alpha)\beta_n \left\| \text{TF} \left( (S_{0,\alpha}^t)^j \cup S_{i-1,1-\alpha} \right) - \text{TF} \left( (S_{0,\alpha}^t)^j \cup S'_{i-1,1-\alpha} \right) \right\|_{\ell_2} \\
&\lesssim 2n(1-\alpha)\beta_n \frac{2\widetilde{B}_W^L}{2n+1} \left[ ((1-\alpha)\widetilde{B}_W^L)^{i-1} + \alpha \frac{1 - ((1-\alpha)\widetilde{B}_W^L)^{i-1}}{1 - (1-\alpha)\widetilde{B}_W^L} \right] \\
&= 2n(1-\alpha)\beta_n \gamma_n^{i-1}. \quad (48)
\end{aligned}
$$

For the uniform stability parameter $\beta_n$ of SGD algorithm, we can derive the bound from Lemma 11. Substituting above results into Theorem 3, we obtain the following conclusion:

$$\left| R_{\mathcal{D}_0}(\mathcal{A}(\widetilde{S}_i)) - \widehat{R}_{\widetilde{S}_i}(\mathcal{A}(\widetilde{S}_i)) \right|$$

$$\leq \left( (1-\alpha)\beta_n \log(n(1-\alpha)) + \alpha \left( \beta_n + (1-\alpha)\rho^2\gamma_n^{i-1} \right) \log(n\alpha) \right) \log(\frac{1}{\delta})$$

$$+ \left( \sqrt{(1-\alpha)n}\alpha\beta_n + Mn^{-1/2}(\sqrt{1-\alpha} + \sqrt{\alpha}) \right) \sqrt{\log(\frac{1}{\delta})} + 2M \left( 1 - (1-\alpha)^i \right) \alpha^{-1} d_{\mathrm{TV}}(n)$$

$$\leq \beta_n \left[ (1-\alpha) \log(n(1-\alpha)) \log(\frac{1}{\delta}) + \alpha \log(n\alpha) \log(\frac{1}{\delta}) + \alpha\sqrt{(1-\alpha)n \log\frac{1}{\delta}} \right]$$

$$+ \gamma_n^{i-1}\alpha(1-\alpha)\rho^2 \log(n\alpha) \log(\frac{1}{\delta}) + n^{-1/2}M(\sqrt{1-\alpha} + \sqrt{\alpha})\sqrt{\log(\frac{1}{\delta})} + 2d_{\mathrm{TV}}(n)M \left( 1 - (1-\alpha)^i \right) \alpha^{-1}$$

$$\lesssim n^{-1/2} \log(n)M\rho^2\alpha\sqrt{1-\alpha} \log\frac{1}{\delta} + n^{-1}\rho^2((1-\alpha)\widetilde{B}_W^L)^i\alpha \log^2(n) \log\left(\frac{1}{\delta}\right)$$

$$+ n^{-1/4}\alpha^{-1}M \left( 1 - (1-\alpha)^i \right) \log(\frac{1}{\delta}). \tag{49}$$

$\square$

## A.9 PROOF OF THEOREM 4

In this section, we prove Theorem 4. The proof follows a similar approach to that of Theorem 3; however, it is more intricate due to the fact that the mixed dataset in Theorem 4 contains synthetic data from all previous generations. Each generation's synthetic dataset depends on the synthetic datasets of previous generations, leading to a more complex non-i.i.d. setting. Similar to Theorem 3, we begin by decomposing the generalization error into two components: the *Cumulative Distribution Shift Across Generations* and the *Generalization Error on Mixed Distributions*.

The main proof is as follows:

*Proof of Theorem 4.* We begin by decomposing the generalization error as follows:

$$\left| R_{\mathcal{D}_0}(\mathcal{A}(\widetilde{S}_i)) - \widehat{R}_{\widetilde{S}_i}(\mathcal{A}(\widetilde{S}_i)) \right| \leq \underbrace{\left| R_{\mathcal{D}_0}(\mathcal{A}(\widetilde{S}_i)) - R_{\widetilde{\mathcal{D}}_i}(\mathcal{A}(\widetilde{S}_i)) \right|}_{\text{Cumulative distribution shift across generations}} + \underbrace{\left| R_{\widetilde{\mathcal{D}}_i}(\mathcal{A}(\widetilde{S}_i)) - \widehat{R}_{\widetilde{S}_i}(\mathcal{A}(\widetilde{S}_i)) \right|}_{\text{Generalization error on mixed distributions}}.$$

**Upper Bounding Cumulative Distribution Shift Term**

For the term $\left| R_{\mathcal{D}_0}(\mathcal{A}(\widetilde{S}_i)) - R_{\widetilde{\mathcal{D}}_i}(\mathcal{A}(\widetilde{S}_i)) \right|$, we first note that $\widetilde{\mathcal{D}}_i = \frac{1}{1+i\lambda}\mathcal{D}_0 + \frac{\lambda}{1+i\lambda}\mathcal{D}_1 + \frac{\lambda}{1+i\lambda}\mathcal{D}_2 + ... + \frac{\lambda}{1+i\lambda}\mathcal{D}_i$. Therefore, we obtain:

$$\left| R_{\mathcal{D}_0}(\mathcal{A}(\widetilde{S}_i)) - R_{\widetilde{\mathcal{D}}_i}(\mathcal{A}(\widetilde{S}_i)) \right|$$

$$= \left| R_{\mathcal{D}_0}(\mathcal{A}(\widetilde{S}_i)) - \frac{1}{1+i\lambda}R_{\mathcal{D}_0}(\mathcal{A}(\widetilde{S}_i) - \frac{\lambda}{1+i\lambda}R_{\mathcal{D}_1}(\mathcal{A}(\widetilde{S}_1)) - ... - \frac{\lambda}{1+i\lambda}R_{\mathcal{D}_i}(\mathcal{A}(\widetilde{S}_i)) \right|$$

$$= \left| \frac{i\lambda}{1+i\lambda}R_{\mathcal{D}_0}(\mathcal{A}(\widetilde{S}_i) - \frac{\lambda}{1+i\lambda}R_{\mathcal{D}_1}(\mathcal{A}(\widetilde{S}_1)) - ... - \frac{\lambda}{1+i\lambda}R_{\mathcal{D}_i}(\mathcal{A}(\widetilde{S}_i)) \right|$$

$$\leq \frac{\lambda}{1+i\lambda} \left| R_{\mathcal{D}_0}(\mathcal{A}(\widetilde{S}_i) - R_{\mathcal{D}_1}(\mathcal{A}(\widetilde{S}_i)) \right| + ... + \frac{\lambda}{1+i\lambda} \left| R_{\mathcal{D}_0}(\mathcal{A}(\widetilde{S}_i) - R_{\mathcal{D}_i}(\mathcal{A}(\widetilde{S}_i)) \right|$$

$$\leq \frac{\lambda}{1+i\lambda} \sum_{j=1}^{i} \left| R_{\mathcal{D}_0}(\mathcal{A}(\widetilde{S}_i) - R_{\mathcal{D}_j}(\mathcal{A}(\widetilde{S}_i)) \right|. \tag{50}$$

Furthermore, we can further decompose it as follows:

$$\left| R_{\mathcal{D}_0}(\mathcal{A}(\widetilde{S}_i) - R_{\mathcal{D}_j}(\mathcal{A}(\widetilde{S}_i)) \right| \leq \left| R_{\mathcal{D}_0}(\mathcal{A}(\widetilde{S}_i)) - R_{\widetilde{\mathcal{D}}_{j-1}}(\mathcal{A}(\widetilde{S}_i)) \right| + \left| R_{\widetilde{\mathcal{D}}_{j-1}}(\mathcal{A}(\widetilde{S}_i)) - R_{\mathcal{D}_j}(\mathcal{A}(\widetilde{S}_i)) \right|. \tag{51}$$

By substituting inequality 51 into inequality 50, we obtain:

$$\left| R_{\mathcal{D}_0}(\mathcal{A}(\widetilde{S}_i)) - R_{\widetilde{\mathcal{D}}_i}(\mathcal{A}(\widetilde{S}_i)) \right|$$

$$\leq \frac{\lambda}{1 + i\lambda} \sum_{j=1}^{i} \left( \left| R_{\mathcal{D}_0}(\mathcal{A}(\widetilde{S}_i)) - R_{\widetilde{\mathcal{D}}_{j-1}}(\mathcal{A}(\widetilde{S}_i)) \right| + \left| R_{\widetilde{\mathcal{D}}_{j-1}}(\mathcal{A}(\widetilde{S}_i)) - R_{\mathcal{D}_j}(\mathcal{A}(\widetilde{S}_i)) \right| \right). \quad (52)$$

Thus, from equation 46 in the proof of Theorem 3 and lemma 12, we obtain:

$$\left| R_{\widetilde{\mathcal{D}}_{j-1}}(\mathcal{A}(\widetilde{S}_i)) - R_{\mathcal{D}_j}(\mathcal{A}(\widetilde{S}_i)) \right| \leq 2MTV \left( \mathbb{P}\left( \cdot \mid \widetilde{S}_{j-1} \right), \mathbb{P}_{\hat{\theta}}\left( \cdot \mid \widetilde{S}_{j-1} \right) \right)$$

$$\lesssim M n_{j-1}^{-1/4} \log n_{j-1} \log(1/\delta). \quad (53)$$

Incorporating inequality 53 into inequality 52, we arrive at:

$$|R_{\mathcal{D}_0}(\mathcal{A}(\widetilde{S}_i)) - R_{\widetilde{\mathcal{D}}_i}(\mathcal{A}(\widetilde{S}_i))|$$

$$\lesssim \frac{\lambda}{1 + i\lambda} \sum_{j=1}^{i} \left| R_{\mathcal{D}_0}(\mathcal{A}(\widetilde{S}_i)) - R_{\widetilde{\mathcal{D}}_{j-1}}(\mathcal{A}(\widetilde{S}_i)) \right| + \frac{\lambda}{1 + i\lambda} \sum_{j=0}^{i-1} M n_j^{-1/4} \log n_j \log(1/\delta). \quad (54)$$

Let $f(i) = \sum_{j=0}^{i-1} M n_j^{-1/4} \log n_j \log(1/\delta)$, Then, we obtain:

$$|R_{\mathcal{D}_0}(\mathcal{A}(\widetilde{S}_i)) - R_{\widetilde{\mathcal{D}}_i}(\mathcal{A}(\widetilde{S}_i))|$$

$$\lesssim \frac{\lambda}{1 + i\lambda} \left| R_{\mathcal{D}_0}(\mathcal{A}(\widetilde{S}_i)) - R_{\widetilde{\mathcal{D}}_{i-1}}(\mathcal{A}(\widetilde{S}_i)) \right| + ... + \frac{\lambda}{1 + i\lambda} \left| R_{\mathcal{D}_0}(\mathcal{A}(\widetilde{S}_i)) - R_{\widetilde{\mathcal{D}}_1}(\mathcal{A}(\widetilde{S}_i)) \right| + \frac{\lambda}{1 + i\lambda} f(i).$$

Similarly, we get:

$$|R_{\mathcal{D}_0}(\mathcal{A}(\widetilde{S}_i)) - R_{\widetilde{\mathcal{D}}_{i-1}}(\mathcal{A}(\widetilde{S}_i))|$$

$$\lesssim \frac{\lambda}{1 + (i-1)\lambda} \left| R_{\mathcal{D}_0}(\mathcal{A}(\widetilde{S}_i)) - R_{\widetilde{\mathcal{D}}_{i-2}}(\mathcal{A}(\widetilde{S}_i)) \right| + ... + \frac{\lambda}{1 + (i-1)\lambda} \left| R_{\mathcal{D}_0}(\mathcal{A}(\widetilde{S}_i)) - R_{\widetilde{\mathcal{D}}_1}(\mathcal{A}(\widetilde{S}_i)) \right|$$

$$+ \frac{\lambda}{1 + (i-1)\lambda} f(i-1).$$

Then, we have

$$|R_{\mathcal{D}_0}(\mathcal{A}(\widetilde{S}_i)) - R_{\widetilde{\mathcal{D}}_i}(\mathcal{A}(\widetilde{S}_i))| \lesssim \frac{\lambda}{1 + i\lambda} f(i) + \frac{\lambda}{1 + i\lambda} \frac{\lambda}{1 + (i-1)\lambda} f(i-1) +$$

$$(\frac{\lambda}{1 + i\lambda} + \frac{\lambda}{1 + i\lambda} \frac{\lambda}{1 + (i-1)\lambda})(\left| R_{\mathcal{D}_0}(\mathcal{A}(\widetilde{S}_i)) - R_{\widetilde{\mathcal{D}}_{i-2}}(\mathcal{A}(\widetilde{S}_i)) \right| + ... + \left| R_{\mathcal{D}_0}(\mathcal{A}(\widetilde{S}_i)) - R_{\widetilde{\mathcal{D}}_1}(\mathcal{A}(\widetilde{S}_i)) \right|).$$

$$(55)$$

Thus, by applying recursive techniques, we obtain the following result:

$$|R_{\mathcal{D}_0}(\mathcal{A}(\widetilde{S}_i)) - R_{\widetilde{\mathcal{D}}_i}(\mathcal{A}(\widetilde{S}_i))|$$

$$\lesssim \frac{\lambda}{1 + i\lambda} f(i) + \frac{\lambda}{1 + i\lambda} \frac{\lambda}{1 + (i-1)\lambda} f(i-1) + (\frac{\lambda}{1 + i\lambda} \frac{\lambda}{1 + (i-2)\lambda} + \mathcal{O}(\frac{1}{(1 + i\lambda)^2})) f(i-2)$$

$$+ ... + (\frac{\lambda}{1 + i\lambda} \frac{\lambda}{1 + \lambda} + \mathcal{O}(\frac{1}{(1 + i\lambda)})) f(1)$$

$$\lesssim \frac{\lambda}{1 + i\lambda} \left[ f(i) + \frac{\lambda}{1 + (i-1)\lambda} f(i-1) + \frac{\lambda}{1 + (i-2)\lambda} f(i-2) + ... + \frac{\lambda}{1 + \lambda} f(1) \right]$$

$$\lesssim M \log \frac{1}{\delta} \frac{\lambda}{1 + i\lambda} \left[ n_{i-1}^{-\frac{1}{4}} \log(n_{i-1}) + (1 + \frac{\lambda}{1 + (i-1)\lambda}) n_{i-2}^{-\frac{1}{4}} \log(n_{i-2}) + \right.$$

$$\left. (1 + \frac{\lambda}{1 + (i-1)\lambda} + \frac{\lambda}{1 + (i-2)\lambda}) n_{i-3}^{-\frac{1}{4}} \log(n_{i-3}) + ... + (1 + ... + \frac{\lambda}{1 + \lambda}) n_0^{-\frac{1}{4}} \log(n_0) \right]$$

$$\lesssim n^{-\frac{1}{4}} \log((1 + i\lambda)n) M \log \frac{1}{\delta}. \quad (56)$$

**Upper Bounding Generalization Error on Mixed Distributions Term**

Next, we turn our attention to the term $|R_{\widetilde{\mathcal{D}}_i}(\mathcal{A}(\widetilde{S}_i)) - \widehat{R}_{\widetilde{S}_i}(\mathcal{A}(\widetilde{S}_i))|$. Our primary objective is to establish a moment bound for this expression.

$$\left\| R_{\widetilde{\mathcal{D}}_i}(\mathcal{A}(\widetilde{S}_i)) - \widehat{R}_{\widetilde{S}_i}(\mathcal{A}(\widetilde{S}_i)) \right\|_p$$

$$= \left\| \frac{1}{1+i\lambda} R_{\mathcal{D}_0}(\mathcal{A}(\widetilde{S}_i)) + \frac{\lambda}{1+i\lambda} R_{\mathcal{D}_1}(\mathcal{A}(\widetilde{S}_i)) + \frac{\lambda}{1+i\lambda} R_{\mathcal{D}_2}(\mathcal{A}(\widetilde{S}_i))... + \frac{\lambda}{1+i\lambda} R_{\mathcal{D}_i}(\mathcal{A}(\widetilde{S}_i)) \right.$$

$$\left. - \frac{1}{(1+i\lambda)n} \sum_{\boldsymbol{z}_i \in S_0} \ell(\mathcal{A}(\widetilde{S}_i), \boldsymbol{z}_i) - \frac{1}{(1+i\lambda)n} \sum_{\boldsymbol{z}_i \in S_1} \ell(\mathcal{A}(\widetilde{S}_i), \boldsymbol{z}_i) - ... - \frac{1}{(1+i\lambda)n} \sum_{\boldsymbol{z}_i \in S_i} \ell(\mathcal{A}(\widetilde{S}_i), \boldsymbol{z}_i) \right\|_p$$

$$\leq \underbrace{\left\| \frac{1}{1+i\lambda} R_{\mathcal{D}_0}(\mathcal{A}(\widetilde{S}_i)) - \frac{1}{(1+i\lambda)n} \sum_{\boldsymbol{z}_i \in S_0} \ell(\mathcal{A}(\widetilde{S}_i), \boldsymbol{z}_i) \right\|_p}_{\text{Term 0}} + \underbrace{\left\| \frac{\lambda}{1+i\lambda} R_{\mathcal{D}_1}(\mathcal{A}(\widetilde{S}_i)) - \frac{1}{(1+i\lambda)n} \sum_{\boldsymbol{z}_i \in S_1} \ell(\mathcal{A}(\widetilde{S}_i), \boldsymbol{z}_i) \right\|_p}_{\text{Term 1}}$$

$$+ .. + \underbrace{\left\| \frac{\lambda}{1+i\lambda} R_{\mathcal{D}_i}(\mathcal{A}(\widetilde{S}_i)) - \frac{1}{(1+i\lambda)n} \sum_{\boldsymbol{z}_i \in S_i} \ell(\mathcal{A}(\widetilde{S}_i), \boldsymbol{z}_i) \right\|_p}_{\text{Term i}}. \tag{57}$$

Fixing $S_0, S_1, \ldots, S_{i-1}$, the data in $S_i$ are independent. Following a similar approach to the proof of Theorem 1, we utilize this property along with Lemma 8 to bound Term i. Consequently, from Equation 15 in the proof of Theorem 1, we obtain:

$$\left\| \frac{\lambda}{1+i\lambda} R_{\mathcal{D}_i}(\mathcal{A}(\widetilde{S}_i)) - \frac{1}{(1+i\lambda)n} \sum_{\boldsymbol{z}_i \in S_i} \ell(\mathcal{A}(\widetilde{S}_i), \boldsymbol{z}_i) \right\|_p \lesssim p \frac{\lambda}{1+i\lambda} \beta_{(1+i\lambda)n} \log(\lambda n) + \frac{M}{1+i\lambda} \sqrt{\frac{p\lambda}{n}}. \tag{58}$$

Next, we consider Term 0. Similar to Proof of Theorem 3, we first introduce a set of functions and apply Lemma 8 to bound Term 0. Specifically, we define $h_j(S)$, which serves a similar role to the $g_i$'s in Lemma 8, as follows:

$$h_j(S_0)$$
$$= \mathbb{E}_{\boldsymbol{z}'_{0,j} \sim \mathcal{D}_0} \left[ \mathbb{E}_{\boldsymbol{z} \sim \mathcal{D}_0} \ell \left( \mathcal{A} \left( S_0^j \cup S_1 \cup ... \cup S_i \right), \boldsymbol{z} \right) - \ell \left( \mathcal{A} \left( S_0^j \cup S_1 \cup ... \cup S_i \right), \boldsymbol{z}_{0,j} \right) \right], \tag{59}$$

where $\boldsymbol{z}_{0,j}$ denote the $j$-th data point in $S_0$, and $S_0^j$ represent the dataset obtained by replacing $\boldsymbol{z}_{0,j}$ with $\boldsymbol{z}'_{0,j}$. Moreover, following the procedure above, we observe that $|h_j| \leq M$ and $\mathbb{E}\left[ h_j \mid S_{0,\alpha}^{\backslash j} \right] = 0$. More intricately, we will now prove that $h_j$ exhibits a bounded difference. However, it is important to note that $S_1, \ldots, S_i$ all depend on $S_0$, so when a single data point in $S_0$ is changed, the corresponding datasets will also change. We denote these modified datasets as $S'_1, \ldots, S'_i$ and consequently, we have the following:

$$|h_j(S_0) - h_j(S_0^t)|$$
$$= |\mathbb{E}_{\boldsymbol{z}'_{0,j} \sim \mathcal{D}_0} \left[ \mathbb{E}_{\boldsymbol{z} \sim \mathcal{D}_0} \ell \left( \mathcal{A} \left( S_0^j \cup S_1 \cup ... \cup S_i \right), \boldsymbol{z} \right) - \ell \left( \mathcal{A} \left( S_0^j \cup S_1 \cup ... \cup S_i \right), \boldsymbol{z}_{0,j} \right) \right] |$$
$$- \mathbb{E}_{\boldsymbol{z}'_{0,j} \sim \mathcal{D}_0} \left[ \mathbb{E}_{\boldsymbol{z} \sim \mathcal{D}_0} \ell \left( \mathcal{A} \left( (S_0^t)^j \cup S'_1 \cup ... \cup S'_i \right), \boldsymbol{z} \right) - \ell \left( \mathcal{A} \left( (S_0^t)^j \cup S'_1 \cup ... \cup S'_i \right), \boldsymbol{z}_{0,j} \right) \right]$$
$$\leq 2\beta_{(1+i\lambda)n} \left( \|S_0^j - (S_0^t)^j\|_{\ell_2} + \|S_1 - S'_1\|_{\ell_2} + ... + \|S_i - S'_i\|_{\ell_2} \right). \tag{60}$$

Thus, by applying the recursive stability established in Theorem 2, it is important to first note that in Theorem 2, the mixed dataset is defined as $\widetilde{S}_j = \alpha S_0 + (1-\alpha)S_j$, whereas in this theorem, the mixed dataset is defined as $\widetilde{S}_i = \sum_{j=0}^i S_j$. Therefore, by following the proof steps outlined in Theorem 2, we can derive the following:

$$|h_j(S_0) - h_j(S_0^t)| \lesssim 2\beta_{(1+i\lambda)n} \left( i! \widetilde{B}_W^{iL} \right).$$

Thus, we apply lemma 8:

$$\left\| \sum_{j=1}^{n} h_j(S_0) \right\|_p \leq 12\sqrt{2}pn2\beta_{(1+i\lambda)n}\left(i!\widetilde{B}_W^{iL}\right)\log(n) + 4M\sqrt{pn}$$

$$\lesssim p\frac{\rho^2}{1+i\lambda}\left(i!\widetilde{B}_W^{iL}\right)\log(n(1+i\lambda)) + M\sqrt{pn}.$$

We observe that the difference between Term 0 and $\frac{1}{(1+i\lambda)n}\left\|\sum_{j=1}^{n}h_j(S_0)\right\|_p$ is negligible. Thus, we can bound Term 0 as follows:

$$\left\| \frac{1}{1+i\lambda}R_{\mathcal{D}_0}(\mathcal{A}(\widetilde{S}_i)) - \frac{1}{(1+i\lambda)n}\sum_{\boldsymbol{z}_i \in S_0}\ell(\mathcal{A}(\widetilde{S}_i), \boldsymbol{z}_i) \right\|_p$$

$$\lesssim p\frac{\rho^2}{(1+i\lambda)^2 n}\left(i!\widetilde{B}_W^{iL}\right)\log(n(1+i\lambda)) + \frac{1}{1+i\lambda}M\sqrt{p/n}. \tag{61}$$

Using the same method, for Term $j$, where $1 \leq j \leq i-1$, we can derive the following:

$$\left\| \frac{\lambda}{1+i\lambda}R_{\mathcal{D}_j}(\mathcal{A}(\widetilde{S}_i)) - \frac{1}{(1+i\lambda)n}\sum_{\boldsymbol{z}_i \in S_1}\ell(\mathcal{A}(\widetilde{S}_j), \boldsymbol{z}_i) \right\|_p$$

$$\lesssim p\frac{\rho^2}{(1+i\lambda)^2 n}\left(j!\widetilde{B}_W^{jL}\right)\log(n(1+i\lambda)) + \frac{1}{1+i\lambda}M\sqrt{p/n}. \tag{62}$$

In summary, we can finally bound the Generalization Error on the Mixed Distributions term as follows:

$$\left\| R_{\widetilde{\mathcal{D}}_i}(\mathcal{A}(\widetilde{S}_i)) - \widehat{R}_{\widetilde{S}_i}(\mathcal{A}(\widetilde{S}_i)) \right\|_p$$

$$\lesssim p\frac{\rho^2}{(1+i\lambda)^2 n}\log((1+i\lambda)n)i!\widetilde{B}_W^{(i+1)L} + \frac{Mi}{1+i\lambda}\sqrt{\frac{p}{n}}.$$

Then, according to Lemma 9, we obtain, with probability at least $1 - \delta$:

$$\left\| R_{\widetilde{\mathcal{D}}_i}(\mathcal{A}(\widetilde{S}_i)) - \widehat{R}_{\widetilde{S}_i}(\mathcal{A}(\widetilde{S}_i)) \right\|_p$$

$$\lesssim \frac{\rho^2}{(1+i\lambda)^2 n}\log((1+i\lambda)n)i!\widetilde{B}_W^{(i+1)L}\log\frac{1}{\delta} + \frac{Mi}{1+i\lambda}\sqrt{\frac{1}{n}\log\frac{1}{\delta}}.$$

Then, combine the above inequality with inequality 56, we obtain:

$$\left| R_{\mathcal{D}_0}(\mathcal{A}(\widetilde{S}_i)) - \widehat{R}_{\widetilde{S}_i}(\mathcal{A}(\widetilde{S}_i)) \right|$$

$$\lesssim n^{-\frac{1}{4}}\log((1+i\lambda)n)M\log\frac{1}{\delta} + \frac{\rho^2}{(1+i\lambda)^2 n}\log((1+i\lambda)n)i!\widetilde{B}_W^{(i+1)L}\log\frac{1}{\delta} + \frac{Mi}{1+i\lambda}\sqrt{\frac{1}{n}\log\frac{1}{\delta}}$$

$$\lesssim n^{-\frac{1}{2}}\frac{Mi}{1+i\lambda}\sqrt{\log\frac{1}{\delta}} + n^{-1}\frac{\rho^2}{(1+i\lambda)^2}\log((1+i\lambda)n)i!\widetilde{B}_W^{(i+1)L}\log\frac{1}{\delta}$$

$$+ n^{-\frac{1}{4}}\log((1+i\lambda)n)M\log\frac{1}{\delta}.$$

The proof is complete.

$\square$

# B EXPERIMENTS

In this section, we present some experimental results. Specifically, we trained transformer models to in-context learn linear functions within STLs.

In these experiments, we considered the class of linear functions:

$$\mathcal{F} = \left\{ f \mid f(\boldsymbol{x}) = \boldsymbol{w}^\top \boldsymbol{x}, \boldsymbol{w} \in \mathbb{R}^d \right\},$$

in $d = 5$ dimensions. We sampled $\boldsymbol{x}_1, \ldots, \boldsymbol{x}_k, \boldsymbol{x}_{\text{query}}$, and $\boldsymbol{w}$ independently from the isotropic Gaussian distribution $\mathcal{N}(0, I_d)$. For each $x_i$, we computed $y_i = \boldsymbol{w}^\top \boldsymbol{x}_i$ and constructed the prompt as:

$$P = (\boldsymbol{x}_1, y_1, \boldsymbol{x}_2, y_2, \ldots, \boldsymbol{x}_k, y_k, \boldsymbol{x}_{\text{query}}).$$

We employed a 12-layer, 8-head GPT-2 model with a hidden size of 256, trained on an $\mathbb{R}^5$ linear regression task with 40 in-context examples. Two cases were considered:

- **Mixed Case:** Fresh data and generated data were mixed in a 0.5 ratio.
- **Full Synthetic Case:** No fresh data was used.

The results of these experiments are summarized below:

| Loop | 1 | 2 | 3 | 4 | 5 | 6 |
|---|---|---|---|---|---|---|
| Full Synthetic | 0.3817 | 1.4975 | 1.5396 | 2.0836 | 2.3912 | 2.8764 |
| Mixed | 0.3817 | 0.4208 | 0.4391 | 0.4503 | 0.4641 | 0.4702 |

As observed, the error accumulates progressively with more self-consuming loops, particularly in the full synthetic case, where the error grows rapidly. In contrast, maintaining a constant-sized proportion of real data effectively reduces the loss, which is consistent with our theoretical findings.

