# OpenReview forum: "A Theoretical Perspective: How to Prevent Model Collapse in Self-consuming Training Loops"
_ICLR.cc/2025/Conference — ICLR 2025 Poster_

### Official Review · Reviewer_navH · 2024-10-31

**Soundness:** 3
**Presentation:** 2
**Contribution:** 2
**Rating:** 6
**Confidence:** 3

**Summary:**

The authors attempt to present a theoretical framework to describe the generalization error of models when they are trained on synthetic data. They are motivated by the observation that multiple workflows presented in the literature on model collapse / self-consuming loops arrive at different conclusions. They introduce the notion of recursive stability, inspired by past notions of algorithmic stability like uniform stability, using which they quantify the propagation of error as model-fitting iterations progress. Finally, using these tools, they perform an analysis of in-context learning in transformers, obtaining generalization error estimates that describe the impact of real data in the model-fitting loop.

**Strengths:**

Through their notion of recursive stability, the authors have considered a much more general class of algorithms / models than what the prior literature has described. They have been able to go beyond explicit model calculations or estimation strategies. This notion allows them to decompose the generalization error and explicitly attribute each term to a specific statistical reason. Interestingly, the authors are also able to identify, through these upper bounds on the generalization gap, the effect of proportion of real data in the STL.

**Weaknesses:**

The authors have tried to be too general - this has significantly affected the readability of the paper. There are no examples of models provided. What models are recursively stable? How do we know the TV bounds will be of the same order? The paper really needs some examples for clarity. As of now, it is too hard to parse. Understandably it is a math paper, but a lot of what is written needs to be appropriately motivated as the premise addressed by the authors is an important empirical problem (the problem of model collapse) which the authors are tackling theoretically. I would recommend the authors to rethink their presentation for greater impact.

Further, to me the bounds derived are a rather straightforward application of "working with upper bounds" when the correct assumptions are in place. Thus, I feel the technical novelty is limited, and I am not sure if we are learning something substantially new from these upper bounds, than what we already knew about STLs.

Finally, the literature review is a little too crude. The authors should connect their results with those obtained by Gerstgrasser et. al. (2024), Dohmatob et. al. (2024), Alemohammad et. al. (2023), etc. because we know how specific workflows function already, and it is important to clarify that the present authors are not contradicting any existing result.

**Questions:**

1. In line 195, does $S_0'$ depend on $j$ or not? (the previous sentence suggests it does) If it does, then the provided formula doesn't make sense.

2. With regards to equation 1, let's consider $\alpha=0$ i.e. no real data is used. What is the behavior of the error gap? It seems to me that only the last term will matter and using the fact that $\lim_{\alpha\to0} (1 - (1-\alpha)^i)/\alpha = i$, this predicts a linear rate of increase of the generalization gap. Is this correct?

3. Following the point I make in 2 above, it seems to me that the discussion in lines 257-260 needs to change, because when $\alpha$ is too small, yes $\alpha^{-1}$ gets large but it's multiplied by $1-(1-\alpha)^i$ which is small too. They balance each other out exactly giving $i$.

4. I don't understand the final conclusion of Theorem 4. There are too many terms - which one matters the most? Gerstgrasser et. al. (2024), Marchi et. al. (2024) and Seddik et. al. (2024) have shown that the gap is like $\sum_i 1/i^2$, which is kind of similar to the second term in the upper bound, but it has $n^{-1}$ in front which makes me think that this term is a higher order term. The presented bound makes me think that only the first term $O(n^{-1/4})$ matters, but then the rate $\log(i)$ (essentially) seemingly contradicts past findings. I would urge the authors to actually connect this part with the more recent literature on accumulating data for avoiding model collapse.

---

> ### Author Response · Authors · 2024-11-21
> **Clarifications on Model Examples and TV Distance Bounds**
>
> Thank you for your thorough review and constructive comments. All your concerns have been carefully addressed as below. The paper has been thoroughly revised, with the revised sections highlighted in blue for clarity. We sincerely hope our responses fully address your questions.
>
> ---
>
> **Q1: There are no examples of models provided. What models are recursively stable? How do we know the TV bounds will be of the same order? The paper really needs some examples for clarity.**
>
> ---
>
> **A:** We appreciate the reviewer’s insightful feedback. Below, we address the concerns raised regarding examples of recursively stable models, the TV distance bounds, and the need for clarity through examples:
>
> ### 1. What models are recursively stable?
> Recursive stability, as introduced in our paper, measures the stability of generative models within STLs in response to perturbations in the initial data. This property depends on both the model architecture and the ratio $\alpha$ of real to synthetic data within STLs.
>
> 1. **Model Architectures:**
>    Generative models satisfying recursive stability must maintain distributional fidelity even under perturbations in the training dataset. This depends on the model's capacity to approximate distributions effectively. In addition to rigorously proving recursive stability for transformers in in-context learning within this paper, other examples include GANs ([1] Farnia 2021) and Gaussian Mixture Models (GMMs). In the final part of this response, we provide a detailed example demonstrating how a GMM satisfies recursive stability.
>
> 2. **Effect of $\alpha$:**
>    In addition to the impact of model architecture, the ratio $\alpha$ in STLs plays a critical role. The specific value of $\alpha$ required to control recursive stability depends on the stability capacity of the chosen generative model. As shown in Theorem 2, when $\alpha \to 0$, the recursive stability of transformers increases rapidly. However, when $\alpha$ is set to $1 - \tilde{B}_W^{-L}$, recursive stability can be effectively controlled. In addition, we demonstrate in the subsequent GMM example that a constant value of $\alpha$ (e.g., 0.1) is sufficient to control recursive stability.
>
> ### 2. Why are TV bounds assumed to be of the same order?
> The assumption that TV bounds will be of the same order is a common and well-established hypothesis, as also adopted in ([2] Fu et al. 2024). In generalization theory, the TV distance bound for generative models with the same architecture is primarily determined by the size of the training dataset $n$ ([3] Li 2023, [4] Zhang 2023). Following the settings of prior works ([2] Fu et al. 2024, [5] Dohmatob et al. 2024, [6] Seddik et al. 2024), where the training dataset size $n$ is assumed to remain consistent across generations, it is reasonable to extend this assumption and infer that the TV bounds will likewise remain of the same order.

---

> ### Author Response · Authors · 2024-11-21
> **Clarity through Examples**
>
> ### 3. Clarity through Examples
> To address the reviewer’s concern, we apply our framework to a tractable example—a GMM in a binary classification task. We adopt the setup from prior works [5,6] and consider a binary classification task where $Y =\\{-1, 1\\}$. Given a vector $\mu \in \mathbb{R}^d$ with $\\|\mu\\|_2 = 1$ and noise variance $\sigma^2 > 0$, the data distribution is specified as follows: $y \sim \text{uniform}\\{-1, 1\\}$ and $x \mid y \sim \mathcal{N}(y \mu, \sigma^2 I_d)$. We define the conditional generative model using parameters $\mu_y$ and $\sigma_k^2$, where $y \in \\{-1, 1\\}$ and $k \in [d]$. For $n$ data points, let $n_y$ represent the number of samples in class $y$. The means $\hat{\mu}_y$ of the Gaussian mixture model are estimated as
>
> $$\frac{\sum_{y_i = y} x_i}{n_y},$$
>
> while the variances $\hat{\sigma}_k^2$ are calculated as
>
> $$\sum_y \frac{n_y}{n} \frac{\sum_{y_i = y} (x_{ik} - \hat{\mu}_{yk})^2}{n_y - 1}.$$
>
> Then we can generate new samples from the distribution: $y \sim \text{uniform}\\{-1, 1\\}$ and $x \mid y \sim \mathcal{N}(\hat{\mu}_y, \Sigma)$.
>
>  Additionally, the learning algorithm functions as a linear classifier, parameterized by $\theta \in \mathbb{R}^d$, with predictions given by:
> $$
> \hat{y} = \text{sign}(\theta^\top \mathbf{x}).
> $$
> The loss function is defined as:
> $$
> \ell(\theta, (x, y)) = \frac{1}{2\sigma^2} (x - y\theta)^\top (x - y\theta).
> $$
> Thus, the output is:
> $$
> \hat{\theta} = \frac{1}{m} \sum_{i=1}^m y_i x_i.
> $$
>
> In this setting, we demonstrate recursive stability for the Gaussian mixture model as follows:
>
> ---
> **Theorem (Recursive Stability)**: Let $S_0, S_0'$ denote two initial real datasets differing by a single example. Let $n$ represent the sample size of the mixed dataset $\tilde{S}_j$, where $\tilde{S}_j = \alpha S_0 + (1 - \alpha) S_j$ for $1 \leq j \leq i$. Choose $m = \mathcal{O}(\sqrt{n})$. Consider the previously described sampling and learning steps, where real data samples are drawn from the Gaussian Mixture Model distribution $\mathcal{D}$, and the synthetic data for the $i$-th generation is generated from the learned Gaussian Mixture distribution of the $i$-th generation. Then with probability at least $1-\delta$, we have:
> $$
> \gamma_n^{i,\alpha} \lesssim n^{-1/2} \alpha^{-1}(1-(1-\alpha)^i)\log (nd/\delta),
> $$
> where the measure for the recursive stability parameter is taken as the KL divergence.
>
> ---
>
> As $\alpha$ approaches 0, indicating that no real data is incorporated during each generation of training, we observe:
> $$
> \gamma_n^i \lesssim i n^{-1/2} \log \frac{nd}{\delta},
> $$
> which suggests a linear accumulation of errors. This finding aligns closely with the theoretical insights presented in [3,7], where a Gaussian model trained without real data demonstrated a linear divergence in variance. Thus, this underscores the validity of our theoretical results, confirming that the derived bound is meaningful and not vacuous.
>
> Moreover, by leveraging the generalization error bound established in Theorem 1, we derive the following:
>
> ---
>
> **Theorem(Generalization Error Bound)**: Consider the Gaussian Mixture Model in the setting outlined above. Let $n$ represent the sample size of the mixed dataset $\tilde{S}_j$, where $\tilde{S}_j = \alpha S_0 + (1 - \alpha) S_j$ for $1 \leq j \leq i$. Suppose the loss function is defined above. Let $\mathcal{A}(\tilde{S}_i)$ denote the output of applying the linear classifier described above to the mixed dataset $\tilde{S}_i$. Then, for any $\delta \in (0,1)$, with probability at least $1-\delta$, the following holds:
>
> $$
>  |Generalization| \lesssim n^{-1/2}(d+\log(n/\delta))\log n\log (1/\delta)
> +n^{-1/4}(1-(1-\alpha)^i)\alpha^{-1}(d+\log (n/\delta))\sqrt{d\log(nd/\delta)}.
> $$
>
> ---
>
> We observe that when $\alpha$ is set to a constant (e.g., $\alpha = 0.1$), the generalization error can be effectively controlled, preventing model collapse. This result aligns with the experimental findings in [2] for Gaussian models.
>
> The above discussions have been carefully added to our paper.

---

> ### Author Response · Authors · 2024-11-21
> **Technical Novelty and Advancements in Our Framework**
>
> **Q2: Further, to me the bounds derived are a rather straightforward application of ``working with upper bounds'' when the correct assumptions are in place. Thus, I feel the technical novelty is limited, and I am not sure if we are learning something substantially new from these upper bounds, than what we already knew about STLs.**
>
> ---
>
> **A:** Thank you for your feedback and for raising concerns about the technical novelty of our work. We would like to highlight how our contributions go beyond previous theoretical studies and address substantially more complex and realistic scenarios.
>
> ### 1. From Traditional Models to Generative Models
> As you have kindly noted, “the authors have considered a **much more general class** of algorithms/models than what the prior literature has described." Unlike prior theoretical results that focus on linear regression (e.g., [5] Dohmatob et al. 2024, [7] Gerstgrasser et al. 2024) and Gaussian models (e.g., [9] Alemohammad et al. 2023, [11] Shumailov 2024), our theoretical findings are applicable to a broader class of generative models, such as the **transformer-based architectures** commonly used in large models. This general setting results in significant technical challenges, including the analysis of **nonconvex optimization** for **nonlinear attention models** under recursive training, which drives us to propose a **novel analytical tool—recursive stability**.
>
> ### 2. From i.i.d Data to Non-i.i.d Data
> To simplify the mathematical analysis of models and algorithms, many theoretical analyses rely on the i.i.d data assumption, such as [12] Bousquet et al. In our manuscript, we tackle this challenge by leveraging additional conditional independence properties. This enables our theoretical results to handle **mixed data distributions** in recursive training.
>
> ### 3. From Statistical Error to the Integration of Statistical, Functional, and Optimization Errors
> Previous works primarily focus on statistical approximation error while **ignoring** the functional approximation error arising from generative model training and the optimization error associated with learning algorithms (e.g., [5] Dohmatob et al. 2024, [7] Gerstgrasser et al. 2024, [6] Seddik et al. 2024). In contrast, our work introduces the intriguing notion of **recursive stability** and performs a detailed analysis of the **dynamic interactions** of complex generative models and learning algorithms in STLs. By doing so, we elucidate how **statistical approximation error, functional approximation error, and optimization error** interact and accumulate during the training process.
>
> ### 4. From Asymptotic Analysis to Finite Analysis
> Unlike [8] Marchi et al. 2024, which relies on the assumption of an **infinite** number of training generations for their asymptotic analysis, we analyze **finite generations**, which is more practical and relevant. Most experimental setups in the literature limit generations to fewer than 10 (e.g., [11] Shumailov 2024), making our results more applicable to real-world scenarios.
>
> ---
>
> ### Technical Innovations
> **Recursive Stability:**
> We introduce the intriguing notion of recursive stability to quantify the differences in a complex generative model’s outputs after STLs when small perturbations are applied to the initial real dataset. Notably, this concept is significantly more intricate than algorithm stability due to the inherent challenges posed by the recursive structure of STLs.
>
> By addressing these challenges, our results go beyond merely "working with upper bounds" and provide **new insights** into the interplay between real and synthetic data, the architecture of generative models, and the behavior of learning algorithms under recursive training schemes.
>
> ---
>
> We believe our work introduces substantial technical novelty by combining recursive stability with comprehensive error analysis in a realistic and challenging setting, offering significant advancements over existing theoretical studies. Thank you again for your feedback, which allowed us to better articulate the contributions and significance of our work.

---

> ### Author Response · Authors · 2024-11-21
> **Strengthening the Literature Review: Connecting to Recent Works**
>
> **Q3: The literature review is a little too crude. The authors should connect their results with those obtained by Gerstgrasser et al. (2024), Dohmatob et al. (2024), Alemohammad et al. (2023), etc. because we know how specific workflows function already, and it is important to clarify that the present authors are not contradicting any existing result.**
>
> ---
>
> **A:** Thank you for your valuable feedback. Based on your suggestion, we have expanded the literature review by including detailed comparisons with recent works in the remarks following Theorem 1 and Theorem 4. These additions highlight how our results align with or extend previous findings, ensuring that our work is properly situated within the context of existing research. Below is a summary of the changes:
>
> ---
>
> ### 1.  Additions to the Remark Following Theorem 1
>
> `` Dohmatob et al. (2024) examined a linear regression setting, focusing solely on statistical approximation error without addressing the functional approximation error described in Shumailov (2024). They did not consider incorporating real data to prevent collapse and demonstrated a linear dependency of degradation on the generation number in the case of fully synthetic data. Similarly, Alemohammad et al. (2023) and Shumailov (2024) provided theoretical insights using simple Gaussian models without incorporating real data, proving that the variance diverges linearly with the generation number. Seddik et al. (2024) explored a linear softmax classifier and, while also neglecting functional approximation error, demonstrated that adding real data can mitigate model collapse. Marchi et al. (2024) used asymptotic analysis to study parameter variance, assuming an infinite number of training generations and considering scenarios where the generative model is controlled via a "temperature'' parameter. They proved that parameter variance is bounded under these conditions.
>
> In contrast, our work addresses a much more complex and realistic scenario by introducing the novel concept of **recursive stability** and providing the **first** generalization analysis for STLs. Our analysis accounts for **statistical approximation error, functional approximation error, and optimization error** during the training of generative models. Unlike the settings explored in prior theoretical works, such as linear regression (Dohmatob et al. 2024, Gerstgrasser et al. 2024), Gaussian models (Alemohammad et al. 2023, Shumailov 2024), or asymptotic assumptions (Marchi et al. 2024), our framework accommodates more complex generative model architectures, such as transformers. Specifically, we reveal how both **model architecture** and **the ratio** of real to synthetic data influence the success of STLs. For example, in Theorem 3, we demonstrate how our general generalization bound applies to transformer-based generative models, providing a theoretical framework that aligns with practical and more sophisticated use cases.
>
> Additionally, while Marchi et al. (2024) assumed an **infinite number** of training generations for their asymptotic analysis, we consider **finite generations**, which is more practical since most experimental setups limit generations to fewer than 10 (as noted in Shumailov 2024). Moreover, our results confirm that when $\alpha = 0$ (i.e., no real data is used), the last term in our bound, representing the Cumulative Distribution Shift ($d_{\text{TV}}(n) M (1 - (1 - \alpha)^i) \alpha^{-1}$), grows linearly. This finding aligns with the theoretical results of Dohmatob et al. (2024), Alemohammad et al. (2023), Shumailov (2024), and Fu et al. (2024). Furthermore, we show that introducing even a constant proportion of real data significantly mitigates model collapse, aligning with experimental findings by Alemohammad et al. (2023) and Bertrand et al. (2024).''
>
> ---
>
> ### 2.  Additions to the Remark Following Theorem 4
>
> ``Gerstgrasser et al. (2024) also explored the use of accumulating data to prevent model collapse. They considered a linear regression setting without accounting for the dynamic process of training generative models, focusing solely on statistical approximation error. They demonstrated that under the assumption of fixed synthetic data quality matching the original real data, statistical approximation error can be controlled.
>
> By contrast, our work addresses a much more complex and realistic scenario, incorporating the dynamic behavior of transformer-based generative models, learning algorithms, and both statistical and functional approximation errors. Additionally, we allow for dynamic regulation of synthetic data size via a $\lambda$ coefficient, enabling us to identify the optimal synthetic dataset size for avoiding model collapse in these more challenging settings.''

---

> ### Author Response · Authors · 2024-11-21
> **Clarification on Ambiguities in Definitions and Error Behavior**
>
> **Q4: In line 195, does $S'_0$ depend on $j$ or not? (The previous sentence suggests it does.) If it does, then the provided formula doesn't make sense.**
>
> ---
>
> **A:** Thank you for pointing out the potential ambiguity in our definition on line 195. To clarify, $S'_0$ does not depend on $j$. Our intent was to state that for all datasets $S_0$ and $S'_0$ such that $S_0$ and $S'_0$ differ by a single example, we use the recursive stability parameter to quantify the difference in outputs produced by the generative model when $S_0$ and $S'_0$ are provided as the initial training datasets.
>
> To address this ambiguity, we have revised the definition on line 195 as follows:
>
> **Definition 2. (Recursive Stability)**
> Let $S_0$ represent the original real dataset, and $S'_0$ denote a dataset differing from $S_0$ by a single example. A generative model $\mathcal{G}$ is said to be recursively $\gamma_n^{i,\alpha}$-stable with respect to the distance measure $d$ after the $i$-th generation of STLs, where the ratio of real to synthetic data is set to $\alpha$, if the following condition holds:
> $$
> \forall S_0, S'_0 \in \mathbb{Z}^n, \quad d\left(\mathcal{G}^{(i)}(S_0), \mathcal{G}^{(i)}(S'_0)\right) \leq \gamma_n^{i,\alpha}.
> $$
>
> This revised definition explicitly removes any ambiguity regarding $j$ and ensures that the recursive stability parameter is clearly defined. We appreciate your attention to this detail and have updated the manuscript accordingly.
>
> ---
>
> **Q5: With regards to equation 1, let's consider $\alpha = 0$, i.e., no real data is used. What is the behavior of the error gap? It seems to me that only the last term will matter and using the fact that $\lim_{\alpha \to 0} \frac{(1 - (1 - \alpha)^i)}{\alpha} = i$, this predicts a linear rate of increase of the generalization gap. Is this correct? Following the point I make above, it seems to me that the discussion in lines 257-260 needs to change, because when $\alpha$ is too small, yes $\alpha^{-1}$ gets large but it's multiplied by $1 - (1 - \alpha)^i$, which is small too. They balance each other out exactly giving $i$.**
>
> ---
>
> **A:** Thank you for your detailed observation and insightful question. Your understanding is absolutely correct!
>
> In the context of Equation 1, our general generalization bound, when we consider $\alpha = 0$ (i.e., no real data is used), it indeed holds that $\lim_{\alpha \to 0} \frac{(1 - (1 - \alpha)^i)}{\alpha} = i$. As a result, the last term, representing the Cumulative Distribution Shift,
> $$
> d_{\text{TV}}(n) M \left(1 - (1 - \alpha)^i\right) \alpha^{-1},
> $$
> grows linearly with $i$, thereby becoming the dominant term in our general generalization bound.
>
> This finding aligns closely with prior theoretical results:
> 1. **[5] Dohmatob et al. 2024:** Considers a linear regression setting and shows a similar linear dependency of degradation on the generation number.
> 2. **[11] Shumailov et al. 2024:** Demonstrates using a one-dimensional Gaussian model that the variance diverges linearly when no real data is incorporated.
> 3. **[2] Fu et al. 2024:** Proves that for diffusion models under fully synthetic scenarios, the distributional shift error accumulates linearly with the number of generations.
>
> These results further validate the correctness and applicability of our theoretical bounds.
>
> In response to your comment, we have revised the discussion in lines 257-260 to better reflect this understanding. Specifically, we have clarified:
>
> ``When $\alpha \to 0$, we observe that $\frac{(1 - (1 - \alpha)^i)}{\alpha} \to i$, leading to a linear accumulation of errors due to the Distribution Shift, making it increasingly challenging to control the overall error. This observation aligns with the theoretical results reported in Shumailov et al. 2024, Dohmatob et al. 2024, and Fu et al. 2024. ''
>
>
> We appreciate your valuable feedback, which has helped us refine the presentation of our results. Thank you again!

---

> ### Author Response · Authors · 2024-11-21
> **Understanding Key Terms in Theorem 4 and Connections to Related Work**
>
> **Q6: I don't understand the final conclusion of Theorem 4. There are too many terms - which one matters the most? Gerstgrasser et al. (2024), Marchi et al. (2024), and Seddik et al. (2024) have shown that the gap is like $\sum_i \frac{1}{i^2}$, which is kind of similar to the second term in the upper bound, but it has $n^{-1}$ in front which makes me think that this term is a higher order term. The presented bound makes me think that only the first term $\mathcal{O}(n^{-1/4})$ matters, but then the rate $\log(i)$ seemingly contradicts past findings. I would urge the authors to connect this part with the more recent literature on accumulating data for avoiding model collapse.**
>
> ---
>
> **A:** Thank you for your thoughtful comments. Below, we address your concerns regarding the terms in Theorem 4, their connection to related works, and the changes we made to incorporate this feedback.
>
> ---
>
> ### 1. Clarification on the Terms in Theorem 4
>
> In Theorem 4, the generalization bound is expressed as:
> $$
> |Generalization| \lesssim n^{-1/4}\log ((1+i\lambda)n)+ n^{-1}\frac{\rho^2}{(1+i\lambda)^2} \log ((1+i\lambda) n)i!\widetilde{B}_W^{(i+1)L}+n^{-1/2}\frac{Mi}{1+i\lambda}.
> $$
>
> - **Cumulative Distribution Shift Term:**
>   The first term, $\mathcal{O}(n^{-1/4}\log((1+i\lambda) n))$, reflects the Cumulative Distribution Shift. This arises because generative models cannot perfectly fit the training distribution, leading to distributional drift with each generation. The magnitude of this term heavily depends on the model's capacity to approximate the data distribution. In this theorem, where the generative model is a transformer and the learning algorithm is SGD, the Cumulative Distribution Shift term dominates. Importantly, this term grows with the number of generations $i$ at a rate of $\mathcal{O}(\log(i))$. **This growth does not contradict past findings but rather aligns with the intuition that each generation introduces incremental drift**, which accumulates over time, as also stated by [2] Fu et al. (2024). Similar behavior has been observed in $\Delta\text{Bias}$ terms discussed in [5] Dohmatob et al. (2024), where it was also shown to grow with $i$.
>
> - **Generalization Error on Mixed Distributions:**
>   The second and third terms capture the Generalization Error on Mixed Distributions, which depends on the stability of the learning algorithm and the recursive stability of the generative model. These terms are influenced by both the choice of learning algorithm and the architecture of the generative model. For example, if a less stable learning algorithm is used, these terms could dominate the generalization bound.
>
> ---
>
> ### 2. Connection to Related Literature
>
> We appreciate the references to [7] Gerstgrasser et al. (2024), [8] Marchi et al. (2024), and [6] Seddik et al. (2024), which you mentioned in relation to terms involving $\sum_i \frac{1}{i^2}$. We note the following distinctions:
>
> - **[7] Gerstgrasser et al. (2024):**
>   Gerstgrasser et al. analyzed a linear regression setting without considering the dynamic process of training generative models. Their analysis focuses on statistical approximation error, assuming that synthetic data generated in the $i$-th generation occupies a proportion of $1/i$ in the training set. Under this assumption and using **squared error**, they derive a term involving $\sum_i \frac{1}{i^2}$. In contrast, our work accounts for both statistical approximation error and functional approximation error, as introduced by Shumailov (2024), as well as optimization error arising in transformer training. Furthermore, our second term includes a component resembling $ \frac{1}{(1+i\lambda)^2}$, which arises because, in the $i$-th generation, the synthetic data constitutes $1/(1+i\lambda)$ of the training set, and the **stability** of the learning algorithm is of the order $1/((1+i\lambda)n)$. The product of these factors naturally results in the $\frac{1}{(1+i\lambda)^2}$ term, which is distinct from their result.
>
> - **[8] Marchi et al. (2024):**
>   The $\sum_i \frac{1}{i^2}$ term in Marchi et al. arises under a completely different context. **They assume the dataset size grows with $i$ (i.e., $i$ represents dataset size, not generation number as in our work or [7] Gerstgrasser et al.).** Their analysis focuses on asymptotic parameter variance under the assumption of infinite generations. By contrast, our results do not rely on asymptotic analysis and apply to finite numbers of generations while considering a broader set of error sources.
>
> - **[6] Seddik et al. (2024):**
>   We could not identify terms resembling $\sum_i \frac{1}{i^2}$ in their work. Their focus appears unrelated to the setting of our Theorem 4.
>
> Overall, while the $\sum_i \frac{1}{i^2}$ terms in [7] Gerstgrasser et al. and [8] Marchi et al. appear superficially similar to the $\frac{1}{(1+i\lambda)^2}$ term in our bound, they are derived under **different assumptions and address fundamentally different problems**.

---

> ### Author Response · Authors · 2024-11-21
> **Revised Discussion in the Paper**
>
> ### 3. Revised Discussion in the Paper
>
> To connect our work more explicitly to recent literature on accumulating data for avoiding model collapse, we have added the following discussion to the paper:
>
>
> ``Gerstgrasser et al. (2024) also explored the use of accumulating data to prevent model collapse. They considered a linear regression setting without accounting for the dynamic process of training generative models, focusing solely on statistical approximation error. They demonstrated that under the assumption of fixed synthetic data quality matching the original real data, statistical approximation error can be controlled.
>
> By contrast, our work addresses a much more complex and realistic scenario, incorporating the dynamic behavior of transformer-based generative models, learning algorithms, and both statistical and functional approximation errors. Additionally, we allow for dynamic regulation of synthetic data size via a $\lambda$ coefficient, enabling us to identify the optimal synthetic dataset size for avoiding model collapse in these more challenging settings.''
>
>
> We hope this revision addresses your concerns and provides a clearer connection to existing literature while emphasizing the unique contributions of our work. Thank you for your valuable feedback!
>
>
> Reference:
>
> $\left[1\right]$ Farnia F, Ozdaglar A. Train simultaneously, generalize better: Stability of gradient-based minimax learners[C]//International Conference on Machine Learning, 2021.
>
> $\left[2\right]$ Fu S, Zhang S, Wang Y, et al. Towards Theoretical Understandings of Self-Consuming Generative Models[C]//Forty-first International Conference on Machine Learning.
>
> $\left[3\right]$ Li P, Li Z, Zhang H, et al. On the generalization properties of diffusion models[J]. Advances in Neural Information Processing Systems, 2023.
>
> $\left[4\right]$ Zhang Y, Zhang F, Yang Z, et al. What and how does in-context learning learn? bayesian model averaging, parameterization, and generalization[J], 2023.
>
> $\left[5\right]$ Dohmatob E, Feng Y, Kempe J. Model collapse demystified: The case of regression[J]. arXiv, 2024.
>
> $\left[6\right]$ Seddik M E A, Chen S W, Hayou S, et al. How bad is training on synthetic data? a statistical analysis of language model collapse[J]. arXiv, 2024.
>
> $\left[7\right]$ Gerstgrasser M, Schaeffer R, Dey A, et al. Is model collapse inevitable? breaking the curse of recursion by accumulating real and synthetic data[J]. arXiv, 2024.
>
> $\left[8\right]$ Marchi M, Soatto S, Chaudhari P, et al. Heat Death of Generative Models in Closed-Loop Learning[J]. arXiv, 2024.
>
> $\left[9\right]$ Alemohammad S, Casco-Rodriguez J, Luzi L, et al. Self-Consuming Generative Models Go MAD[C]//The Twelfth International Conference on Learning Representations.
>
> $\left[10\right]$ Zheng C, Wu G, Li C. Toward understanding generative data augmentation[J]. Advances in neural information processing systems, 2023.
>
> $\left[11\right]$ Shumailov I, Shumaylov Z, Zhao Y, et al. AI models collapse when trained on recursively generated data[J]. Nature, 2024.
>
> $\left[12\right]$ Bousquet O, Klochkov Y, Zhivotovskiy N. Sharper bounds for uniformly stable algorithms[C]//Conference on Learning Theory. PMLR, 2020: 610-626.

---

> > ### Comment · Reviewer_navH · 2024-11-24
> > **Reviewer response**
> >
> > I thank the authors for a detailed response. Based on this, I have raised the score to 6.

---

> > > ### Author Response · Authors · 2024-11-25
> > > **Thank you**
> > >
> > > Thank you for your support! We are glad that we have addressed your concerns.

---

### Official Review · Reviewer_AyDT · 2024-11-03

**Soundness:** 4
**Presentation:** 4
**Contribution:** 4
**Rating:** 8
**Confidence:** 3

**Summary:**

This paper theoretically examines the properties of self-training loops in training generative models. This is an area fo great interest and relatively limited theoretical analysis in current research. This paper aims to prove generalization bounds for self-consuming training loops. As they note, this can be difficult due to the distribution shift across training iterations, as well as the non-i.i.d nature of synthetic-real mixed training datasets. To address the challenge, this paper proposes a novel theoretical notion of recursive algorithmic stability, which allows for controlling the propagation of error across training generations. Leveraging this new theoretical concept, this paper proves a generalization bound. They show that this generalization bound resembles existing qualitative and empirical trends observed in the use of synthetic data. For example, the necessary role of including an appropriate proportion of real data. In addition to their general, model-independent results, this paper also studies the relevant problem of in-context learning in transformers. They provide a bound of the recursive stability of transformers on this task and demonstrate that a constant fraction of real data is sufficient to maintain stability. They additionally remark on the conflicting role played by self-consuming data: it increases the distribution shift in the training loop while decreasing the generalization error component of each step.

**Strengths:**

Overall, I am very supportive of this paper. Self-consuming data loops are currently an important topic in practical foundation model settings --both intentionally from the point of view of synthetic data and unintentionally from the point of view of leakage of model-generated content on the internet. While some existing works have theoretically demonstrated that self-generated data can lead to model collapse, as a practical matter self-generated data can often be leveraged in harmless or beneficial ways -- albeit without theoretical justification. This paper thus makes a significant contribution to placing the role of model-generated data on a more solid theoretical footing. They convincingly isolate some central technical challenges in understanding self-consuming training loops and propose an elegant and intuitive concept of recursive algorithmic stability to confront these challenges. This enables them to make a *key contribution* of presenting the **first** generalization bound on self-consuming training loops. The analysis of the transformer architecture is additionally interesting and instructive, establishing that their proposed recursive stability criterion holds naturally on realistic architectures and tasks. Throughout the paper, they demonstrate that their theory formalizes and quantifies many previous intuitions or folklore on properties of self-consuming training loops (such as the necessity of maintaining a proportion of real data -- or the benefits of synthetic data primarily arising in low real-data regimes).

The paper is extremely clearly written. The proof sketch was insightful, yet accessible. Moreover, the authors interleave their technical and theoretical content with real-world motivation and implications. I found the paper quite enjoyable and educational to read.

**Weaknesses:**

I don't have any serious concerns about this paper. One potential weakness is that the paper primarily focuses on the theoretical angle and  mostly relates their findings to empirical observations in prior work. It could also be interesting if the authors were to add some of their own numerical simulations justifying their findings. Although it is understandable to exclude this, it would be interesting to have some numerical backing in the paper itself. The construction of a simulated setting or dataset would also help in understanding the relationships between the setting studied in this work and real-world setups.

The authors primarily use their theory to explain or justify existing empirical or conceptual observations about self-consuming training loops. However, I feel it would be quite exciting if authors could propose and test (even in simulated settings) some novel predictions made by their theory. Perhaps, this might also be a question of merely highlighting the currently novel predictions made by theory a bit more explicitly.

**Questions:**

1. Perhaps I missed something but what is the notation $\sum\limits_{z_{i} \in S_{i}, \alpha}$ referring to in line 358?
2. To clarify, in the in-context learning setup, the "synthetic data" inputs are not generated by a model but sampled according to the ground-truth distribution correct?
3. Follow-up from (2), I wonder if you could comment on the fact that in many "real-world" synthetic data settings both the "input" and "output" components are sampled from the model. For example, when generating instruction/chat tuning datasets, I believe that both instructions and responses are synthetically generated often. How do you believe that would change your analysis? Seemingly, it would make recursive stability worse due to the additional error in the input distribution.

---

> ### Author Response · Authors · 2024-11-21
> **Numerical Simulations, Notation, and In-Context Learning Setup**
>
> Thank you for your constructive comments and kind support! All your concerns have been carefully addressed as below. The paper has been thoroughly revised, with the revised sections highlighted in blue for clarity. We sincerely hope our responses fully address your questions.
>
> ---
>
> **Q1: It could also be interesting if the authors were to add some of their own numerical simulations justifying their findings. Although it is understandable to exclude this, it would be interesting to have some numerical backing in the paper itself.**
>
> ---
>
> **A:** We thank the reviewer for the suggestion to include numerical simulations to support our theoretical findings. Following this recommendation, we conducted additional experiments during the discussion period to validate our results. Specifically, we trained transformer models to in-context learn linear functions within STLs.
>
> In these experiments, we considered the class of linear functions:
>
> $$
> \mathcal{F} = \\{ f \mid f(x) = w^\top x, w \in \mathbb{R}^d \\},
> $$
>
> in $d=5$ dimensions. We sampled $x_1, \ldots, x_k, x_{\text{query}}$, and $w$ independently from the isotropic Gaussian distribution $\mathcal{N}(0, I_d)$. For each $x_i$, we computed $y_i = w^\top x_i$ and constructed the prompt as:
> $$
> P = (x_1, y_1, x_2, y_2, \ldots, x_k, y_k, x_{\text{query}}).
> $$
>
> We employed a 12-layer, 8-head GPT-2 model with a hidden size of 256, trained on an $\mathbb{R}^5$ linear regression task with 40 in-context examples. Two cases were considered:
> - **Mixed Case:** Fresh data and generated data were mixed in a 0.5 ratio.
> - **Full Synthetic Case:** No fresh data was used.
>
> The results of these experiments are summarized below:
> | #Loop             | 1      | 2      | 3      | 4      |
> |--------------------|--------|--------|--------|--------|
> | Full Synthetic     | 0.3817 | 1.4975 | 1.5396 | 2.0836 |
> | Mixed              | 0.3817 | 0.4208 | 0.4391 | 0.4503 |
>
> As observed, the error accumulates progressively with more self-consuming loops, particularly in the full synthetic case, where the error grows rapidly. In contrast, maintaining a constant-sized proportion of real data effectively reduces the loss, which is consistent with our theoretical findings.
>
> Furthermore, we commit to conducting additional and more comprehensive experiments to strengthen our results in the final version of the paper.
>
> ---
>
> **Q2: Perhaps I missed something but what is the notation $\sum_{z_i \in S_{i, \alpha}}$ referring to in line 358?**
>
> ---
>
> **A:** We sincerely thank the reviewer for their detailed and careful question regarding the notation $\sum_{z_i \in S_{0,\alpha}}$ and $\sum_{z_i \in S_{i,1-\alpha}}$ on line 358.
>
> To clarify, this notation is explained in detail in the appendix (line 822). Specifically,
> $S_{0,\alpha}$ refers to a subset of the real dataset $S_0$,
> where $S_{0,\alpha} \subseteq S_0$ and its size is defined as
> $\alpha \times |S_0|$. In this context, $S_{0,\alpha}$ represents
> a proportion $\alpha$ of the $n$ data points in $S_0$, leading to a total
> of $n \times \alpha$ data points. Similarly, $S_{i,1-\alpha}$ denotes a
> subset of the synthetic dataset $S_i$, where $S_{i,1-\alpha} \subseteq S_i$
> and its size is $(1 - \alpha) \times |S_i|$. This subset, $S_{i,1-\alpha}$,
> contains a proportion $1 - \alpha$ of the $n$ data points in $S_i$,
> resulting in $n \times (1 - \alpha)$ data points.
>
> We greatly appreciate your observation and have revised the paper to explicitly include this explanation
> on line 358 to enhance clarity for all readers.
>
> ---
>
> **Q3: To clarify, in the in-context learning setup, the "synthetic data" inputs are not generated by a model but sampled according to the ground-truth distribution, correct?**
>
> ---
>
> **A:** We thank the reviewer for their question and for seeking clarification. Your understanding is entirely correct. In our work, we follow the standard in-context learning setup as outlined in [1]. Specifically, this involves predicting the label for a given query input. The "synthetic data" in this context refers to query inputs that are known and sampled from the ground-truth distribution. However, their corresponding output labels are generated by the model.
>
> We hope this addresses your concern and are happy to provide further clarifications if needed.

---

> ### Author Response · Authors · 2024-11-21
> **Extension of Theoretical Results to Instruction Tuning**
>
> **Q4: I wonder if you could comment on the fact that in many "real-world" synthetic data settings both the "input" and "output" components are sampled from the model. For example, when generating instruction/chat tuning datasets, I believe that both instructions and responses are synthetically generated often. How do you believe that would change your analysis? Seemingly, it would make recursive stability worse due to the additional error in the input distribution.**
>
> ---
>
> **A:** We sincerely thank the reviewer for this insightful question. While our paper focuses on transformers in in-context learning under STLs, assuming query inputs are drawn from the ground-truth distribution, our theoretical framework extends naturally to Self-Generated Instruction tuning, where both inputs and outputs are synthetically generated.
>
> Our recursive stability analysis does not require query inputs to come from the ground-truth distribution but rather from any known distribution, including synthetic distributions generated by prior model iterations. Following [2], in the Self-Generated Instruction tuning setting, the initial dataset $S_0$ contains ground-truth (human-written) instruction-input-output examples. For the next-generation dataset $S_1$, the generation process involves three steps:
> 1. **Instruction Generation:** The model generates new instructions based on real in-context examples.
> 2. **Input Generation:** The generated instructions are mixed with ground-truth instructions to form in-context examples for generating inputs.
> 3. **Output Generation:** The generated inputs are mixed with ground-truth inputs to form in-context examples for generating outputs.
>
> This process produces a synthetic instruction-input-output dataset for the next generation.
>
> This recursive setup significantly increases the challenge of maintaining stability, as it involves three levels of recursive training per generation, leading to faster error accumulation. For example, in Theorem 2, the generation number $i$ in the recursive stability bound:
>
> $$
> \gamma_n^{i,\alpha}\lesssim
> (1-\alpha)^i \frac{\widetilde{B}_W^{(i+1)L}}{2n+1},
> $$
> scales to $3i$, yielding:
> $$
> \gamma_n^{i,\alpha}\lesssim
> (1-\alpha)^{3i} \frac{\widetilde{B}_W^{(3i+3)L}}{2n+1}.
> $$
>
> This escalation makes controlling recursive stability more difficult, particularly when real data is scarce, as the stability parameter decays at a faster exponential rate. However, by setting $\alpha$ to $1 - \widetilde{B}_W^{-L}$, recursive stability can still be effectively managed.
>
>
> Reference:
>
> $\left[1\right]$ Dong Q, Li L, Dai D, et al. A survey on in-context learning[J]. arXiv preprint arXiv, 2022.
>
> $\left[2\right]$ Wang Y, Kordi Y, Mishra S, et al. Self-Instruct: Aligning Language Models with Self-Generated Instructions[C]. ACL. 2023.

---

> > ### Comment · Reviewer_AyDT · 2024-11-26
> > **Thank you for your reply**
> >
> > I thank the authors for their response. I will maintain my score but would like to say that my questions have been addressed well.

---

> > > ### Author Response · Authors · 2024-11-27
> > > **Thank you**
> > >
> > > Thank you for your support! We are pleased to know that your concerns have been effectively addressed.

---

### Official Review · Reviewer_CLcc · 2024-11-03

**Soundness:** 2
**Presentation:** 2
**Contribution:** 2
**Rating:** 6
**Confidence:** 3

**Summary:**

This paper theoretically studies Self-consuming Training Loops (STLs), where generative models generate their own training data. The key concept that the paper introduces is the recursive stability which the authors prove to be vital for generalization and for avoiding model collapse. After the general result on how the recursive stability is important, the authors move to provide an upper bound for the recursive ability for transformers. The theoretical results suggest a trade-off of synthetic data augmentation.

**Strengths:**

* The paper introduces a concept called recursive stability that addresses the complexity and difficulty of self-consuming training loops. It can be used to derive an upper bound on the generalization error of learning algorithms trained on the self generated (mixture of self-generated and real) dataset after any iterations of the self-consuming loop.

* The application to transformers is nice to have as it bridges to the most popular architecture in practice.

**Weaknesses:**

*  The reviewer is concerned about the validity of this recursive stability which is supposed to be the core contribution of this paper. Details follow. The recursive stability $\gamma^i_n$ is a rather strong assumption, and it hides a lot of things underneath. For example, it is uniform not only in terms of the data point modification, but also uniform w.r.t. to any randomness in the STL process. It also depends on how the STL is performed, e.g., the ratio $\alpha$ of the data mixture. The paper derives an upper-bound on the  recursive stability $\gamma^i_n$ for transformer in Theorem 2 but the bound has an exponential factor $e^{B_WL}$ where $B_W$ is the bound on the norm of weights and $L$ is the depth. So it appears that $\alpha$ is required to be very close to $1$, i.e., almost only using real data, for the guarantee on model collapse to stand.
* Following up on the above concern, it is not really satisfying to see the paper jumps directly to the application to transformers in-context learning which is notoriously difficult to make the theory accurate. Why not apply the recursive stability and Theorem 1 to simple cases first, e.g., Gaussian models, where everything can be made accurate, to examine if the proposed recursive is indeed a valid property to look at? This means to examine the LHS and RHS of Theorem 1 accurately for simple cases and show that the bound is not vacuous.

----------------after rebuttal--------------------

The authors have well addressed my major concerns, so the score is raised.

**Questions:**

see the questions in the above section.

---

> ### Author Response · Authors · 2024-11-21
> **Clarification on Recursive Stability and the ratio $\alpha$**
>
> Thank you for your thorough review and constructive comments. All your concerns have been carefully addressed as below. The paper has been thoroughly revised, with the revised sections highlighted in blue for clarity. We sincerely hope our responses fully address your questions.
>
> ---
>
> **Q1: The recursive stability $\gamma_n^i$ is a rather strong assumption, and it hides a lot of things underneath. For example, it is uniform not only in terms of the data point modification but also uniform w.r.t. to any randomness in the STL process. It also depends on how the STL is performed, e.g., the ratio $\alpha$ of the data mixture. The paper derives an upper-bound on the recursive stability $\gamma_n^i$ for transformer in Theorem 2, but the bound has an exponential factor $e^{B_W L}$, where $B_W$ is the bound on the norm of weights and $L$ is the depth. So it appears that $\alpha$ is required to be very close to 1, i.e., almost only using real data, for the guarantee on model collapse to stand.**
>
> ---
>
> **A:** Thank you for your thoughtful feedback. We fully understand your concerns, particularly regarding two key points: (1) whether recursive stability constitutes an overly strict assumption, and (2) whether the derived bounds imply that $\alpha$ must be very close to 1. Below, we address each point in detail.
>
> ### 1. Recursive Stability as a Metric, Not a Strong Assumption
> We respectfully clarify that recursive stability is introduced as an evaluation metric rather than a strong assumption.
>
> Similar to how algorithmic stability is a foundational concept in generalization theory for quantifying a learning algorithm's sensitivity to data perturbations [1], recursive stability serves as a critical metric for evaluating the sensitivity of generative models in STLs to perturbations in the initial data. A wide range of generative models exhibit recursive stability at varying levels. Beyond the transformer architecture analyzed in this paper, we also provide an upper bound for the recursive stability of Gaussian Mixture Models (GMMs) in the subsequent discussion. This metric offers valuable insights into how model architecture and the real-synthetic data ratio $\alpha$ influence STL performance, enabling the identification of scenarios where recursive stability grows rapidly (leading to model collapse) and those where stability is maintained.
>
> Additionally, following the reviewer’s suggestion, we have revised our definition of recursive stability to explicitly express the role of the ratio $\alpha$:
>
> **Definition 2. (Recursive Stability)**
> Let $S_0$ represent the original real dataset, and $S'_0$ denote a dataset differing from $S_0$ by a single example. A generative model $\mathcal{G}$ is said to be recursively $\gamma_n^{i,\alpha}$-stable with respect to the distance measure $d$ after the $i$-th generation of STLs, where the ratio of real to synthetic data is set to $\alpha$, if the following condition holds:
> $$
> \forall S_0, S'_0 \in \mathbb{Z}^n, \quad d\left(\mathcal{G}^{(i)}(S_0), \mathcal{G}^{(i)}(S'_0)\right) \leq \gamma_n^{i,\alpha}.
> $$
>
> ### 2. Ratio $\alpha$ Does Not Need to Be Close to 1
> We clarify that the requirements for the ratio $\alpha$ vary across different generative model architectures. For instance, as demonstrated in our response to the next question, when $\alpha = 0.1$, the recursive stability of the GMM can still be effectively controlled, thereby preventing model collapse. This observation aligns with the experimental results reported in [3].
>
> For transformers, the depth $L$ is typically **small** in practical settings. For example, studies on LLM performance in STLs, such as [2], often employ models with $L = 6$. Furthermore, techniques like layer normalization effectively **constrain the norm of weights** $B_W$ to values close to 1, ensuring numerical stability during training. Therefore, in practical scenarios, the combination of small $L$ and well-controlled $B_W$ ensures that the recursive stability bound does not necessitate $\alpha$ being excessively close to 1.
>
> We also acknowledge that addressing the exponential dependency on $L$ in convergence analyses for in-context learning is indeed an important research direction. Existing works, such as [4], have started tackling this challenge. However, this issue is beyond the primary scope of our paper. Our core contribution lies in presenting the first theoretical generalization analysis that demonstrates how model architecture and the real-synthetic data ratio $\alpha$ influence the performance and success of STLs.
>
> We sincerely thank the reviewer for their detailed observations. In the final version of the paper, we will incorporate a discussion on recursive stability’s role as a metric, the practical implications of $L$ and $B_W$, and its adaptability to different models and settings.

---

> ### Author Response · Authors · 2024-11-21
> **Validating Recursive Stability with Gaussian Mixture Models**
>
> **Q2: It is not really satisfying to see the paper jumps directly to the application to transformers in-context learning which is notoriously difficult to make the theory accurate. Why not apply the recursive stability and Theorem 1 to simple cases first, e.g., Gaussian models, where everything can be made accurate, to examine if the proposed recursive is indeed a valid property to look at? This means to examine the LHS and RHS of Theorem 1 accurately for simple cases and show that the bound is not vacuous.**
>
> ---
>
> **A:** Thank you for your insightful suggestion. We fully agree on the value of validating our theoretical framework with simpler models to ensure its applicability and robustness. Following your guidance, we have applied our recursive stability metric and the general generalization bound (Theorem 1) to a Gaussian mixture model.
>
> We adopt the setup from prior works [5,6] and consider a binary classification task where $Y =\\{-1, 1\\}$. Given a vector $\mu \in \mathbb{R}^d$ with $\\|\mu\\|_2 = 1$ and noise variance $\sigma^2 > 0$, the data distribution is specified as follows: $y \sim \text{uniform}\\{-1, 1\\}$ and $x \mid y \sim \mathcal{N}(y \mu, \sigma^2 I_d)$. We define the conditional generative model using parameters $\mu_y$ and $\sigma_k^2$, where $y \in \\{-1, 1\\}$ and $k \in [d]$. For $n$ data points, let $n_y$ represent the number of samples in class $y$. The means $\hat{\mu}_y$ of the Gaussian mixture model are estimated as
>
> $$\frac{\sum_{y_i = y} x_i}{n_y},$$
>
> while the variances $\hat{\sigma}_k^2$ are calculated as
>
> $$\sum_y \frac{n_y}{n} \frac{\sum_{y_i = y} (x_{ik} - \hat{\mu}_{yk})^2}{n_y - 1}.$$
>
> Then we can generate new samples from the distribution: $y \sim \text{uniform}\\{-1, 1\\}$ and $x \mid y \sim \mathcal{N}(\hat{\mu}_y, \Sigma)$.
>
>  Additionally, the learning algorithm functions as a linear classifier, parameterized by $\theta \in \mathbb{R}^d$, with predictions given by:
> $$
> \hat{y} = \text{sign}(\theta^\top \mathbf{x}).
> $$
> The loss function is defined as:
> $$
> \ell(\theta, (x, y)) = \frac{1}{2\sigma^2} (x - y\theta)^\top (x - y\theta).
> $$
> Thus, the output is:
> $$
> \hat{\theta} = \frac{1}{m} \sum_{i=1}^m y_i x_i.
> $$
>
> In this setting, we demonstrate recursive stability for the Gaussian mixture model as follows:
>
> ---
> **Theorem (Recursive Stability)**: Let $S_0, S_0'$ denote two initial real datasets differing by a single example. Let $n$ represent the sample size of the mixed dataset $\tilde{S}_j$, where $\tilde{S}_j = \alpha S_0 + (1 - \alpha) S_j$ for $1 \leq j \leq i$. Choose $m = \mathcal{O}(\sqrt{n})$. Consider the previously described sampling and learning steps, where real data samples are drawn from the Gaussian Mixture Model distribution $\mathcal{D}$, and the synthetic data for the $i$-th generation is generated from the learned Gaussian Mixture distribution of the $i$-th generation. Then with probability at least $1-\delta$, we have:
> $$
> \gamma_n^{i,\alpha} \lesssim n^{-1/2} \alpha^{-1}(1-(1-\alpha)^i)\log (nd/\delta),
> $$
> where the measure for the recursive stability parameter is taken as the KL divergence.
>
> ---
>
> As $\alpha$ approaches 0, indicating that no real data is incorporated during each generation of training, we observe:
> $$
> \gamma_n^i \lesssim i n^{-1/2} \log \frac{nd}{\delta},
> $$
> which suggests a linear accumulation of errors. This finding aligns closely with the theoretical insights presented in [3,7], where a Gaussian model trained without real data demonstrated a linear divergence in variance. Thus, this underscores the validity of our theoretical results, confirming that the derived bound is meaningful and not vacuous.
>
> Moreover, by leveraging the generalization error bound established in Theorem 1, we derive the following:
>
> ---
>
> **Theorem (Generalization Error Bound)**: Consider the Gaussian Mixture Model in the setting outlined above. Let $n$ represent the sample size of the mixed dataset $\tilde{S}_j$, where $\tilde{S}_j = \alpha S_0 + (1 - \alpha) S_j$ for $1 \leq j \leq i$. Suppose the loss function is defined above. Let $\mathcal{A}(\tilde{S}_i)$ denote the output of applying the linear classifier described above to the mixed dataset $\tilde{S}_i$. Then, for any $\delta \in (0,1)$, with probability at least $1-\delta$, the following holds:
>
> $$
>  |Generalization| \lesssim n^{-1/2}(d+\log(n/\delta))\log n\log (1/\delta)
> +n^{-1/4}(1-(1-\alpha)^i)\alpha^{-1}(d+\log (n/\delta))\sqrt{d\log(nd/\delta)}.
> $$
>
> ---
>
> We observe that when $\alpha$ is set to a constant (e.g., $\alpha = 0.1$), the generalization error can be effectively controlled, preventing model collapse. This result aligns with the experimental findings in [2] for Gaussian models.
>
> The above discussions have been carefully added to our paper to strengthen the validation of our theoretical framework and highlight its applicability to different generative model settings. Thank you for your valuable feedback.

---

> ### Author Response · Authors · 2024-11-21
> **References**
>
> References:
>
> $\left[1\right]$ Bousquet O, Elisseeff A. Stability and generalization[J]. The Journal of Machine Learning Research, 2002, 2: 499-526.
>
> $\left[2\right]$ Briesch M, Sobania D, Rothlauf F. Large language models suffer from their own output: An analysis of the self-consuming training loop[J]. arXiv 2023.
>
> $\left[3\right]$ Alemohammad S, Casco-Rodriguez J, Luzi L, et al. Self-Consuming Generative Models Go MAD[C]//The Twelfth International Conference on Learning Representations.
>
> $\left[4\right]$ Chen B, Li X, Liang Y, et al. Bypassing the Exponential Dependency: Looped Transformers Efficiently Learn In-context by Multi-step Gradient Descent[J]. arXiv 2024.
>
> $\left[5\right]$ He H, Yan H, Tan V Y F. Information-theoretic characterization of the generalization error for iterative semi-supervised learning[J]. Journal of Machine Learning Research, 2022.
>
> $\left[6\right]$ Zheng C, Wu G, Li C. Toward understanding generative data augmentation[J]. Advances in neural information processing systems, 2023.
>
> $\left[7\right]$ Shumailov I, Shumaylov Z, Zhao Y, et al. AI models collapse when trained on recursively generated data[J]. Nature, 2024.

---

> ### Comment · Reviewer_CLcc · 2024-11-26
>
> I appreciate the authors’ additional efforts to strengthen their theoretical argument. Since both of my major concerns are well addressed, I have raised my score.

---

> > ### Author Response · Authors · 2024-11-27
> > **Thank you**
> >
> > Thank you for your thoughtful feedback. Your suggestions have been invaluable in enhancing the quality of our paper, and we’re delighted to know that your concerns have been fully resolved.

---

### Author Response · Authors · 2024-11-22
**Resolving Formula Rendering Issues: Use Google Chrome or Refresh the Page**

We report a potential technical limitation on the OpenReview platform that some readers have observed. Specifically, mathematical formulas in official comments uploaded **may not render** correctly in certain situations.

Our investigation shows that this issue can typically be resolved by opening the page using the **Google Chrome** browser or **refreshing** the page a few times.

We apologize for any inconvenience this may cause and appreciate your understanding. Please feel free to reach out if you encounter further difficulties.

---

### Author Response · Authors · 2024-11-24
**Kindly Reminder: Discussion Period Ending Soon**

Dear Reviewers,

We sincerely appreciate the time and effort you have dedicated to reviewing our manuscript. Your expert opinions and constructive feedback are invaluable in helping us improve the quality of our work.

We are writing to kindly remind you that the rebuttal discussion period will conclude in **less than three days**. We would be most grateful if you could provide any further comments or suggestions at your earliest convenience. It is important for us to confirm whether our responses have adequately addressed your concerns, and your additional input would greatly contribute to enhancing our manuscript.

We look forward to your valuable feedback and a productive discussion. Thank you once again for your time and thoughtful consideration.

Best regards,

The Authors.

---

### Meta-Review · Area_Chair_kuKP · 2024-12-20

**Metareview:**

This work provides theoretical studies for the setting of Self-consuming Training Loops (STLs), where the generative model increasingly generate their own data for further training. As the reviewers' reach the consensus that the problem setting is interesting and challenging, the contribution is sound and the paper is well-written, it is recommended to appear in the conference for our community. Due to the vibrant and fruitful discussion, incorporating additional comments and discussions (as some of them already been updated) would be beneficial for the manuscript.

**Additional Comments On Reviewer Discussion:**

The reviewers concerns are well-addressed.

---

### Decision · Program_Chairs · 2025-01-22

Accept (Poster)